# Self-supervised Transfer Learning via Adversarial Contrastive Training

## Abstract

Learning a data representation with strong transferability from an unlabeled scenario is both crucial and challenging. In this paper, we propose a novel self-supervised transfer learning approach via Adversarial Contrastive Training (ACT). Additionally, we establish an end-to-end theoretical understanding for self-supervised contrastive pretraining and its implications for downstream classification tasks in a misspecified, over-parameterized setting. Our theoretical findings highlight the provable advantages of adversarial contrastive training in the source domain towards improving the accuracy of downstream tasks in the target domain. Furthermore, we illustrate that downstream tasks necessitate only a minimal sample size when working with a well-trained representation, offering valuable insights on few-shot learning. Last but not least, extensive experiments across various datasets demonstrate a significant enhancement in classification accuracy when compared to existing state-of-the-art self-supervised learning methods.

## 1 Introduction

Collecting unlabeled data is far more convenient and cost-effective than gathering labeled data in real-world applications. As a result, learning representations from abundant unlabeled data has become a critical and foundational challenge. Pretraining on unlabeled data enables the capture of more general, abstract features without the need for specific labels. Consequently, the learned task-invariant representations demonstrate superior transferability to unseen data, making them highly effective in transfer learning scenarios.

One of the most popular approaches to learning representations from unlabeled data is self-supervised contrastive learning, which has garnered significant attention due to its impressive performance. The rationale behind contrastive learning involves acquiring a representation that maintains augmentation invariance while preventing model collapse. The latter aspect is crucial, as solely bringing positive pairs closer could result in trivial solutions. The initial body of work heavily relies on the utilization of negative samples, such as Ye et al. (2019); He et al. (2020); Chen et al. (2020a;b); HaoChen et al. (2021); Zhang et al. (2023). These studies prevent representation collapse by pushing negative pairs apart in the feature space. However, the construction of negative pairs poses significant challenges. Firstly, augmented views from distinct data points sharing the same semantic meaning may inadvertently be treated as negative pairs, impeding semantic extraction. Secondly, the quality of the representation is highly dependent on the number of negative pairs, necessitating substantial computational and memory resources.

In recent years, there has been a surge of interests in developing self-supervised learning methods that eschew the use of negative samples (Grill et al., 2020; Caron et al., 2020; 2021; Ermolov et al., 2021; Zbontar et al., 2021; Chen & He, 2021; Bardes et al., 2022; Ozsoy et al., 2022; HaoChen et al., 2022; Wang et al., 2024). Among above mentioned studies, the most prominent works include Zbontar et al. (2021); Bardes et al. (2022); Ozsoy et al. (2022); HaoChen et al. (2022); Zhang et al. (2023), which prevent the model collapse by incorporating a regularization term into the loss function. However, as demonstrated later, either the population counterpart of Zbontar et al. (2021); Bardes et al. (2022) is still under-investigated, or the sample version of population losses (HaoChen et al., 2022; HaoChen & Ma, 2023) exhibits bias, presenting a significant challenge in terms of theoretical analysis. Moreover, due to this bias, the learned representation does not close to the

minimizer of the population loss. Specifically, when trained on mini-batch data, the limited sample size in each mini-batch can amplify the bias, leading to accuracy loss, as shown in Table 1.

In this study, we introduce a novel self-supervised learning approach called **A**dversarial **C**ontrastive **T**raining (ACT), designed to learn representations without the need for constructing negative samples, while avoiding the bias between population loss and sample-level loss. Particularly, let

$$\mathcal{R}(f, G) = \left\langle \mathbb{E}_{\boldsymbol{x}} \mathbb{E}_{\boldsymbol{x}_1, \boldsymbol{x}_2 \in \mathcal{A}(\boldsymbol{x})} \left[ f(\boldsymbol{x}_1) f(\boldsymbol{x}_2)^\top \right] - I_{d^*}, G \right\rangle_F, \tag{1}$$

where $f : \mathbb{R}^d \to \mathbb{R}^{d^*}$ is a representation function, $G$ is a matrix in $\mathbb{R}^{d^* \times d^*}$, and the Frobenius inner product is defined as $\langle \boldsymbol{A}, \boldsymbol{B} \rangle_F := \text{tr}(\boldsymbol{A}^\top \boldsymbol{B})$ for any $\boldsymbol{A}, \boldsymbol{B} \in \mathbb{R}^{d_1 \times d_2}$. Then we learn the contrastive representation through a minimax optimization problem

$$\min_f \max_{G \in \mathcal{G}(f)} \mathcal{L}(f, G) = \mathbb{E}_{\boldsymbol{x}} \mathbb{E}_{\boldsymbol{x}_1, \boldsymbol{x}_2 \in \mathcal{A}(\boldsymbol{x})} \left[ \| f(\boldsymbol{x}_1) - f(\boldsymbol{x}_2) \|_2^2 \right] + \lambda \mathcal{R}(f, G), \tag{2}$$

where the first term in (2) facilitates achieving augmentation invariance in the representation, similar with the previous works (Zbontar et al., 2021; Bardes et al., 2022; HaoChen et al., 2022). Here $\mathcal{A}(\boldsymbol{x})$ denotes the set of augmentations of a sample $\boldsymbol{x}$, $\lambda > 0$ is the regularization parameter and $\mathcal{G}(f) := \left\{ G \in \mathbb{R}^{d^* \times d^*} : \|G\|_F \leq \|\mathbb{E}_{\boldsymbol{x}} \mathbb{E}_{\boldsymbol{x}_1, \boldsymbol{x}_2 \in \mathcal{A}(\boldsymbol{x})} [f(\boldsymbol{x}_1) f(\boldsymbol{x}_2)^\top] - I_{d^*} \|_F \right\}$ is the feasible set of $G$. In fact, the inner maximization problem has a explicit solution that $G = \mathbb{E}_{\boldsymbol{x}} \mathbb{E}_{\boldsymbol{x}_1, \boldsymbol{x}_2 \in \mathcal{A}(\boldsymbol{x})} [f(\boldsymbol{x}_1) f(\boldsymbol{x}_2)^\top] - I_{d^*}$, therefore (2) is equivalent to minimizing following loss

$$\mathcal{L}(f) := \mathbb{E}_{\boldsymbol{x}} \mathbb{E}_{\boldsymbol{x}_1, \boldsymbol{x}_2 \in \mathcal{A}(\boldsymbol{x})} \left[ \| f(\boldsymbol{x}_1) - f(\boldsymbol{x}_2) \|_2^2 \right] + \lambda \left\| \mathbb{E}_{\boldsymbol{x}} \mathbb{E}_{\boldsymbol{x}_1, \boldsymbol{x}_2 \in \mathcal{A}(\boldsymbol{x})} [f(\boldsymbol{x}_1) f(\boldsymbol{x}_2)^\top] - I_{d^*} \right\|_F^2. \tag{3}$$

The second term in $\mathcal{L}(f)$ encourages the separation of category centers within the latent space, thereby avoiding collapse and improving classification accuracy in downstream tasks, so as $\mathcal{R}(f, G)$. More details can be found in Appendix A. Thanks to the minimax formulation in (2), we propose the following loss of our ACT at the sample level

$$\widehat{\mathcal{L}}(f, G) := \frac{1}{n_s} \sum_{i=1}^{n_s} \left[ \| f(\boldsymbol{x}_1^{(i)}) - f(\boldsymbol{x}_2^{(i)}) \|_2^2 + \lambda \left\langle f(\boldsymbol{x}_1^{(i)}) f(\boldsymbol{x}_2^{(i)})^\top - I_{d^*}, G \right\rangle_F \right], \tag{4}$$

where $\boldsymbol{x}^{(1)}, \ldots, \boldsymbol{x}^{(n_s)}$ are unlabeled data, $\boldsymbol{x}_1^{(i)}$ and $\boldsymbol{x}_2^{(i)}$ are independent augmentations of $\boldsymbol{x}^{(i)}$. It can be shown that (4) is unbiased in the sense that $\mathbb{E}_{D_s}[\widehat{\mathcal{L}}(f, G)] = \mathcal{L}(f, G)$ for each fixed $G \in \mathbb{R}^{d^* \times d^*}$.

However, directly discretizing the expectation in (3) yields a biased sample-level loss as

$$\widehat{\mathcal{L}}(f) := \frac{1}{n_s} \sum_{i=1}^{n_s} \| f(\boldsymbol{x}_1^{(i)}) - f(\boldsymbol{x}_2^{(i)}) \|_2^2 + \lambda \left\| \frac{1}{n_s} \sum_{i=1}^{n_s} f(\boldsymbol{x}_1^{(i)}) f(\boldsymbol{x}_2^{(i)})^\top - I_{d^*} \right\|_F^2.$$

Specifically, we have $\mathbb{E}_{D_s}[\widehat{\mathcal{L}}(f)] \neq \mathcal{L}(f)$ due to the non-commutativity between the expectation and the Frobenius norm, where $D_s$ represents the dataset used for pretraining. While this biased discretization method has been employed in previous studies (HaoChen et al., 2022; HaoChen & Ma, 2023), its application presents a significant challenge in terms of theoretical analysis. For instance, despite that Huang et al. (2023) establish a theoretical analysis for Zbontar et al. (2021) at the population-level, the extensions of these findings to the sample-level is not straightforward due to the bias of the estimation. HaoChen & Ma (2023) establish a theoretical understanding for HaoChen et al. (2022) at the sample-level, nonetheless, the results are subject to strong assumptions given the biased nature of the estimation.

From a theoretical perspective, we establish a rigorous end-to-end theoretical analysis for both the contrastive pre-training and the downstream classification under mild conditions. Further, our findings demonstrate the provable advantages of self-supervised contrastive pre-training and provides theoretical insights into determining the sample size and selecting the appropriate scale for deep neural networks. Our experiment yields remarkable classification accuracy when employing both fine-tuned linear probes and the $K$-nearest neighbor ($K$-NN) protocol across a range of benchmark datasets. These results showcase a high level of competitiveness with current state-of-the-art self-supervised learning methodologies, as illustrated in Table 1.

## 1.1 RELATED WORK

**Self-supervised transfer learning**   Thanks to the robust transferability inherent in representations learned by self-supervised learning, the field of few-shot learning, which aims to train models with only a limited number of labeled samples, has significantly advanced through self-supervised methodologies. This progression is evidenced by the contributions of Liu et al. (2021); Rizve et al. (2021); Yang et al. (2022); Lim et al. (2023). However, current work only demonstrates the effectiveness of self-supervised learning for few-shot learning mainly empirically. Theoretical explanations remain scarce. Understanding how the learned representations from unlabeled data enhance prediction performance with only a few labeled samples in downstream tasks is a critical question that requires further investigation. Especially investigating the impact of upstream samples on downstream samples. Therefore, a thorough theoretical analysis at sample level is urgently needed.

Although Saunshi et al. (2019); HaoChen et al. (2021); Garrido et al. (2022); Awasthi et al. (2022); Ash et al. (2022); HaoChen et al. (2022); Lei et al. (2023); Huang et al. (2023) have offered some theoretical progresses in understanding self-supervised learning, all these studies either remain at the population level, or focus solely on the generalization property of hypothesis space with a finite complexity measure. The effects of both upstream and downstream sample sizes are still unknown.

HaoChen & Ma (2023) use augmented graphs to provide a more thorough theoretical analysis at sample level for the self-supervised learning loss proposed in HaoChen et al. (2022). They establish a theoretical guarantees at the sample level, under certain strong assumptions, including Assumptions 4.2 and 4.4. Assumption 4.2 assumes the existence of a neural network capable of sufficiently minimize the loss. In contrast, we demonstrate the existence of a measurable function that can vanish our loss by accounting for additional approximation error. This necessitates an extension of the well-specified setting to a misspecified setting. Moreover, the most important problem in self-supervised transfer learning theory pertains to elucidating the mechanism through which the representation acquired from the upstream task facilitates the learning process of the downstream task. While HaoChen & Ma (2023) assume this relationship as Assumption 4.4 in their research, our study surpasses the current body of literature by conducting a comprehensive investigation into the impact of approximation error and generalization error during the pre-training phase on downstream test error. This analysis sheds light on how the size of the upstream sample influences the downstream task, particularly in scenarios where the availability of downstream samples is constrained.

**Comparison with existing contrastive learning algorithms**   HaoChen et al. (2022) can be regarded as a special version of our model with the constraint $x_1 = x_2$ at the population level. However, its loss at the sample level adopts a biased discretization method, which leads to a different optimization direction compared to ACT, especially in the mini-batch scenario. More discussion can be found in Remark 2.1. Besides that, the loss at the sample level provided by Zbontar et al. (2021) is also similar to our loss, but its unbiased counterpart at the population level is still unknown.

## 1.2 CONTRIBUTIONS

Our main contributions can be summarized as follows.

- We introduce a novel self-supervised transfer learning method called **A**dversarial **C**ontrastive **T**raining (ACT). This approach learns representations from unlabeled data by tackling a minimax optimization problem, which aims to de-bias the initially proposed risk, thereby providing a foundation for establishing a thorough theoretical understanding.

- Our experimental results demonstrate outstanding classification accuracy using both fine-tuned linear probe and $K$-nearest neighbor ($K$-NN) protocol on various benchmark datasets, showing competitiveness with existing state-of-the-art self-supervised learning methods.

- In the context of transfer learning, we present a thorough theoretical understanding for both ACT and its downstream classification tasks within a misspecified and overparameterized scenario. Our theoretical results offer insights into determining the samples size for pre-training and appropriate depth, width, and norm restrictions of neural networks. These findings illuminate the advantages of ACT in enhancing the accuracy of downstream tasks.

Furthermore, we demonstrate that leveraging the representations learned by ACT in the source domain enables high accuracy in the downstream tasks of the target domain, even when only a small amount of data is available.

## 1.3 ORGANIZATIONS

The remainder of this paper is organized as follows. In Section 2, we introduce basic notations and presents the adversarial self-supervised learning loss, along with an alternating optimization algorithm to address the minimax problem. Section 3 showcases experimental results for representations learned by ACT across various real datasets and evaluation protocols. Section 4 provides an end-to-end theoretical guarantee for ACT. Conclusions are discussed in Section 5, respectively. Detailed proofs and experimental details are differed to Section B and C respectively.

## 2 ADVERSARIAL CONTRASTIVE TRAINING

In this section, we provide a novel method for unsupervised transfer learning via adversarial contrastive training (ACT). We begin with some notations in Section 2.1. Then, we introduce ACT method and alternating optimization algorithm in Section 2.2. Finally, we outline the setup of the downstream task in Section 2.3.

### 2.1 PRELIMINARIES AND NOTATIONS

Denote by $\|\cdot\|_2$ and $\|\cdot\|_\infty$ the 2-norm and $\infty$-norm of the vector, respectively. Let $\boldsymbol{A}, \boldsymbol{B} \in \mathbb{R}^{d_1 \times d_2}$ be two matrices. Define the Frobenius inner product $\langle \boldsymbol{A}, \boldsymbol{B} \rangle_F = \mathrm{tr}(\boldsymbol{A}^\top \boldsymbol{B})$. Denote by $\|\cdot\|_F$ the Frobenius norm induced by Frobenius inner product. We denote the $\infty$-norm of the matrix as $\|\boldsymbol{A}\|_\infty := \sup_{\|\boldsymbol{x}\|_\infty \leq 1} \|\boldsymbol{A}\boldsymbol{x}\|_\infty$, which is the maximum 1-norm of the rows of $\boldsymbol{A}$. The Lipschitz norm of a map $f$ from $\mathbb{R}^{d_1}$ to $\mathbb{R}^{d_2}$ is defined as $\|f\|_{\mathrm{Lip}} := \sup_{\boldsymbol{x} \neq \boldsymbol{y}} \frac{\|f(\boldsymbol{x})-f(\boldsymbol{y})\|_2}{\|\boldsymbol{x}-\boldsymbol{y}\|_2}$.

Let $L, N_1, \ldots, N_L \in \mathbb{N}, 0 < B_1 \leq B_2$. A deep ReLU neural network hypothesis space is defined as

$$\mathcal{NN}_{d_1,d_2}(W, L, \mathcal{K}, B_1, B_2) := \left\{ \begin{matrix} \phi_{\boldsymbol{\theta}}(\boldsymbol{x}) = \boldsymbol{A}_L \sigma(\boldsymbol{A}_{L-1}\sigma(\cdots \sigma(\boldsymbol{A}_0 \boldsymbol{x} + \boldsymbol{b}_0)) + \boldsymbol{b}_{L-1}), \\ W = \max\{N_1, \ldots, N_L\}, \ \kappa(\boldsymbol{\theta}) \leq \mathcal{K}, \ B_1 \leq \|\phi_{\boldsymbol{\theta}}\|_2 \leq B_2 \end{matrix} \right\},$$

where $\sigma(x) := x \vee 0$ is the ReLU activate function, $N_0 = d_1$, $N_{L+1} = d_2$, $\boldsymbol{A}_i \in \mathbb{R}^{N_{i+1} \times N_i}$ and $\boldsymbol{b}_i \in \mathbb{R}^{N_{i+1}}$. The integers $W$ and $L$ are called the width and depth of the neural network, respectively. $B_1 \leq \|\phi_{\boldsymbol{\theta}}\|_2 \leq B_2$ is used to indicate any $\boldsymbol{u} \in [0,1]^d$, $B_1 \leq \|\phi_{\boldsymbol{\theta}}(\boldsymbol{u})\|_2 \leq B_2$. The parameters set of the neural network is defined as $\boldsymbol{\theta} := ((\boldsymbol{A}_0, \boldsymbol{b}_0), \ldots, (\boldsymbol{A}_{L-1}, \boldsymbol{b}_{L-1}), \boldsymbol{A}_L)$. Further, $\kappa(\boldsymbol{\theta})$ is defined as

$$\kappa(\boldsymbol{\theta}) := \|\boldsymbol{A}_L\|_\infty \prod_{l=0}^{L-1} \max\{\|(\boldsymbol{A}_l, \boldsymbol{b}_l)\|_\infty, 1\}.$$

Appendix B.1 shows that $\|\phi_{\boldsymbol{\theta}}\|_{\mathrm{Lip}} \leq \mathcal{K}$ for each $\phi_{\boldsymbol{\theta}} \in \mathcal{NN}_{d_1,d_2}(W, L, \mathcal{K}, B_1, B_2)$.

### 2.2 ADVERSARIAL CONTRASTVE TRAINING

Learning representations from large amounts of unlabeled data has recently gained significant attention, as highly transferable representations offer substantial benefits for downstream tasks. Adversarial contrastve training is driven by two key factors: augmentation invariance and a regularization term to prevent model collapse. Specifically, augmentation invariance aims to make representations of different augmented views of the same sample as similar as possible. However, a trivial representation that maps all augmented views to the same point is ineffective for downstream tasks, making the regularization term essential.

Data augmentation $A : \mathbb{R}^d \to \mathbb{R}^d$ is essentially a transformation of the original sample before training. A commonly-used augmentation is the composition of random transformations, such as RandomCrop, HorizontalFlip, and Color distortion (Chen et al., 2020a). Denote by $\mathcal{A} = \{A_\gamma(\cdot) : \gamma \in [m]\}$ the collection of data augmentations, and denote the source domain as $\mathcal{X}_s \subseteq [0,1]^d$, with

its corresponding unknown distribution denoted by $P_s$. Let $\{\boldsymbol{x}^{(1)}, \ldots, \boldsymbol{x}^{(n_s)}\}$ be $n_s$ i.i.d. unlabeled samples from the source distribution. For each sample $\boldsymbol{x}^{(i)}$, we define the corresponding augmented pair as

$$\tilde{\boldsymbol{x}}^{(i)} = (\boldsymbol{x}_1^{(i)}, \boldsymbol{x}_2^{(i)}) = (A(\boldsymbol{x}^{(i)}), A'(\boldsymbol{x}^{(i)})), \tag{5}$$

where $A$ and $A'$ are drawn from the uniform distribution on $\mathcal{A}$ independently. Further, the augmented dataset for ACT is defined as $D_s := \{\tilde{\boldsymbol{x}}^{(i)}\}_{i \in [n_s]}$.

The ACT method can be formulated as a minimax problem

$$\hat{f}_{n_s} \in \arg\min_{f \in \mathcal{F}} \sup_{G \in \widehat{\mathcal{G}}(f)} \widehat{\mathcal{L}}(f, G), \tag{6}$$

where the empirical risk is defined as

$$\widehat{\mathcal{L}}(f, G) := \frac{1}{n_s} \sum_{i=1}^{n_s} \left[ \|f(\boldsymbol{x}_1^{(i)}) - f(\boldsymbol{x}_2^{(i)})\|_2^2 + \lambda \langle f(\boldsymbol{x}_1^{(i)}) f(\boldsymbol{x}_2^{(i)})^\top - I_{d^*}, G \rangle_F \right], \tag{7}$$

and $\lambda > 0$ is the regularization parameter, the hypothesis space $\mathcal{F}$ is chosen as the neural network class $\mathcal{NN}_{d,d^*}(W, L, \mathcal{K}, B_1, B_2)$, and the feasible set $\widehat{\mathcal{G}}(f)$ is defined as

$$\widehat{\mathcal{G}}(f) := \left\{ G \in \mathbb{R}^{d^* \times d^*} : \|G\|_F \leq \left\| \frac{1}{n_s} \sum_{i=1}^{n_s} f(\boldsymbol{x}_1^{(i)}) f(\boldsymbol{x}_2^{(i)})^\top - I_{d^*} \right\|_F \right\}.$$

The first term of (7) helps the representation to achieve the augmentation invariance while the second term is used to prevent model collapse. It is worth noting that, unlike existing contrastive learning methods (Ye et al., 2019; He et al., 2020; Chen et al., 2020a;b; HaoChen et al., 2021), the loss function of ACT (7) does not need to construct negative pairs for preventing model collapse, avoiding the issues introduced by negative samples.

We now introduce an alternating algorithm for solving the minimax problem (6). We take the $t$-th iteration as an example. Observe that the inner maximization problem is linear. Given the previous representation mapping $f_{(t-1)} : \mathbb{R}^d \to \mathbb{R}^{d^*}$, the explicit solution to the maximization problem is given as

$$\widehat{G}_{(t)} = \frac{1}{n_s} \sum_{i=1}^{n_s} f_{(t-1)}(\boldsymbol{x}_1^{(i)}) f_{(t-1)}(\boldsymbol{x}_2^{(i)})^\top - I_{d^*}. \tag{8}$$

Then it suffices to solve the outer minimization problem

$$\hat{f}_{(t)} \in \arg\min_{f \in \mathcal{F}} \frac{1}{n_s} \sum_{i=1}^{n_s} \left[ \|f(\boldsymbol{x}_1^{(i)}) - f(\boldsymbol{x}_2^{(i)})\|_2^2 + \lambda \langle f(\boldsymbol{x}_1^{(i)}) f(\boldsymbol{x}_2^{(i)})^\top - I_{d^*}, \widehat{G}_{(t)} \rangle_F \right]. \tag{9}$$

Solving the inner problem (8) and the outer problem (9) alternatively yields the desired representation mapping. The detailed algorithm is summarized as Algorithm 1.

---

**Algorithm 1** Adversarial contrastive training (ACT)

---

**Require:** Augmented dataset $D_s = \{\tilde{\boldsymbol{x}}^{(i)}\}_{i \in [n]}$, initial representation $\hat{f}_{(0)}$, iteration horizon $T$.
1: **for** $t \in [T]$ **do**
2:      Update $G$ by solving the inner problem (8).
3:      Update the representation by solving the outer problem (9).
4: **end for**
5: **return** The learned representation mapping $\hat{f}_{(T)}$.

---

*Remark* 2.1. We note that $\widehat{G}_{(t)}$ will be detached from the computational graph when solving the outer problem (9) in practice, which means that the gradient of the second term in (9) should be written as $\langle \nabla_{\boldsymbol{\theta}} \frac{1}{n_s} \sum_{i=1}^{n_s} f_{\boldsymbol{\theta}}(\boldsymbol{x}_1^{(i)}) f_{\boldsymbol{\theta}}(\boldsymbol{x}_2^{(i)})^\top - I_{d^*}, \widehat{G}_{(t)} \rangle$ instead of $\nabla_{\boldsymbol{\theta}} \left\| \frac{1}{n_s} \sum_{i=1}^{n_s} f_{\boldsymbol{\theta}}(\boldsymbol{x}_1^{(i)}) f_{\boldsymbol{\theta}}(\boldsymbol{x}_2^{(i)})^\top - I_{d^*} \right\|_F^2$, which is a biased discretization of $\left\| \mathbb{E}_{\boldsymbol{x}} \mathbb{E}_{\boldsymbol{x}_1, \boldsymbol{x}_2 \in \mathcal{A}(\boldsymbol{x})} [f(\boldsymbol{x}_1) f(\boldsymbol{x}_2)^\top] - I_{d^*} \right\|_F^2$.

## 2.3 DOWNSTREAM TASK

With the help of the representations learned by ACT, we address the downstream classification task in the target domain. Let $\mathcal{X}_t \subseteq [0,1]^d$ represent the target domain, and let $P_t$ be the corresponding unknown distribution. Suppose we have $n_t$ i.i.d. labeled samples $\{(\boldsymbol{z}^{(1)}, y_1), \ldots, (\boldsymbol{z}^{(n_t)}, y_{n_t})\} \subseteq \mathcal{X}_t \times [K]$ for the downstream task. We will say that $\boldsymbol{z} \in C_t(k)$ if its label is $k \in [K]$. By a similar process as in obtaining (5), we can construct the augmented dataset in the target domain as follows.

$$ D_t = \{(\tilde{\boldsymbol{z}}^{(i)}, y_i) : \tilde{\boldsymbol{z}}^{(i)} = (\boldsymbol{z}_1^{(i)}, \boldsymbol{z}_2^{(i)})\}_{i \in [n_t]}, \quad \boldsymbol{z}_1^{(i)} = A(\boldsymbol{z}^{(i)}), \; \boldsymbol{z}_2^{(i)} = A'(\boldsymbol{z}^{(i)}), $$

where $A$ and $A'$ are drawn from the uniform distribution on $\mathcal{A}$ independently.

Given the representation $\hat{f}_{n_s}$ learned by our self-supervised learning method (6), we adopt following linear probe as the classifier for downstream task:

$$ Q_{\hat{f}_{n_s}}(\boldsymbol{z}) = \underset{k \in [K]}{\arg\max} \left(\widehat{W} \hat{f}_{n_s}(\boldsymbol{z})\right)_k, \tag{10} $$

where the $k$-th row of $\widehat{W}$ is given as

$$ \widehat{\mu}_t(k) = \frac{1}{2n_t(k)} \sum_{i=1}^{n_t} (\hat{f}_{n_s}(\boldsymbol{z}_1^{(i)}) + \hat{f}_{n_s}(\boldsymbol{z}_2^{(i)})) \mathbb{1}\{y_i = k\}, \quad n_t(k) := \sum_{i=1}^{n_t} \mathbb{1}\{y_i = k\}. $$

This means that we build a template for each class of downstream task through calculating the average representations of each class. Whenever a new sample needs to be classified, simply classify it into the category of the template that it most closely resembles. The algorithm for downstream task can be summarized as Algorithm 2. Finally, the misclassification rate is defined as

$$ \text{Err}(Q_{\hat{f}_{n_s}}) = \sum_{k=1}^{K} P_t\big(Q_{\hat{f}_{n_s}}(\boldsymbol{z}) \neq k, \boldsymbol{z} \in C_t(k)\big), \tag{11} $$

which are used to evaluate the performance of the representation learned by ACT.

---

**Algorithm 2** Downstream classification

---

**Require:** Representation mapping $\hat{f}_{n_s}$, augmented dataset in the target domain $D_t = \{(\tilde{\boldsymbol{z}}^{(i)}, y_i)\}_{i \in [n_t]}$, testing data $\boldsymbol{z}$.
1: Fit the linear probe according to

$$ \widehat{W}(k, :) = \frac{1}{2n_t(k)} \sum_{i=1}^{n_t} (\hat{f}_{n_s}(\boldsymbol{z}_1^{(i)}) + \hat{f}_{n_s}(\boldsymbol{z}_2^{(i)})) \mathbb{1}\{y_i = k\} $$

2: Predict the label of testing data by (10).
3: **return** The predicted label of testing data $Q_{\hat{f}_{n_s}}(\boldsymbol{z})$.

---

## 3 REAL DATA ANALYSIS

As the experiments conducted in existing self-supervised learning methods, we pretrain the representation on CIFAR-10, CIFAR-100 and Tiny ImageNet, and subsequently conduct fine-tuning on each dataset with annotations. Table 1 shows the classification accuracy of representations learned by ACT, compared with the results reported in Ermolov et al. (2021). We can see that ACT consistently outperforms previous mainstream self-supervised methods across various datasets and evaluation metrics.

The experimental details are deferred to Appendix C. The PyTorch code be found in https://anonymous.4open.science/r/Adversarial-Contrastive-Training.

Table 1: Classification accuracy (top 1) of a linear classifier and a 5-nearest neighbors classifier for different loss functions and datasets. While the results for Barlow Twins are from Bandara et al. (2023), the remains are derived from Ermolov et al. (2021).

| Method | CIFAR-10 | | CIFAR-100 | | Tiny ImageNet | |
|---|---|---|---|---|---|---|
| | linear | 5-NN | linear | 5-NN | linear | 5-NN |
| SimCLR (Ermolov et al. (2021)) | 91.80 | 88.42 | 66.83 | 56.56 | 48.84 | 32.86 |
| BYOL (Ermolov et al. (2021)) | 91.73 | 89.45 | 66.60 | 56.82 | **51.00** | 36.24 |
| W-MSE 2 (Ermolov et al. (2021)) | 91.55 | 89.69 | 66.10 | 56.69 | 48.20 | 34.16 |
| W-MSE 4 (Ermolov et al. (2021)) | 91.99 | 89.87 | 67.64 | 56.45 | 49.20 | 35.44 |
| BarlowTwins (Bandara et al. (2023)) | 87.76 | 86.66 | 61.64 | 55.94 | 41.80 | 33.60 |
| VICReg (our repro.) | 86.76 | 83.70 | 57.13 | 44.63 | 40.04 | 30.46 |
| HaoChen et al. (2022) (our repro.) | 86.53 | 84.20 | 59.68 | 49.26 | 35.80 | 20.36 |
| ACT (our repro.) | **92.11** | **90.01** | **68.24** | **58.35** | 49.72 | **36.40** |

## 4 THEORETICAL ANALYSIS

In this section, we will explore an end-to-end theoretical guarantee for ACT. It is crucial to introduce several assumptions while expounding on their rationale in Section 4.1. The main theorem and its proof sketch are presented in Section 4.2. The formal version of the main theorem and further details of the proof can be found in Appendix B.2.

We first define the population ACT risk minimizer as

$$f^* \in \underset{f:B_1 \leq \|f\|_2 \leq B_2}{\arg\min} \sup_{G \in \mathcal{G}(f)} \mathcal{L}(f, G), \tag{12}$$

where $\mathcal{L}(\cdot, \cdot)$, the unbiased population counterpart of $\widehat{\mathcal{L}}(\cdot, \cdot)$ (7), is defined as

$$\mathcal{L}(f, G) = \mathbb{E}_{\boldsymbol{x}} \mathbb{E}_{\boldsymbol{x}_1, \boldsymbol{x}_2 \in \mathcal{A}(\boldsymbol{x})} \big[ \|f(\boldsymbol{x}_1) - f(\boldsymbol{x}_2)\|_2^2 \big] + \lambda \big\langle \mathbb{E}_{\boldsymbol{x}} \mathbb{E}_{\boldsymbol{x}_1, \boldsymbol{x}_2 \in \mathcal{A}(\boldsymbol{x})} \big[ f(\boldsymbol{x}_1) f(\boldsymbol{x}_2)^\top \big] - I_{d^*}, G \big\rangle_F,$$

and the population feasible set is defined as

$$\mathcal{G}(f) = \big\{ G \in \mathbb{R}^{d^* \times d^*} : \|G\|_F \leq \|\mathbb{E}_{\boldsymbol{x}} \mathbb{E}_{\boldsymbol{x}_1, \boldsymbol{x}_2 \in \mathcal{A}(\boldsymbol{x})} [f(\boldsymbol{x}_1) f(\boldsymbol{x}_2)^\top] - I_{d^*} \|_F \big\}.$$

Here $B_1$ and $B_2$ are two positive constant, and we will detail how to set $B_1$ and $B_2$ later.

### 4.1 ASSUMPTIONS

In this subsection, we will put forward certain assumptions that are necessary to establish our main theorem. We first assume that each component of $f^*$ exhibits a certain regularity and smoothness.

**Definition 4.1** (Hölder class). Let $d \in \mathbb{N}$ and $\alpha = r + \beta > 0$, where $r \in \mathbb{N}_0$ and $\beta \in (0, 1]$. We denote the Hölder class $\mathcal{H}^\alpha(\mathbb{R}^d)$ as

$$\mathcal{H}^\alpha(\mathbb{R}^d) := \Big\{ f : \mathbb{R}^d \to \mathbb{R}, \max_{\|\boldsymbol{s}\|_1 \leq r} \sup_{\boldsymbol{x} \in \mathbb{R}^d} |\partial^{\boldsymbol{s}} f(\boldsymbol{x})| \leq 1, \max_{\|\boldsymbol{s}\|_1 = r} \sup_{\boldsymbol{x} \neq \boldsymbol{y}} \frac{\partial^{\boldsymbol{s}} f(\boldsymbol{x}) - \partial^{\boldsymbol{s}} f(\boldsymbol{y})}{\|\boldsymbol{x} - \boldsymbol{y}\|_\infty^\beta} \leq 1 \Big\},$$

where the multi-index $\boldsymbol{s} \in \mathbb{N}_0^d$. Furthermore, we denote $\mathcal{H}^\alpha := \{ f : [0, 1]^d \to \mathbb{R}, f \in \mathcal{H}^\alpha(\mathbb{R}^d) \}$ as the restriction of $\mathcal{H}^\alpha(\mathbb{R}^d)$ to $[0, 1]^d$.

The Hölder class is known to be a highly comprehensive functional class, providing a precise characterization of the low-order regularity of functions.

**Assumption 4.1.** There exists $\alpha = r + \beta$ with $r \in \mathbb{N}_0$ and $\beta \in (0, 1]$ s.t $f_i^* \in \mathcal{H}^\alpha$ for each $i \in [d^*]$.

Assumption 4.1 is a standard assumption in nonparametric statistics (Tsybakov, 2008; Schmidt-Hieber, 2020), more specifically in studies of neural network approximation capacity (Yarotsky,

2018; Yarotsky & Zhevnerchuk, 2020). It is a pretty mild requirement due to the universality of Hölder class.

Next we enumerate the assumptions about the data augmentations $\mathcal{A}$.

**Assumption 4.2** (Lipschitz augmentation). Any data augmentation $A_\gamma \in \mathcal{A}$ is $M$-Lipschitz, i.e., $\|A_\gamma(\boldsymbol{u}_1) - A_\gamma(\boldsymbol{u}_2)\|_2 \leq M\|\boldsymbol{u}_1 - \boldsymbol{u}_2\|_2$ for any $\boldsymbol{u}_1, \boldsymbol{u}_2 \in [0,1]^d$.

A typical example to understand Assumption 4.2 is that the resulting augmented data obtained through cropping would not undergo drastic changes when minor perturbations are applied to the original image.

Denote the corresponding latent classes on source domain by $\{C_s(k)\}_{k \in [K]}$. Beyond the general assumption regarding data augmentation $\mathcal{A}$ above, we require a more precise way to describe the intensity of data augmentations $\mathcal{A}$. A more general version of the $(\sigma, \delta)$-*augmentation* employed by Huang et al. (2023) is adopted by us to distinguish the efficiency of data augmentations.

**Definition 4.2** $((\sigma_s, \sigma_t, \delta_s, \delta_t)$-Augmentation). The augmentations in $\mathcal{A}$ is $(\sigma_s, \sigma_t, \delta_s, \delta_t)$-augmentations, that is, for each $k \in [K]$, there exists a subset $\widetilde{C}_s(k) \subseteq C_s(k)$ and $\widetilde{C}_t(k) \subseteq C_t(k)$, such that

$$P_s\big(\boldsymbol{x} \in \widetilde{C}_s(k)\big) \geq \sigma_s P_s\big(\boldsymbol{x} \in C_s(k)\big), \qquad \sup_{\boldsymbol{x}_1, \boldsymbol{x}_2 \in \widetilde{C}_s(k)} \min_{\boldsymbol{x}_1' \in \mathcal{A}(\boldsymbol{x}_1), \boldsymbol{x}_2' \in \mathcal{A}(\boldsymbol{x}_2)} \|\boldsymbol{x}_1' - \boldsymbol{x}_2'\|_2 \leq \delta_s,$$

$$P_t\big(\boldsymbol{z} \in \widetilde{C}_t(k)\big) \geq \sigma_t P_t\big(\boldsymbol{z} \in C_t(k)\big), \qquad \sup_{\boldsymbol{z}_1, \boldsymbol{z}_2 \in \widetilde{C}_t(k)} \min_{\boldsymbol{z}_1' \in \mathcal{A}(\boldsymbol{z}_1), \boldsymbol{z}_2' \in \mathcal{A}(\boldsymbol{z}_2)} \|\boldsymbol{z}_1' - \boldsymbol{z}_2'\|_2 \leq \delta_t,$$

$$P_t\big(\bigcup_{k=1}^{K} \widetilde{C}_t(k)\big) \geq \sigma_t,$$

where $\sigma_s, \sigma_t \in (0,1]$ and $\delta_s, \delta_t \geq 0$.

*Remark* 4.1. The $(\sigma_s, \sigma_t, \delta_s, \delta_t)$-augmentation methods emphasize that a robust data augmentation should adhere to the principle that when the semantic information of the original images exhibit heightened similarity, augmented views from them should be close according to specific criteria. Among above requirements, $P_t\big(\bigcup_{k=1}^{K} \widetilde{C}_t(k)\big) \geq \sigma_t$, which is used to replace the assumption $\mathcal{A}(C_t(i)) \cap \mathcal{A}(C_t(j)) = \emptyset$ of Huang et al. (2023), implies that the augmentation used should be intelligent enough to recognize objectives aligned with the image labels for the majority of samples in the dataset. For instance, consider a downstream task involving classifying images of cats and dogs, where the dataset includes some images featuring both cats and dogs together. This requirement demands that the data augmentation intelligently selects dog-specific augmentations when the image is labeled as dog, and similarly for cat-specific augmentations when the image is labeled as cat. A simple alternative to this requirement is assuming different class $C_t(k)$ are pairwise disjoint, i.e., $\forall i \neq j, C_t(i) \cap C_t(j) = \emptyset$, which implies $P_t\big(\bigcup_{k=1}^{K} \widetilde{C}_t(k)\big) = \sum_{k=1}^{K} P_t(\widetilde{C}_t(k)) \geq \sigma_t \sum_{k=1}^{K} P_t(C_t(k)) = \sigma_t$.

**Assumption 4.3** (Existence of augmentation sequence). Assume there exists a sequence of $(\sigma_s^{(n_s)}, \sigma_t^{(n_s)}, \delta_s^{(n_s)}, \delta_t^{(n_s)})$-data augmentations $\mathcal{A}_{n_s} = \{A_\gamma^{(n_s)}(\cdot) : \gamma \in [m]\}$ and $\tau > 0$ such that

$$\max\{\delta_s^{(n_s)}, \delta_t^{(n_s)}\} \leq n_s^{-\frac{\tau+d+1}{2(\alpha+d+1)}}, \quad \min\{\sigma_s^{(n_s)}, \sigma_t^{(n_s)}\} \overset{n_s \to \infty}{\to} 1$$

It is worth mentioning that this assumption essentially aligns with Assumption 3.5 in HaoChen et al. (2021), both stipulating the augmentations must be sufficiently robust so that the internal connections within latent classes are strong enough to prevent instance clusters from being separated. Recently, methods for building stronger data augmentation, as discussed by Jahanian et al. (2022) and Trabucco et al. (2024), are constantly being proposed, making it more feasible to meet the theoretical requirements for data augmentation.

Next we are going to introduce the assumption about distribution shift. For simplicity, denote $p_s(k) = P_s(\boldsymbol{x} \in C_s(k))$ and $P_s(k)$ be the conditional distribution of $P_s(\boldsymbol{x}|\boldsymbol{x} \in C_s(k))$ on the upstream data, $p_t(k) = P_t(\boldsymbol{z} \in C_t(k))$ and $P_t(k)$ be the conditional distribution $P_t(\boldsymbol{z}|\boldsymbol{z} \in C_t(k))$ on the downstream task. Following assumption is needed to quantify our requirement on domain shift.

**Assumption 4.4.** Assume there exists $\nu > 0$ and $\varsigma > 0$ such that

$$\max_{k \in [K]} \mathcal{W}(P_s(k), P_t(k)) \leq n_s^{-\frac{\nu+d+1}{2(\alpha+d+1)}}, \quad \max_{k \in [K]} |p_s(k) - p_t(k)| \leq n_s^{-\frac{\varsigma}{2(\alpha+d+1)}},$$

where $\mathcal{W}$ is the Wasserstein-1 distance.

A trivial scenario occurs when there is no gap between the upstream and downstream distributions, i.e., when $(\mathcal{X}_s, P_s) = (\mathcal{X}_t, P_t)$, leading to both $\max_{k \in [K]} \mathcal{W}(P_s(k), P_t(k))$ and $\max_{k \in [K]} |p_s(k) - p_t(k)|$ vanishing.

**Assumption 4.5.** Assume there exists a measurable partition $\{\mathcal{P}_1, \ldots, \mathcal{P}_{d^*}\}$ of $\mathcal{X}_s$, such that $1/B_2^2 \leq P_s(\mathcal{P}_i) \leq 1/B_1^2$ for each $i \in [d^*]$.

Assumption 4.5 is used to construct a measurable function $\tilde{f}$ with $B_1 \leq \|\tilde{f}\|_2 \leq B_2$, such that $\mathcal{L}(\tilde{f}) = 0$, tackling one of theoretical challenges introduced in Theorem 4.2 of HaoChen & Ma (2023), further implying that $\mathcal{L}(f^*)$ vanishes (see B.2.6 for more details). It suggests that the data distribution in the source domain should not be overly singular. All common continuous distributions defined on Borel algebra apparently satisfy these requirements, as the measure of any single point is zero.

## 4.2 END-TO-END THEORETICAL GUARANTEE

Our main theoretical result is stated as follows.

**Theorem 4.2.** *Suppose Assumptions 4.1-4.5 hold. Set the width, depth and the Lipschitz constraint of the deep neural network as*

$$W \geq \mathcal{O}\big(n_s^{\frac{2d+\alpha}{4(\alpha+d+1)}}\big), \quad L \geq \mathcal{O}(1), \quad \mathcal{K} = \mathcal{O}\big(n_s^{\frac{d+1}{2(\alpha+d+1)}}\big).$$

*Then the following inequality holds*

$$\mathbb{E}_{D_s}\big[\mathrm{Err}(Q_{\hat{f}_{n_s}})\big] \leq (1 - \sigma_t^{(n_s)}) + \mathcal{O}\big(n_s^{-\frac{\min\{\alpha, \nu, \varsigma\}}{8(\alpha+d+1)}}\big),$$

*with probability at least $\sigma_s^{(n_s)} - \mathcal{O}\big(n_s^{-\frac{\min\{\alpha, \nu, \varsigma, \tau\}}{16(\alpha+d+1)}}\big) - \mathcal{O}\big(\frac{1}{\sqrt{\min_k n_t(k)}}\big)$ for $n_s$ sufficiently large.*

*Remark* 4.3. Note that only the probability term depends on the downstream sample size and the failure probability decays rapidly with respect to $\min_k n_t(k)$ with order $1/2$, implying that the learned representation via ACT from a large amount of unlabeled data can indeed help capture downstream knowledge, despite a limited downstream sample size. This demonstrates the proven advantage of ACT and provides an explanation for the empirical success of few-shot learning, which aligns with the concept of $K$-way $\min_k n_t(k)$-shot learning. Apart from that, note the conditions of Theorem 4.2 only require $W \geq \mathcal{O}(n_s^{\frac{2d+\alpha}{4(\alpha+d+1)}}), L \geq \mathcal{O}(1)$ and $\mathcal{K} = \mathcal{O}(n_s^{\frac{d+1}{2(\alpha+d+1)}})$, which implies that the number of network parameters could be arbitrarily large if we control the norm of weight properly, which is coincide with the concept of over-parametrization.

## 4.3 PROOF SKETCH OF THEOREM 4.2

**Step 1.** In Appendix B.2.1, we initially investigate the sufficient condition for achieving a low error rate in a downstream task at the population level in Lemma B.1. It reveals that the misclassification rate bounded by the strength of data augmentations $1 - \sigma_s$, and the augmented concentration, represented by $R_t(\varepsilon, f)$. This dependence arises when the divergence between different classes, quantified by $\mu_t(i)^\top \mu_t(j)$, is sufficiently dispersed.

**Step 2.** Subsequently in Appendix B.2.2 and B.2.3, we regard $\sup_{G \in \mathcal{G}(f)} \mathcal{L}(f, G)$ as the weighted summation of $\mathcal{L}_{\mathrm{align}}(f)$ and $\mathcal{L}_{\mathrm{div}}(f)$, then attempt to show they are the upper bound of $R_t(\varepsilon, f), \max_{i \neq j} |\mu_t(i)^\top \mu_t(j)|$ respectively in Lemma B.4, which implies that optimizing our adversarial self-supervised learning loss is equivalent to optimize the upper bound of $R_t(\varepsilon, f)$ and $\max_{i \neq j} |\mu_t(i)^\top \mu_t(j)|$ simultaneously, because $\mathcal{L}_{\mathrm{align}}(f)$ and $\mathcal{L}_{\mathrm{div}}(f)$ are positive. Finally, apply

Lemma B.1 and Lemma B.4 to $\hat{f}_{n_s}$, combining with the Markov inequality, to conclude Theorem B.1, which is population version of Theorem 4.2.

**Step 3.** To further obtain an end-to-end theoretical guarantee, we subsequently decompose $\mathcal{E}(\hat{f}_{n_s})$, the excess risk defined in Definition B.3, into three parts: statistical error: $\mathcal{E}_{\text{sta}}$, approximation error introduced by neural network class: $\mathcal{E}_{\mathcal{F}}$, and the error brought by $\widehat{\mathcal{G}}$: $\mathcal{E}_{\widehat{\mathcal{G}}}$ in Appendix B.2.7. Note that the unbiased design of ACT plays a key role in such misspecified decomposition. We successively deal each produced term. For $\mathbb{E}_{D_s}[\mathcal{E}_{\text{sta}}]$, we claim it can be bounded by $\frac{\mathcal{K}\sqrt{L}}{\sqrt{n_s}}$ by adopting some typical techniques of empirical process and the result claimed by Golowich et al. (2018) in Appendix B.2.8. For $\mathcal{E}_{\mathcal{F}}$, according to the existing conclusion of Jiao et al. (2023), we can show $\mathcal{E}_{\mathcal{F}}$ can be bounded by $\mathcal{K}^{-\alpha/(d+1)}$ in Appendix B.2.9. By leveraging the unbiased property of ACT, the problem bounding $\mathbb{E}_{D_s}[\mathcal{E}_{\widehat{\mathcal{G}}}]$ can be transformed into a common problem of mean convergence rate, so that it can be controlled by $\frac{1}{n_s^{1/4}}$ with high probability, shown as Appendix B.2.10. Trading off over three errors helps us determine a appropriate $\mathcal{K}$ to bound $\mathbb{E}_{D_s}[\mathcal{E}(\hat{f}_{n_s})]$, more details is showed in Appendix B.2.11.

**Step 4.** However, $\mathcal{L}(f^*)$, the difference between the excess risk and the term $\mathcal{L}(\hat{f}_{n_s})$ involving in Theorem B.1, still impedes us from building an end-to-end theoretical guarantee for ACT. To address this issue, in Appendix B.2.6, we construct a representation making this term vanishing under Assumption 4.5. Finally, just set appropriate parameters of Theorem B.1 to conclude Lemma B.12, whose direct corollary is Theorem 4.2, and proof is presented in Appendix B.12. The bridge between Lemma B.12 and Theorem 4.2 is shown in Appendix B.2.12.

## 5 CONCLUSIONS

In this paper, we propose a novel adversarial contrastive learning method for unsupervised transfer learning. Our experimental results achieved state-of-the-art classification accuracy under both fine-tuned linear probe and $K$-NN protocol on various real datasets, comparing with the self-supervised learning methods. Meanwhile, we present end to end theoretical guarantee for the downstream classification task under misspecified and over-parameterized setting. Our theoretical results not only indicate that the misclassification rate of downstream task solely depends on the strength of data augmentation on the large amount of unlabeled data, but also bridge the gap in the theoretical understanding of the effectiveness of few-shot learning for downstream tasks with small sample size.

Minimax rates for supervised transfer learning are established in Cai & Wei (2019); Kpotufe & Martinet (2021); Cai & Pu (2024). However, the minimax rate for unsupervised transfer learning remains unclear. Establishing a lower bound to gain a deeper understanding of our ACT model presents an interesting and challenging problem for future research.

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

## A EXPLANATION OF THE REGULARIZATION TERM

In brief, contrastive learning utilizes data augmentation to construct the loss function (specifically, the first term in our loss) that aligns representations of the same class. However, to avoid trivial solutions, an additional regularization term is necessary to ensure that clusters representing different classes are well-separated. We measure this separation using the angles between the centroids of different classes. While these angles are ideal for quantifying separation, they cannot be directly optimized because the latent class annotations are unavailable in the upstream task. As an alternative, we propose finding an appropriate computable loss function that serves as an upper bound for these angles, effectively achieving the desired separation. Denote

$$\mathcal{L}_{\text{div}}(f) = \left\| \mathbb{E}_{\boldsymbol{x}} \mathbb{E}_{\boldsymbol{x}_1, \boldsymbol{x}_2 \in \mathcal{A}(\boldsymbol{x})}[f(\boldsymbol{x}_1)f(\boldsymbol{x}_2)^{\top}] - I_{d^*} \right\|_F^2.$$

It can severs as a regularization term since in Lemma B.4, we can show

$$\mu_s(i)^{\top}\mu_s(j) \lesssim \left\| \mathbb{E}_{\boldsymbol{x}} \mathbb{E}_{\boldsymbol{x}_1, \boldsymbol{x}_2 \in \mathcal{A}(\boldsymbol{x})}[f(\boldsymbol{x}_1)f(\boldsymbol{x}_2)^{\top}] - I_{d^*} \right\|_F, \tag{13}$$

where $\mu_s(i) = \mathbb{E}_{\boldsymbol{x} \in C_s(i)}\mathbb{E}_{\boldsymbol{x}' \in \mathcal{A}(\boldsymbol{x})}[f(\boldsymbol{x}')]$ is the center of the latent class $i$. (13) implies that a lower value of the regularization term leads the separation between different categories' center, thereby benefits classification in downstream tasks.

At the sample level, one can use $\widehat{\mathcal{L}}_{\text{div}}(f) = \left\| \frac{1}{n_s} \sum_{i=1}^{n_s} f(\boldsymbol{x}_1^{(i)}) f(\boldsymbol{x}_2^{(i)})^\top - I_{d^*} \right\|_F^2$ to estimate $\mathcal{L}_{\text{div}}(f)$. However, this lead to a bias loss, i.e.,

$$\mathbb{E}_{D_s}[\widehat{\mathcal{L}}_{\text{div}}(f)] \neq \mathcal{L}_{\text{div}}(f),$$

where $D_s$ is augmented dataset. This bias is caused by the non-commutativity of the expectation and the Frobenius norm. To overcome this we can reformulate it as an equivalent form

$$\mathcal{L}_{\text{div}}(f) = \sup_{G \in \mathcal{G}(f)} \mathcal{R}(f, G) := \langle \mathbb{E}_{\boldsymbol{x}} \mathbb{E}_{\boldsymbol{x}_1, \boldsymbol{x}_2 \in \mathcal{A}(\boldsymbol{x})}[f(\boldsymbol{x}_1) f(\boldsymbol{x}_2)^\top] - I_{d^*}, G \rangle_F.$$

The counterpart of $\mathcal{R}(f, G)$ at the sample level is

$$\widehat{\mathcal{R}}(f, G) = \langle \frac{1}{n_s} \sum_{i=1}^{n_s} f(\boldsymbol{x}_1^{(i)}) f(\boldsymbol{x}_2^{(i)})^\top - I_{d^*}, G \rangle_F.$$

We can see that $\mathbb{E}_{D_s}[\widehat{\mathcal{R}}(f, G)] = \mathcal{R}(f, G)$ for any fixed $G$ due to the linearity of Frobenius inner product, combining this property with the new decomposition method proposed by us, we build an end-to-end theoretical guarantee in the transfer learning setting to provide an explanation for few shot learning. And using an alternative optimization method to optimize this loss is natural.

## B  DEFERRED PROOF

The Section B will be divided into two parts. The first part B.1 is used to prove $\|\phi_{\boldsymbol{\theta}}\|_{\text{Lip}} \leq \mathcal{K}$ for any $\phi_{\boldsymbol{\theta}} \in \mathcal{NN}_{d_1, d_2}(W, L, \mathcal{K}, B_1, B_2)$. The proof of Theorem 4.2 is shown in the second part B.2.

### B.1  $\mathcal{K}$-LIPSCHITZ PROPERTY OF $\mathcal{NN}_{d_1, d_2}(W, L, \mathcal{K}, B_1, B_2)$

*Proof.* To claim any $\phi_{\boldsymbol{\theta}} \in \mathcal{NN}_{d_1, d_2}(W, L, \mathcal{K}, B_1, B_2)$ is $\mathcal{K}$-Lipschitz function, we need to define two special classes of neural network functions, the first is

$$\mathcal{NN}_{d_1, d_2}(W, L, \mathcal{K}) := \{\phi_{\boldsymbol{\theta}}(\boldsymbol{x}) = \boldsymbol{A}_L \sigma(\boldsymbol{A}_{L-1} \sigma(\cdots \sigma(\boldsymbol{A}_0 \boldsymbol{x}))) : \kappa(\boldsymbol{\theta}) \leq \mathcal{K}\}, \quad (14)$$

which equivalent to $\mathcal{NN}_{d_1, d_2}(W, L, \mathcal{K}, B_1, B_2)$ ignoring the condition $\|\phi_{\boldsymbol{\theta}}\|_2 \in [B_1, B_2]$, and the second one

$$\mathcal{SNN}_{d_1, d_2}(W, L, \mathcal{K}) := \{\breve{\phi}(\boldsymbol{x}) = \breve{\boldsymbol{A}}_L \sigma(\breve{\boldsymbol{A}}_{L-1} \sigma(\cdots \sigma(\breve{\boldsymbol{A}}_0 \breve{\boldsymbol{x}}))) : \prod_{l=1}^{L} \|\breve{\boldsymbol{A}}_l\|_\infty \leq \mathcal{K}\}, \qquad \breve{\boldsymbol{x}} := \begin{pmatrix} \boldsymbol{x} \\ 1 \end{pmatrix},$$

where $\breve{\boldsymbol{A}}_l \in \mathbb{R}^{N_{l+1} \times N_l}$ with $N_0 = d_1 + 1$.

It is obvious that $\mathcal{NN}_{d_1, d_2}(W, L, \mathcal{K}, B_1, B_2) \subseteq \mathcal{NN}_{d_1, d_2}(W, L, \mathcal{K})$ and every element in $\mathcal{SNN}_{d_1, d_2}(W, L, \mathcal{K})$ is $\mathcal{K}$-Lipschitz function as the 1-Lipschitz property of ReLU, thus it suffices to show that $\mathcal{SNN}_{d_1, d_2}(W, L, \mathcal{K}) \subseteq \mathcal{NN}_{d_1, d_2}(W, L, \mathcal{K}) \subseteq \mathcal{SNN}_{d_1, d_2}(W+1, L, \mathcal{K})$ to yield what we desired.

In fact, any $\phi_{\boldsymbol{\theta}}(\boldsymbol{x}) = \boldsymbol{A}_L \sigma(\boldsymbol{A}_{L-1} \sigma(\cdots \sigma(\boldsymbol{A}_0 \boldsymbol{x} + \boldsymbol{b}_0)) + \boldsymbol{b}_{L-1}) \in \mathcal{NN}_{d_1, d_2}(W, L, \mathcal{K})$ can be rewritten as $\breve{\phi}(\boldsymbol{x}) = \breve{\boldsymbol{A}}_L \sigma(\breve{\boldsymbol{A}}_{L-1} \sigma(\cdots \sigma(\breve{\boldsymbol{A}}_0 \breve{\boldsymbol{x}})))$, where

$$\breve{\boldsymbol{x}} := \begin{pmatrix} \boldsymbol{x} \\ 1 \end{pmatrix}, \breve{\boldsymbol{A}} = (\boldsymbol{A}_L, \boldsymbol{0}), \breve{\boldsymbol{A}}_l = \begin{pmatrix} \boldsymbol{A}_l & \boldsymbol{b}_l \\ \boldsymbol{0} & 1 \end{pmatrix}, l = 0, \ldots, L-1.$$

Notice that $\prod_{l=0}^{L} \|\breve{\boldsymbol{A}}\|_\infty = \|\boldsymbol{A}_L\|_\infty \prod_{l=0}^{L-1} \max\{\|(\boldsymbol{A}_l, \boldsymbol{b}_l)\|_\infty, 1\} = \kappa(\boldsymbol{\theta}) \leq \mathcal{K}$, which implies that $\phi_{\boldsymbol{\theta}} \in \mathcal{SNN}_{d_1, d_2}(W+1, L, \mathcal{K})$.

Conversely, since any $\breve{\phi} \in \mathcal{SNN}(W, L, \mathcal{K})$ can also be parameterized in the form of $\boldsymbol{A}_L \sigma(\boldsymbol{A}_{L-1} \sigma(\cdots \sigma(\boldsymbol{A}_0 \boldsymbol{x} + \boldsymbol{b}_0)) + \boldsymbol{b}_{L-1})$ with $\boldsymbol{\theta} = (\breve{\boldsymbol{A}}_0, (\breve{\boldsymbol{A}}_1, \boldsymbol{0}), \ldots, (\breve{\boldsymbol{A}}_{L-1}, \boldsymbol{0}), \breve{\boldsymbol{A}}_L)$, and by the absolute homogeneity of the ReLU function, we can always rescale $\breve{\boldsymbol{A}}_l$ such that $\|\breve{\boldsymbol{A}}_L\|_\infty \leq \mathcal{K}$ and $\|\breve{\boldsymbol{A}}_l\|_\infty = 1$ for $l \neq L$. Hence $\kappa(\boldsymbol{\theta}) = \prod_{l=0}^{L} \|\breve{\boldsymbol{A}}_l\|_\infty \leq \mathcal{K}$, which yields $\breve{\phi} \in \mathcal{NN}(W, L, \mathcal{K})$. $\qquad \square$

### B.2 PROOF OF THEOREM 4.2

We will begin by exploring the sufficient condition for achieving small $\mathrm{Err}(Q_f)$ in B.2.1. Following that, we build the connection between the required condition and optimizing our adversarial self-supervised learning loss in Theorem B.1 of B.2.3, it reveals that small quantity of our loss function may induce significant class divergence and high augmented concentration. Although this theorem can explain the essential factors behind the success of our method to some extent, its analysis still stay at population level, the impact of sample size on $\mathrm{Err}(Q_f)$ remains unresolved. To obtain an end-to-end theoretical guarantee as Theorem 4.2, we first decompose $\mathcal{E}(\hat{f}_{n_s})$, which is the excess risk defined in the Definition B.3, into three parts: statistical error: $\mathcal{E}_{\mathrm{sta}}$, approximation error brought by $\mathcal{F}$: $\mathcal{E}_{\mathcal{F}}$ and the error introduced by using $\widehat{\mathcal{G}}(f)$ to approximate $\mathcal{G}(f)$: $\mathcal{E}_{\widehat{\mathcal{G}}}$ in B.2.7, then successively deal each produced term. For $\mathbb{E}_{D_s}[\mathcal{E}_{\mathrm{sta}}]$, we adopt some typical techniques of empirical process and the result provided by Golowich et al. (2018) in B.2.8 for bounding it by $\frac{\mathcal{K}\sqrt{L}}{\sqrt{n_s}}$. Regarding bounding $\mathcal{E}_{\mathcal{F}}$, we first convert the problem to a function approximation problem and adopt the existing conclusion proposed by Jiao et al. (2023), yielding $\mathcal{E}_{\mathcal{F}}$ can be bounded by $\mathcal{K}^{-\alpha/(d+1)}$ in B.2.9. By leveraging the property $\mathbb{E}_{D_s}[\widehat{\mathcal{L}}(f,G)] = \mathcal{L}(f,G)$, we find that the problem of bounding $\mathbb{E}_{D_s}[\mathcal{E}_{\widehat{\mathcal{G}}}]$ can be transformed into a common problem of mean convergence rate and further control it by $\frac{1}{n_s^{1/4}}$ in B.2.10. After finishing these preliminaries, trade off between these errors to determine a appropriate Lipschitz constant $\mathcal{K}$ of neural network, while bound the expectation of excess risk $\mathbb{E}_{D_s}[\mathcal{E}(\hat{f}_{n_s})]$, more details are deferred to B.2.11. However, $\sup_{G\in\mathcal{G}(f^*)}\mathcal{L}(f^*,G)$, the difference between the excess risk and the term $\sup_{G\in\mathcal{G}(\hat{f}_{n_s})}\mathcal{L}(\hat{f}_{n_s},G)$ involving in Theorem B.1, still impedes us from building an end-to-end theoretical guarantee for ACT. To address this issue, in B.2.6, we construct a representation making this term vanishing under Assumption 4.5. Finally, just set appropriate parameters of Theorem B.1 to conclude Lemma B.12, and the bridge between Lemma B.12 and Theorem 4.2 is built in B.2.12.

### B.2.1 SUFFICIENT CONDITION OF SMALL MISCLASSIFICATION RATE

**Lemma B.1.** *Given a $(\sigma_s, \sigma_t, \delta_s, \delta_t)$-augmentation, if the encoder $f$ such that $B_1 \leq \|f\|_2 \leq B_2$ is $\mathcal{K}$-Lipschitz and*

$$\mu_t(i)^\top \mu_t(j) < B_2^2 \Theta(\sigma_t, \delta_t, \varepsilon, f),$$

*holds for any pair of $(i,j)$ with $i \neq j$, then the downstream error rate of $Q_f$*

$$\mathrm{Err}(Q_f) \leq (1 - \sigma_t) + R_t(\varepsilon, f),$$

*where $\varepsilon > 0$, $\mu_t(k) = \mathbb{E}_{z\in C_t(k)}\mathbb{E}_{z'\in\mathcal{A}(z)}[f(z')]$ for any $k \in [K]$, $\Gamma_{\min}(\sigma_t, \delta_t, \varepsilon, f) = \Big(\sigma_t - \frac{R_t(\varepsilon, f)}{\min_i p_t(i)}\Big)\Big(1 + \big(\frac{B_1}{B_2}\big)^2 - \frac{\mathcal{K}\delta_t}{B_2} - \frac{2\varepsilon}{B_2}\Big) - 1$, $\Delta_{\hat{\mu}_t} = 1 - \frac{\min_{k\in[K]}\|\hat{\mu}_t(k)\|_2^2}{B_2^2}$, $R_t(\varepsilon, f) = P_t\big(z \in \cup_{k=1}^K C_t(k) : \sup_{z_1, z_2\in\mathcal{A}(z)}\|f(z_1) - f(z_2)\|_2 > \varepsilon\big)$ and $\Theta(\sigma_t, \delta_t, \varepsilon, f) = \Gamma_{\min}(\sigma_t, \delta_t, \varepsilon, f) - \sqrt{2 - 2\Gamma_{\min}(\sigma_t, \delta_t, \varepsilon, f)} - \frac{\Delta_{\hat{\mu}_t}}{2} - \frac{2\max_{k\in[K]}\|\hat{\mu}_t(k) - \mu_t(k)\|_2}{B_2}$.*

*Proof.* For any encoder $f$, let $S_t(\varepsilon, f) := \{z \in \cup_{k=1}^K C_t(k) : \sup_{z_1, z_2\in\mathcal{A}(z)}\|f(z_1) - f(z_2)\|_2 \leq \varepsilon\}$, if any $z \in (\widetilde{C}_t(1) \cup \cdots \cup \widetilde{C}_t(K)) \cap S_t(\varepsilon, f)$ can be correctly classified by $Q_f$, it turns out that $\mathrm{Err}(Q_f)$ can be bounded by $(1 - \sigma_t) + R_t(\varepsilon, f)$. In fact,

$$\begin{aligned}
\mathrm{Err}(Q_f) &= \sum_{k=1}^K P_t\big(Q_f(z) \neq k, \forall z \in C_t(k)\big) \\
&\leq P_t\Big(\big((\widetilde{C}_t(1) \cup \cdots \cup \widetilde{C}_t(K)) \cap S_t(\varepsilon, f)\big)^c\Big) \\
&= P_t\big((\widetilde{C}_t(1) \cup \cdots \cup \widetilde{C}_t(K))^c \cup (S_t(\varepsilon, f))^c\big) \\
&\leq (1 - \sigma_t) + P_t\big((S_t(\varepsilon, f))^c\big) \\
&= (1 - \sigma_t) + R_t(\varepsilon, f).
\end{aligned}$$

The first row is derived according to the definition of $\text{Err}(Q_f)$. Since any $z \in (\widetilde{C}_t(1) \cup \cdots \cup \widetilde{C}_t(K)) \cap S_t(\varepsilon, f)$ can be correctly classified by $Q_f$, we yields the second row. De Morgan's laws implies the third row. The fourth row stems from the Definition 4.2. Finally, just note $R_t(\varepsilon, f) = (S_t(\varepsilon, f))^c$ to obtain the last line.

Hence it suffices to show for given $i \in [K]$, $z \in \widetilde{C}_t(i) \cap S_t(\varepsilon, f)$ can be correctly classified by $Q_f$ if for any $j \neq i$,

$$
\mu_t(i)^\top \mu_t(j) < B_2^2 \Big( \Gamma_i(\sigma_t, \delta_t, \varepsilon, f) - \sqrt{2 - 2\Gamma_i(\sigma_t, \delta_t, \varepsilon, f)} - \frac{\Delta_{\hat{\mu}_t}}{2} - \frac{\|\hat{\mu}_t(i) - \mu_t(i)\|_2}{B_2}
$$
$$
- \frac{\|\hat{\mu}_t(j) - \mu_t(j)\|_2}{B_2} \Big),
$$

where $\Gamma_i(\sigma_t, \delta_t, \varepsilon, f) = \left( \sigma_t - \frac{R_t(\varepsilon, f)}{p_t(i)} \right) \left( 1 + \left( \frac{B_1}{B_2} \right)^2 - \frac{\mathcal{K}\delta_t}{B_2} - \frac{2\varepsilon}{B_2} \right) - 1$.

To this end, without losing generality, consider the case $i = 1$. To turn out $z_0 \in \widetilde{C}_t(1) \cap S_t(\varepsilon, f)$ can be correctly classified by $Q_f$, by the definition of $\widetilde{C}_t(1)$ and $S_t(\varepsilon, f)$, It just need to show $\forall k \neq 1, \|f(z_0) - \hat{\mu}_t(1)\|_2 < \|f(z_0) - \hat{\mu}_t(k)\|_2$, which is equivalent to

$$
f(z_0)^\top \hat{\mu}_t(1) - f(z_0)^\top \hat{\mu}_t(k) - \left( \frac{1}{2}\|\hat{\mu}_t(1)\|_2^2 - \frac{1}{2}\|\hat{\mu}_t(k)\|_2^2 \right) > 0.
$$

We will firstly deal with the term $f(z_0)^\top \hat{\mu}_t(1)$,

$$
f(z_0)^\top \hat{\mu}_t(1) = f(z_0)^\top \mu_t(1) + f(z_0)^\top (\hat{\mu}_t(1) - \mu_t(1))
$$
$$
\geq f(z_0)^\top \mathbb{E}_{z \in C_t(1)} \mathbb{E}_{z' \in \mathcal{A}(z)}[f(z')] - \|f(z_0)\|_2 \|\hat{\mu}_t(1) - \mu_t(1)\|_2
$$
$$
\geq \frac{1}{p_t(1)} f(z_0)^\top \mathbb{E}_z \mathbb{E}_{z' \in \mathcal{A}(z)}[f(z')\mathbb{1}\{z \in C_t(1)\}] - B_2\|\hat{\mu}_t(1) - \mu_t(1)\|_2
$$
$$
= \frac{1}{p_t(1)} f(z_0)^\top \mathbb{E}_z \mathbb{E}_{z' \in \mathcal{A}(z)}\big[f(z')\mathbb{1}\{z \in C_t(1) \cap \widetilde{C}_t(1) \cap S_t(\varepsilon, f)\}\big]
$$
$$
+ \frac{1}{p_t(1)} f(z_0)^\top \mathbb{E}_z \mathbb{E}_{z' \in \mathcal{A}(z)}\big[f(z')\mathbb{1}\{z \in C_t(1) \cap (\widetilde{C}_t(1) \cap S_t(\varepsilon, f))^c\}\big]
$$
$$
- B_2\|\hat{\mu}_t(1) - \mu_t(1)\|_2
$$
$$
= \frac{P_t(\widetilde{C}_t(1) \cap S_t(\varepsilon, f))}{p_t(1)} f(z_0)^\top \mathbb{E}_{z \in \widetilde{C}_t(1) \cap S_t(\varepsilon, f)} \mathbb{E}_{z' \in \mathcal{A}(z)}[f(z')]
$$
$$
+ \frac{1}{p_t(1)} \mathbb{E}_z \big[\mathbb{E}_{z' \in \mathcal{A}(z)}[f(z_0)^\top f(z')]\mathbb{1}\{z \in C_t(1) \backslash (\widetilde{C}_t(1) \cap S_t(\varepsilon, f))\}\big]
$$
$$
- B_2\|\hat{\mu}_t(1) - \mu_t(1)\|_2
$$
$$
\geq \frac{P_t(\widetilde{C}_t(1) \cap S_t(\varepsilon, f))}{p_t(1)} f(z_0)^\top \mathbb{E}_{z \in \widetilde{C}_t(1) \cap S_t(\varepsilon, f)} \mathbb{E}_{z' \in \mathcal{A}(z)}[f(z')]
$$
$$
- \frac{B_2^2}{p_t(1)} P_t\big(C_t(1) \backslash (\widetilde{C}_t(1) \cap S_t(\varepsilon, f))\big) - B_2\|\hat{\mu}_t(1) - \mu_t(1)\|_2, \tag{15}
$$

where the second row stems from Cauchy–Schwarz inequality. The third and the last rows are according to the condition $\|f\|_2 \leq B_2$.

Note that

$$
P_t\big(C_t(1) \backslash (\widetilde{C}_t(1) \cap S_t(\varepsilon, f))\big) = P_t\big((C_t(1) \backslash \widetilde{C}_t(1)) \cup (\widetilde{C}_t(1) \cap (S_t(\varepsilon, f))^c)\big)
$$
$$
\leq (1 - \sigma_t)p_t(1) + R_t(\varepsilon, f), \tag{16}
$$

and

$$
P_t\big(\widetilde{C}_t(1) \cap S_t(\varepsilon, f)\big) = P_t(C_t(1)) - P_t\big(C_t(1) \backslash (\widetilde{C}_t(1) \cap S_t(\varepsilon, f))\big)
$$
$$
\geq p_t(1) - ((1 - \sigma_t)p_t(1) + R_t(\varepsilon, f))
$$
$$
= \sigma_t p_t(1) - R_t(\varepsilon, f). \tag{17}
$$

Plugging (16), (17) into (15) yields

$$f(z_0)^\top \hat{\mu}_t(1) \geq \Big(\sigma_t - \frac{R_t(\varepsilon, f)}{p_t(1)}\Big) f(z_0)^\top \mathbb{E}_{z \in \widetilde{C}_t(1) \cap S_t(\varepsilon, f)} \mathbb{E}_{z' \in \mathcal{A}(z)}[f(z')]$$

$$- B_2^2\Big(1 - \sigma_t + \frac{R_t(\varepsilon, f)}{p_t(1)}\Big) - B_2 \|\hat{\mu}_t(1) - \mu_t(1)\|_2. \tag{18}$$

Notice that $z_0 \in \widetilde{C}_t(1) \cap S_t(\varepsilon, f)$. Thus for any $z \in \widetilde{C}_t(1) \cap S_t(\varepsilon, f)$, by the definition of $\widetilde{C}_t(1)$, we have $\min_{z_0' \in \mathcal{A}(z_0), z' \in \mathcal{A}(z)} \|z_0' - z'\|_2 \leq \delta_t$. Further denote $(z_0^*, z^*) = \arg\min_{z_0' \in \mathcal{A}(z_0), z' \in \mathcal{A}(z)} \|z_0' - z'\|_2$, then $\|z_0^* - z^*\|_2 \leq \delta_t$, combining $\mathcal{K}$-Lipschitz property of $f$ to yield $\|f(z_0^*) - f(z^*)\|_2 \leq \mathcal{K}\|z_0^* - z^*\|_2 \leq \mathcal{K}\delta_t$. Besides that, since $z \in S_t(\varepsilon, f), \forall z' \in \mathcal{A}(z), \|f(z') - f(z^*)\|_2 \leq \varepsilon$. Similarly, as $z_0 \in S_t(\varepsilon, f)$ and $z_0, z_0^* \in \mathcal{A}(z_0)$, we know $\|f(z_0) - f(z_0^*)\|_2 \leq \varepsilon$.

Therefore,

$$f(z_0)^\top \mathbb{E}_{z \in \widetilde{C}_t(1) \cap S_t(\varepsilon, f)} \mathbb{E}_{z' \in \mathcal{A}(z)}[f(z')]$$

$$= \mathbb{E}_{z \in \widetilde{C}_t(1) \cap S_t(\varepsilon, f)} \mathbb{E}_{z' \in \mathcal{A}(z)}[f(z_0)^\top f(z')]$$

$$= \mathbb{E}_{z \in \widetilde{C}_t(1) \cap S_t(\varepsilon, f)} \mathbb{E}_{z' \in \mathcal{A}(z)}[f(z_0)^\top (f(z') - f(z_0) + f(z_0))]$$

$$\geq B_1^2 + \mathbb{E}_{z \in \widetilde{C}_t(1) \cap S_t(\varepsilon, f)} \mathbb{E}_{z' \in \mathcal{A}(z)}[f(z_0)^\top (f(z') - f(z_0))]$$

$$= B_1^2 + \mathbb{E}_{z \in \widetilde{C}_t(1) \cap S_t(\varepsilon, f)} \mathbb{E}_{z' \in \mathcal{A}(z)}[f(z_0)^\top \big( \underbrace{f(z') - f(z^*)}_{\|\cdot\|_2 \leq \varepsilon} + \underbrace{f(z^*) - f(z_0^*)}_{\|\cdot\|_2 \leq \mathcal{K}\delta_t} + \underbrace{f(z_0^*) - f(z_0)}_{\|\cdot\|_2 \leq \varepsilon} \big)]$$

$$\geq B_1^2 - (B_2\varepsilon + B_2\mathcal{K}\delta_t + B_2\varepsilon)$$

$$= B_1^2 - B_2(\mathcal{K}\delta_t + 2\varepsilon), \tag{19}$$

where the fourth row is derived from $\|f\|_2 \geq B_1$.

Plugging (19) to the inequality (18) knows

$$f(z_0)^\top \hat{\mu}_t(1) \geq \Big(\sigma_t - \frac{R_t(\varepsilon, f)}{p_t(1)}\Big) f(z_0)^\top \mathbb{E}_{z \in \widetilde{C}_t(1) \cap S_t(\varepsilon, f)} \mathbb{E}_{z' \in \mathcal{A}(z)}[f(z')] - B_2^2\Big(1 - \sigma_t + \frac{R_t(\varepsilon, f)}{p_t(1)}\Big)$$

$$- B_2\|\hat{\mu}_t(1) - \mu_t(1)\|_2$$

$$\geq \Big(\sigma_t - \frac{R_t(\varepsilon, f)}{p_t(1)}\Big)\big(B_1^2 - B_2(\mathcal{K}\delta_t + 2\varepsilon)\big) - B_2^2\Big(1 - \sigma_t + \frac{R_t(\varepsilon, f)}{p_t(1)}\Big)$$

$$- B_2\|\hat{\mu}_t(1) - \mu_t(1)\|_2$$

$$= B_2^2\Big(\Big(1 + \big(\frac{B_1}{B_2}\big)^2\Big)\Big(\sigma_t - \frac{R_t(\varepsilon, f)}{p_t(1)}\Big) - \Big(\sigma_t - \frac{R_t(\varepsilon, f)}{p_t(1)}\Big)\Big(\frac{\mathcal{K}\delta_t}{B_2} + \frac{2\varepsilon}{B_2}\Big) - 1\Big)$$

$$- B_2\|\hat{\mu}_t(1) - \mu_t(1)\|_2$$

$$= B_2^2\Big(\Big(\sigma_t - \frac{R_t(\varepsilon, f)}{p_t(1)}\Big)\Big(1 + \big(\frac{B_1}{B_2}\big)^2 - \frac{\mathcal{K}\delta_t}{B_2} - \frac{2\varepsilon}{B_2}\Big) - 1\Big) - B_2\|\hat{\mu}_t(1) - \mu_t(1)\|_2$$

$$= B_2^2 \Gamma_1(\sigma_t, \delta_t, \varepsilon, f) - B_2\|\hat{\mu}_t(1) - \mu_t(1)\|_2.$$

Similar as above proving process, we can also turn out

$$f(z_0)^\top \mu_t(1) \geq B_2^2 \Gamma_1(\sigma_t, \delta_t, \varepsilon, f). \tag{20}$$

Combining the fact that

$$\|\mu_t(k)\|_2 = \|\mathbb{E}_{z \in \widetilde{C}_t(k)} \mathbb{E}_{z' \in \mathcal{A}(z)}[f(z')]\|_2 \leq \mathbb{E}_{x \in \widetilde{C}_t(k)} \mathbb{E}_{z' \in \mathcal{A}(z)} \|f(z')\|_2 \leq B_2,$$

we can conclude

$$f(z_0)^\top \hat{\mu}_t(k) \leq f(z_0)^\top \mu_t(k) + f(z_0)^\top (\hat{\mu}_t(k) - \mu_t(k))$$

$$\leq f(z_0)^\top \mu_t(k) + \|f(z_0)\|_2 \|\hat{\mu}_t(k) - \mu_t(k)\|_2$$

$$\leq f(z_0)^\top \mu_t(k) + B_2 \|\hat{\mu}_t(k) - \mu_t(k)\|_2$$

$$= (f(\boldsymbol{z}_0) - \mu_t(1))^\top \mu_t(k) + \mu_t(1)^\top \mu_t(k) + B_2 \|\hat{\mu}_t(k) - \mu_t(k)\|_2$$

$$\leq \|f(\boldsymbol{z}_0) - \mu_t(1)\|_2 \cdot \|\mu_t(k)\|_2 + \mu_t(1)^\top \mu_t(k) + B_2 \|\hat{\mu}_t(k) - \mu_t(k)\|_2$$

$$\leq B_2 \sqrt{\|f(\boldsymbol{z}_0)\|_2^2 - 2f(\boldsymbol{z}_0)^\top \mu_t(1) + \|\mu_t(1)\|_2^2} + \mu_t(1)^\top \mu_t(k) + B_2 \|\hat{\mu}_t(k) - \mu_t(k)\|_2$$

$$\leq B_2 \sqrt{2B_2^2 - 2f(\boldsymbol{z}_0)^\top \mu_t(1)} + \mu_t(1)^\top \mu_t(k) + B_2 \|\hat{\mu}_t(k) - \mu_t(k)\|_2$$

$$\leq B_2 \sqrt{2B_2^2 - 2B_2^2 \Gamma_1(\sigma_t, \delta_t, \varepsilon, f)} + \mu_t(1)^\top \mu_t(k) + B_2 \|\hat{\mu}_t(k) - \mu_t(k)\|_2$$

$$= \sqrt{2}B_2^2 \sqrt{1 - \Gamma_1(\sigma_t, \delta_t, \varepsilon, f)} + \mu_t(1)^\top \mu_t(k) + B_2 \|\hat{\mu}_t(k) - \mu_t(k)\|_2.$$

Note that we plug (20) into the seventh row to obtain the inequality of eighth row.

Thus, by $\Delta_{\hat{\mu}_t} = 1 - \min_{k \in [K]} \|\hat{\mu}_t(k)\|_2^2 / B_2^2$, we can conclude

$$f(\boldsymbol{z}_0)^\top \hat{\mu}_t(1) - f(\boldsymbol{z}_0)^\top \hat{\mu}_t(k) - \left( \frac{1}{2} \|\hat{\mu}_t(1)\|_2^2 - \frac{1}{2} \|\hat{\mu}_t(k)\|^2 \right)$$

$$= f(\boldsymbol{z}_0)^\top \hat{\mu}_t(1) - f(\boldsymbol{z}_0)^\top \hat{\mu}_t(k) - \frac{1}{2} \|\hat{\mu}_t(1)\|_2^2 + \frac{1}{2} \|\hat{\mu}_t(k)\|_2^2$$

$$\geq f(\boldsymbol{z}_0)^\top \hat{\mu}_t(1) - f(\boldsymbol{z}_0)^\top \hat{\mu}_t(k) - \frac{1}{2} B_2^2 + \frac{1}{2} \min_{k \in [K]} \|\hat{\mu}_t(k)\|_2^2$$

$$= f(\boldsymbol{z}_0)^\top \hat{\mu}_t(1) - f(\boldsymbol{z}_0)^\top \hat{\mu}_t(k) - \frac{1}{2} B_2^2 \Delta_{\hat{\mu}_t}$$

$$\geq B_2^2 \Gamma_1(\sigma_t, \delta_t, \varepsilon, f) - B_2 \|\hat{\mu}_t(1) - \mu_t(1)\|_2 - \sqrt{2}B_2^2 \sqrt{1 - \Gamma_1(\sigma_t, \delta_t, \varepsilon, f)}$$

$$- \mu_t(1)^\top \mu_t(k) - B_2 \|\hat{\mu}_t(k) - \mu_t(k)\|_2 - \frac{1}{2} B_2^2 \Delta_{\hat{\mu}_t} > 0,$$

which finishes the proof. $\qquad\square$

### B.2.2 PRELIMINARIES FOR LEMMA B.4

To establish Lemma B.4, we must first prove Lemmas B.2 and B.3 in advance. Following the notations in the target domain, we employ $\mu_s(k) := \mathbb{E}_{\boldsymbol{x} \in C_s(k)} \mathbb{E}_{\boldsymbol{x}' \in \mathcal{A}(\boldsymbol{x})}[f(\boldsymbol{x}')] = \frac{1}{p_s(k)} \mathbb{E}_{\boldsymbol{x}} \mathbb{E}_{\boldsymbol{x}' \in \mathcal{A}(\boldsymbol{x})}[f(\boldsymbol{x}') \mathbb{1}\{\boldsymbol{x} \in C_s(k)\}]$ to denote the centre of $k$-th latent class in representation space. Apart from that, it is necessary to introduce following assumption, which is the abstract version of Assumption 4.4.

**Assumption B.1.** Review $P_s(k)$ and $P_t(k)$ are the conditional measures that $P(\boldsymbol{x}|\boldsymbol{x} \in C_s(k))$ and $P(\boldsymbol{z}|\boldsymbol{z} \in C_t(k))$ respectively, assume $\exists \rho > 0$ and $\eta > 0$, $\max_{k \in [K]} \mathcal{W}(P_s(k), P_t(k)) \leq \rho$ and $\max_{k \in [K]} |p_s(k) - p_t(k)| \leq \eta$.

**Lemma B.2.** *If the encoder $f$ is $\mathcal{K}$-Lipschitz and Assumption B.1 holds, for any $k \in [K]$, we have:*

$$\|\mu_s(k) - \mu_t(k)\|_2 \leq \sqrt{d^*} M \mathcal{K} \rho.$$

*Proof.* For all $k \in [K]$,

$$\|\mu_s(k) - \mu_t(k)\|_2^2 = \sum_{l=1}^{d^*} \left( (\mu_s(k))_l - (\mu_t(k))_l \right)^2$$

$$= \sum_{l=1}^{d^*} (\mathbb{E}_{\boldsymbol{x} \in C_s(k)} \mathbb{E}_{\boldsymbol{x}' \in \mathcal{A}(\boldsymbol{x})}[f_l(\boldsymbol{x}')] - \mathbb{E}_{\boldsymbol{z} \in C_t(k)} \mathbb{E}_{\boldsymbol{z}' \in \mathcal{A}(\boldsymbol{z})}[f_l(\boldsymbol{z}')])^2$$

$$= \sum_{l=1}^{d^*} \left[ \frac{1}{m} \sum_{\gamma=1}^{m} \left( \mathbb{E}_{\boldsymbol{x} \in C_s(k)}[f_l(A_\gamma(\boldsymbol{x}))] - \mathbb{E}_{\boldsymbol{z} \in C_t(k)}[f_l(A_\gamma(\boldsymbol{z}))] \right) \right]^2$$

$$\leq d^* M^2 \mathcal{K}^2 \rho^2$$

The final inequality is obtained by Assumption B.1 along with the fact that $f(A_\gamma(\cdot))$ is $M\mathcal{K}$-Lipschitz continuous. In fact, as $f \in \text{Lip}(\mathcal{K})$, then for every $l \in [d^*], f_l \in \text{Lip}(\mathcal{K})$, combining the property that $A_\gamma(\cdot) \in \text{Lip}(M)$ stated in Assumption 4.2, we can turn out $f(A_\gamma(\cdot))$ is $M\mathcal{K}$-Lipschitz continuous.

So that

$$\|\mu_s(k) - \mu_t(k)\|_2 \leq \sqrt{d^*} M\mathcal{K}\rho.$$

$\square$

**Lemma B.3.** *Given a* $(\sigma_s, \sigma_t, \delta_s, \delta_t)$*-augmentation, if the encoder $f$ with $\|f\|_2 \leq B_2$ is $\mathcal{K}$-Lipschitz continuous, then*

$$\mathop{\mathbb{E}}_{\boldsymbol{x} \in C_s(k)} \mathop{\mathbb{E}}_{\boldsymbol{x}_1 \in \mathcal{A}(\boldsymbol{x})} \|f(\boldsymbol{x}_1) - \mu_s(k)\|_2^2 \leq 4B_2^2 \Big[ \Big(1 - \sigma_s + \frac{\mathcal{K}\delta_s + 2\varepsilon}{2B_2} + \frac{R_s(\varepsilon, f)}{p_s(k)}\Big)^2 + \Big(1 - \sigma_s + \frac{R_s(\varepsilon, f)}{p_s(k)}\Big)\Big],$$

*where $R_s(\varepsilon, f) = P_s\big(\boldsymbol{x} \in \cup_{k=1}^K C_s(k) : \sup_{\boldsymbol{x}_1, \boldsymbol{x}_2 \in \mathcal{A}(\boldsymbol{x})} \|f(\boldsymbol{x}_1) - f(\boldsymbol{x}_2)\|_2 > \varepsilon\big)$.*

*Proof.* Let $S_s(\varepsilon, f) := \{\boldsymbol{x} \in \cup_{k=1}^K C_s(k) : \sup_{\boldsymbol{x}_1, \boldsymbol{x}_2 \in \mathcal{A}(\boldsymbol{x})} \|f(\boldsymbol{x}_1) - f(\boldsymbol{x}_2)\|_2 \leq \varepsilon\}$, for each $k \in [K]$,

$$\mathbb{E}_{\boldsymbol{x} \in C_s(k)} \mathbb{E}_{\boldsymbol{x}_1 \in \mathcal{A}(\boldsymbol{x})} \|f(\boldsymbol{x}_1) - \mu_s(k)\|_2^2$$

$$= \frac{1}{p_s(k)} \mathbb{E}_{\boldsymbol{x}} \mathbb{E}_{\boldsymbol{x}_1 \in \mathcal{A}(\boldsymbol{x})} [\mathbb{1}\{\boldsymbol{x} \in C_s(k)\} \|f(\boldsymbol{x}_1) - \mu_s(k)\|_2^2]$$

$$= \frac{1}{p_s(k)} \mathbb{E}_{\boldsymbol{x}} \mathbb{E}_{\boldsymbol{x}_1 \in \mathcal{A}(\boldsymbol{x})} [\mathbb{1}\{\boldsymbol{x} \in \widetilde{C}_s(k) \cap S_s(\varepsilon, f)\} \|f(\boldsymbol{x}_1) - \mu_s(k)\|_2^2]$$

$$\qquad + \frac{1}{p_s(k)} \mathbb{E}_{\boldsymbol{x}} \mathbb{E}_{\boldsymbol{x}_1 \in \mathcal{A}(\boldsymbol{x})} [\mathbb{1}\{\boldsymbol{x} \in C_s(k) \backslash (\widetilde{C}_s(k) \cap S_s(\varepsilon, f))\} \|f(\boldsymbol{x}_1) - \mu_s(k)\|_2^2]$$

$$\leq \frac{1}{p_s(k)} \mathbb{E}_{\boldsymbol{x}} \mathbb{E}_{\boldsymbol{x}_1 \in \mathcal{A}(\boldsymbol{x})} [\mathbb{1}\{\boldsymbol{x} \in \widetilde{C}_s(k) \cap S_s(\varepsilon, f)\} \|f(\boldsymbol{x}_1) - \mu_s(k)\|_2^2]$$

$$\qquad + \frac{4B_2^2 P_s\big(C_s(k) \backslash (\widetilde{C}_s(k) \cap S_s(\varepsilon, f))\big)}{p_s(k)}$$

$$\leq \frac{1}{p_s(k)} \mathop{\mathbb{E}}_{\boldsymbol{x}} \mathop{\mathbb{E}}_{\boldsymbol{x}_1 \in \mathcal{A}(\boldsymbol{x})} [\mathbb{1}\{\boldsymbol{x} \in \widetilde{C}_s(k) \cap S_s(\varepsilon, f)\} \|f(\boldsymbol{x}_1) - \mu_s(k)\|_2^2] + 4B_2^2 \Big(1 - \sigma_s + \frac{R_s(\varepsilon, f)}{p_s(k)}\Big)$$

$$\leq \frac{P_s(\widetilde{C}_s(k) \cap S_s(\varepsilon, f))}{p_s(k)} \mathop{\mathbb{E}}_{\boldsymbol{x} \in \widetilde{C}_s(k) \cap S_s(\varepsilon, f)} \mathop{\mathbb{E}}_{\boldsymbol{x}_1 \in \mathcal{A}(\boldsymbol{x})} \|f(\boldsymbol{x}_1) - \mu_s(k)\|_2^2 + 4B_2^2 \Big(1 - \sigma_s + \frac{R_s(\varepsilon, f)}{p_s(k)}\Big)$$

$$\leq \mathbb{E}_{\boldsymbol{x} \in \widetilde{C}_s(k) \cap S_s(\varepsilon, f)} \mathbb{E}_{\boldsymbol{x}_1 \in \mathcal{A}(\boldsymbol{x})} \|f(\boldsymbol{x}_1) - \mu_s(k)\|_2^2 + 4B_2^2 \Big(1 - \sigma_s + \frac{R_s(\varepsilon, f)}{p_s(k)}\Big), \tag{21}$$

the second inequality is due to

$$P_s\big(C_s(k) \backslash ((\widetilde{C}_s(k) \cap S_s(\varepsilon, f)))\big) = P_s\big((C_s(k) \backslash \widetilde{C}_s(k)) \cup (C_s(k) \backslash S_s(\varepsilon, f))\big)$$
$$\leq (1 - \sigma_s) p_s(k) + R_s(\varepsilon, f).$$

Furthermore,

$$\mathbb{E}_{\boldsymbol{x} \in \widetilde{C}_s(k) \cap S_s(\varepsilon, f)} \mathbb{E}_{\boldsymbol{x}_1 \in \mathcal{A}(\boldsymbol{x})} \|f(\boldsymbol{x}_1) - \mu_s(k)\|_2^2$$

$$= \mathbb{E}_{\boldsymbol{x} \in \widetilde{C}_s(k) \cap S_s(\varepsilon, f)} \mathbb{E}_{\boldsymbol{x}_1 \in \mathcal{A}(\boldsymbol{x})} \|f(\boldsymbol{x}_1) - \mathbb{E}_{\boldsymbol{x}' \in C_s(k)} \mathbb{E}_{\boldsymbol{x}_2 \in \mathcal{A}(\boldsymbol{x}')} f(\boldsymbol{x}_2)\|_2^2$$

$$= \mathbb{E}_{\boldsymbol{x} \in \widetilde{C}_s(k) \cap S_s(\varepsilon, f)} \mathbb{E}_{\boldsymbol{x}_1 \in \mathcal{A}(\boldsymbol{x})} \Big\| f(\boldsymbol{x}_1) - \frac{P(\widetilde{C}_s(k) \cap S_s(\varepsilon, f))}{p_s(k)} \mathbb{E}_{\boldsymbol{x}' \in \widetilde{C}_s(k) \cap S_s(\varepsilon, f)} \mathbb{E}_{\boldsymbol{x}_2 \in \mathcal{A}(\boldsymbol{x}')} f(\boldsymbol{x}_2)$$

$$\qquad - \frac{P_s\big(C_s(k) \backslash (\widetilde{C}_s(k) \cap S_s(\varepsilon, f))\big)}{p_s(k)} \mathbb{E}_{\boldsymbol{x}' \in C_s(k) \backslash (\widetilde{C}_s(k) \cap S_s(\varepsilon, f))} \mathbb{E}_{\boldsymbol{x}_2 \in \mathcal{A}(\boldsymbol{x}')} f(\boldsymbol{x}_2) \Big\|_2^2$$

$$= \mathbb{E}_{\boldsymbol{x} \in \widetilde{C}_s(k) \cap S_s(\varepsilon, f)} \mathbb{E}_{\boldsymbol{x}_1 \in \mathcal{A}(\boldsymbol{x})} \Big\| \frac{P_s(\widetilde{C}_s(k) \cap S_s(\varepsilon, f))}{p_s(k)} \Big( f(\boldsymbol{x}_1) - \mathbb{E}_{\boldsymbol{x}' \in \widetilde{C}_s(k) \cap S_s(\varepsilon, f)} \mathbb{E}_{\boldsymbol{x}_2 \in \mathcal{A}(\boldsymbol{x}')} f(\boldsymbol{x}_2) \Big)$$

$$- \frac{P_s\big(C_s(k)\backslash(\widetilde{C}_s(k) \cap S_s(\varepsilon, f)))}{p_s(k)} \Big(f(\boldsymbol{x}_1) - \mathbb{E}_{\boldsymbol{x}' \in C_s(k)\backslash(\widetilde{C}_s(k) \cap S_s(\varepsilon, f))} \mathbb{E}_{\boldsymbol{x}_2 \in \mathcal{A}(\boldsymbol{x}')} f(\boldsymbol{x}_2)\Big)\Big\|_2^2$$

$$\leq \mathbb{E}_{\boldsymbol{x} \in \widetilde{C}_s(k) \cap S_s(\varepsilon, f)} \mathbb{E}_{\boldsymbol{x}_1 \in \mathcal{A}(\boldsymbol{x})} \Big[\Big\|f(\boldsymbol{x}_1) - \mathbb{E}_{\boldsymbol{x}' \in \widetilde{C}_s(k) \cap S_s(\varepsilon, f)} \mathbb{E}_{\boldsymbol{x}_2 \in \mathcal{A}(\boldsymbol{x}')} f(\boldsymbol{x}_2)\Big\|_2 + 2B_2\Big(1 - \sigma_s + \frac{R_s(\varepsilon, f)}{p_s(k)}\Big)\Big]^2 \tag{22}$$

For any $\boldsymbol{x}, \boldsymbol{x}' \in \widetilde{C}_s(k) \cap S_s(\varepsilon, f)$, by the definition of $\widetilde{C}_s(k)$, we can yield that

$$\min_{\boldsymbol{x}_1 \in \mathcal{A}(\boldsymbol{x}), \boldsymbol{x}_2 \in \mathcal{A}(\boldsymbol{x}')} \|\boldsymbol{x}_1 - \boldsymbol{x}_2\|_2 \leq \delta_s,$$

thus if we denote $(\boldsymbol{x}_1^*, \boldsymbol{x}_2^*) = \arg\min_{\boldsymbol{x}_1 \in \mathcal{A}(\boldsymbol{x}), \boldsymbol{x}_2 \in \mathcal{A}(\boldsymbol{x}')} \|\boldsymbol{x}_1 - \boldsymbol{x}_2\|_2$, we can turn out $\|\boldsymbol{x}_1^* - \boldsymbol{x}_2^*\|_2 \leq \delta_s$, further by $\mathcal{K}$-Lipschitz continuity of $f$, we yield $\|f(\boldsymbol{x}_1^*) - f(\boldsymbol{x}_2^*)\|_2 \leq \mathcal{K}\|\boldsymbol{x}_1^* - \boldsymbol{x}_2^*\|_2 \leq \mathcal{K}\delta_s$. In addition, since $\boldsymbol{x} \in S_s(\varepsilon, f)$, we know for any $\boldsymbol{x}_1 \in \mathcal{A}(\boldsymbol{x})$, $\|f(\boldsymbol{x}_1) - f(\boldsymbol{x}_1^*)\|_2 \leq \varepsilon$. Similarly, $\boldsymbol{x}' \in S_s(\varepsilon, f)$ implies $\|f(\boldsymbol{x}_2) - f(\boldsymbol{x}_2^*)\|_2 \leq \varepsilon$ for any $\boldsymbol{x}_2 \in \mathcal{A}(\boldsymbol{x}')$. Therefore, for any $\boldsymbol{x}, \boldsymbol{x}' \in \widetilde{C}_s(1) \cap S_s(\varepsilon, f)$ and $\boldsymbol{x}_1 \in \mathcal{A}(\boldsymbol{x}), \boldsymbol{x}_2 \in \mathcal{A}(\boldsymbol{x}')$,

$$\|f(\boldsymbol{x}_1) - f(\boldsymbol{x}_2)\|_2 \leq \|f(\boldsymbol{x}_1) - f(\boldsymbol{x}_1^*)\|_2 + \|f(\boldsymbol{x}_1^*) - f(\boldsymbol{x}_2^*)\|_2 + \|f(\boldsymbol{x}_2^*) - f(\boldsymbol{x}_2)\|_2 \leq 2\varepsilon + \mathcal{K}\delta_s. \tag{23}$$

Combining inequalities (21), (22), (23) to conclude

$$\mathbb{E}_{\boldsymbol{x} \in C_s(k)} \mathbb{E}_{\boldsymbol{x}_1 \in \mathcal{A}(\boldsymbol{x})} \|f(\boldsymbol{x}_1) - \mu_s(k)\|_2^2$$

$$\leq \Big[2\varepsilon + \mathcal{K}\delta_s + 2B_2\Big(1 - \sigma_s + \frac{R_s(\varepsilon, f)}{p_s(k)}\Big)\Big]^2 + 4B_2^2\Big(1 - \sigma_s + \frac{R_s(\varepsilon, f)}{p_s(k)}\Big)$$

$$= 4B_2^2\Big[\Big(1 - \sigma_s + \frac{\mathcal{K}\delta_s}{2B_2} + \frac{\varepsilon}{B_2} + \frac{R_s(\varepsilon, f)}{p_s(k)}\Big)^2 + \Big(1 - \sigma_s + \frac{R_s(\varepsilon, f)}{p_s(k)}\Big)\Big]$$

$\square$

### B.2.3 THE EFFECT OF MINIMAXING OUR LOSS

**Lemma B.4.** *Given a $(\sigma_s, \sigma_t, \delta_s, \delta_t)$-augmentation, if $d^* > K$ and the encoder $f$ with $B_1 \leq \|f\|_2 \leq B_2$ is $\mathcal{K}$-Lipschitz continuous, then for any $\varepsilon > 0$,*

$$R_s^2(\varepsilon, f) \leq \frac{m^4}{\varepsilon^2} \mathcal{L}_{\text{align}}(f),$$

$$R_t^2(\varepsilon, f) \leq \frac{m^4}{\varepsilon^2} \mathcal{L}_{\text{align}}(f) + \frac{8m^4}{\varepsilon^2} B_2 d^* M \mathcal{K}\rho + \frac{4m^4}{\varepsilon^2} B_2^2 d^* K\eta,$$

*and*

$$\max_{i \neq j} |\mu_t(i)^\top \mu_t(j)| \leq \sqrt{\frac{2}{\min_{i \neq j} p_s(i)p_s(j)}\Big(\mathcal{L}_{\text{div}}(f) + \varphi(\sigma_s, \delta_s, \varepsilon, f)\Big)} + 2\sqrt{d^*} B_2 M \mathcal{K}\rho.$$

*where $R_s(\varepsilon, f) = P_s\big(\boldsymbol{x} \in \cup_{k=1}^K C_s(k) : \sup_{\boldsymbol{x}_1, \boldsymbol{x}_2 \in \mathcal{A}(\boldsymbol{x})} \|f(\boldsymbol{x}_1) - f(\boldsymbol{x}_2)\| > \varepsilon\big)$ and $\varphi(\sigma_s, \delta_s, \varepsilon, f)$ $:= 4B_2^2\Big[\Big(1 - \sigma_s + \frac{\mathcal{K}\delta_s + 2\varepsilon}{2B_2}\Big)^2 + (1 - \sigma_s) + K R_s(\varepsilon, f)\Big(3 - 2\sigma_s + \frac{\mathcal{K}\delta_s + 2\varepsilon}{B_2}\Big) + R_s^2(\varepsilon, f)\Big(\sum_{k=1}^K \frac{1}{p_s(k)}\Big)\Big] + B_2(\varepsilon^2 + 4B_2^2 R_s(\varepsilon, f))^{\frac{1}{2}}.$*

*Proof.* Recall the Assumption 4.2, the measure on $\mathcal{A}$ is uniform, thus

$$\mathbb{E}_{\boldsymbol{z}_1, \boldsymbol{z}_2 \in \mathcal{A}(\boldsymbol{z})} \|f(\boldsymbol{z}_1) - f(\boldsymbol{z}_2)\|_2 = \frac{1}{m^2} \sum_{\gamma=1}^m \sum_{\beta=1}^m \|f(A_\gamma(\boldsymbol{z})) - f(A_\beta(\boldsymbol{z}))\|_2.$$

so that

$$\sup_{\boldsymbol{z}_1, \boldsymbol{z}_2 \in \mathcal{A}(\boldsymbol{z})} \|f(\boldsymbol{z}_1) - f(\boldsymbol{z}_2)\|_2 = \sup_{\gamma, \beta \in [m]} \|f(A_\gamma(\boldsymbol{z})) - f(A_\beta(\boldsymbol{z}))\|_2$$

$$\leq \sum_{\gamma=1}^{m} \sum_{\beta=1}^{m} \|f(A_\gamma(\boldsymbol{z})) - f(A_\beta(\boldsymbol{z}))\|_2$$

$$= m^2 \mathbb{E}_{\boldsymbol{z}_1, \boldsymbol{z}_2 \in \mathcal{A}(\boldsymbol{z})} \|f(\boldsymbol{z}_1) - f(\boldsymbol{z}_2)\|_2.$$

Denote $S := \{\boldsymbol{z} : \mathbb{E}_{\boldsymbol{z}_1, \boldsymbol{z}_2 \in \mathcal{A}(\boldsymbol{z})} \|f(\boldsymbol{z}_1) - f(\boldsymbol{z}_2)\|_2 > \frac{\varepsilon}{m^2}\}$, by the definition of $R_t(\varepsilon, f)$ along with Markov inequality, we have

$$R_t^2(\varepsilon, f) \leq P_t^2(S)$$

$$\leq \Big( \frac{\mathbb{E}_{\boldsymbol{z}} \mathbb{E}_{\boldsymbol{z}_1, \boldsymbol{z}_2 \in \mathcal{A}(\boldsymbol{z})} \|f(\boldsymbol{z}_1) - f(\boldsymbol{z}_2)\|_2}{\frac{\varepsilon}{m^2}} \Big)^2$$

$$\leq \frac{\mathbb{E}_{\boldsymbol{z}} \mathbb{E}_{\boldsymbol{z}_1, \boldsymbol{z}_2 \in \mathcal{A}(\boldsymbol{z})} \|f(\boldsymbol{z}_1) - f(\boldsymbol{z}_2)\|_2^2}{\frac{\varepsilon^2}{m^4}}$$

$$= \frac{m^4}{\varepsilon^2} \mathbb{E}_{\boldsymbol{z}} \mathbb{E}_{\boldsymbol{z}_1, \boldsymbol{z}_2 \in \mathcal{A}(\boldsymbol{z})} \|f(\boldsymbol{z}_1) - f(\boldsymbol{z}_2)\|_2^2 \tag{24}$$

Similar as above process, we can also get the first part stated in Lemma B.4:

$$R_s^2(\varepsilon, f) \leq \frac{m^4}{\varepsilon^2} \mathbb{E}_{\boldsymbol{x}} \mathbb{E}_{\boldsymbol{x}_1, \boldsymbol{x}_2 \in \mathcal{A}(\boldsymbol{x})} \|f(\boldsymbol{x}_1) - f(\boldsymbol{x}_2)\|_2^2 = \frac{m^4}{\varepsilon^2} \mathcal{L}_{\text{align}}(f).$$

Besides that, we can turn out

$$\mathbb{E}_{\boldsymbol{z}} \mathbb{E}_{\boldsymbol{z}_1, \boldsymbol{z}_2 \in \mathcal{A}(\boldsymbol{z})} \|f(\boldsymbol{z}_1) - f(\boldsymbol{z}_2)\|_2^2$$

$$= \mathbb{E}_{\boldsymbol{x}} \mathbb{E}_{\boldsymbol{x}_1, \boldsymbol{x}_2 \in \mathcal{A}(\boldsymbol{x})} \|f(\boldsymbol{x}_1) - f(\boldsymbol{x}_2)\|_2^2 + \mathbb{E}_{\boldsymbol{z}} \mathbb{E}_{\boldsymbol{z}_1, \boldsymbol{z}_2 \in \mathcal{A}(\boldsymbol{z})} \|f(\boldsymbol{z}_1) - f(\boldsymbol{z}_2)\|_2^2$$

$$\quad - \mathbb{E}_{\boldsymbol{x}} \mathbb{E}_{\boldsymbol{x}_1, \boldsymbol{x}_2 \in \mathcal{A}(\boldsymbol{x})} \|f(\boldsymbol{x}_1) - f(\boldsymbol{x}_2)\|_2^2$$

$$= \frac{1}{m^2} \sum_{\gamma=1}^{m} \sum_{\beta=1}^{m} \Big[ \mathbb{E}_{\boldsymbol{z}} \|f(A_\gamma(\boldsymbol{z})) - f(A_\beta(\boldsymbol{z}))\|_2^2 - \mathbb{E}_{\boldsymbol{x}} \|f(A_\gamma(\boldsymbol{x})) - f(A_\beta(\boldsymbol{x}))\|_2^2 \Big]$$

$$\quad + \mathbb{E}_{\boldsymbol{x}} \mathbb{E}_{\boldsymbol{x}_1, \boldsymbol{x}_2 \in \mathcal{A}(\boldsymbol{x})} \|f(\boldsymbol{x}_1) - f(\boldsymbol{x}_2)\|_2^2$$

$$= \frac{1}{m^2} \sum_{\gamma=1}^{m} \sum_{\beta=1}^{m} \sum_{l=1}^{d^*} \Big[ \mathbb{E}_{\boldsymbol{z}} \big[f_l(A_\gamma(\boldsymbol{z})) - f_l(A_\beta(\boldsymbol{z}))\big]^2 - \mathbb{E}_{\boldsymbol{x}} \big[f_l(A_\gamma(\boldsymbol{x})) - f_l(A_\beta(\boldsymbol{x}))\big]^2 \Big]$$

$$\quad + \mathbb{E}_{\boldsymbol{x}} \mathbb{E}_{\boldsymbol{x}_1, \boldsymbol{x}_2 \in \mathcal{A}(\boldsymbol{x})} \|f(\boldsymbol{x}_1) - f(\boldsymbol{x}_2)\|_2^2,$$

since for all $\gamma \in [m], \beta \in [m]$ and $l \in [d^*]$, we have

$$\mathbb{E}_{\boldsymbol{z}} \big[f_l(A_\gamma(\boldsymbol{z})) - f_l(A_\beta(\boldsymbol{z}))\big]^2 - \mathbb{E}_{\boldsymbol{x}} \big[f_l(A_\gamma(\boldsymbol{x})) - f_l(A_\beta(\boldsymbol{x}))\big]^2$$

$$= \sum_{k=1}^{K} \Big[ p_t(k) \mathbb{E}_{\boldsymbol{z} \in C_t(k)} \big[f_l(A_\gamma(\boldsymbol{z})) - f_l(A_\beta(\boldsymbol{z}))\big]^2 - p_s(k) \mathbb{E}_{\boldsymbol{x} \in C_s(k)} \big[f_l(A_\gamma(\boldsymbol{x})) - f_l(A_\beta(\boldsymbol{x}))\big]^2 \Big]$$

$$= \sum_{k=1}^{K} \Big[ p_t(k) \Big( \mathbb{E}_{\boldsymbol{z} \in C_t(k)} \big[f_l(A_\gamma(\boldsymbol{z})) - f_l(A_\beta(\boldsymbol{z}))\big]^2 - \mathbb{E}_{\boldsymbol{x} \in C_s(k)} \underbrace{\big[f_l(A_\gamma(\boldsymbol{x})) - f_l(A_\beta(\boldsymbol{x}))\big]^2}_{g(\boldsymbol{x})} \Big)$$

$$\quad + \big(p_t(k) - p_s(k)\big) \mathbb{E}_{\boldsymbol{x} \in C_s(k)} \big[f_l(A_\gamma(\boldsymbol{x})) - f_l(A_\beta(\boldsymbol{x}))\big]^2 \Big]$$

$$\leq 8 B_2 M \mathcal{K} \rho + 4 B_2^2 K \eta.$$

It is necessary to claim $g(\boldsymbol{x}) \in \text{Lip}(8B_2 M \mathcal{K})$ at first to obtain the last inequality shown above. In fact, $\forall l \in [d^*], f_l \in \text{Lip}(\mathcal{K})$ as $f \in \text{Lip}(\mathcal{K})$, and review that $A_\gamma(\cdot)$ and $A_\beta(\cdot)$ are both $M$-Lipschitz continuous according to Assumption 4.2, therefore we can turn out $f_l(A_\gamma(\cdot)) - f_l(A_\beta(\cdot)) \in \text{Lip}(2M\mathcal{K})$. In addition, note that $|f_l(A_\gamma(\cdot)) - f_l(A_\beta(\cdot))| \leq 2B_2$ as $\|f\|_2 \leq B_2$, hence the outermost quadratic function remains locally $4B_2$-Lipschitz continuity in $[-2B_2, 2B_2]$, which implies that $g \in \text{Lip}(8B_2 M \mathcal{K})$.

Now let's separately derive the two terms of the last inequality, combine the conclusion that $g \in \mathrm{Lip}(8B_2 M\mathcal{K})$, the definition of Wasserstein distance and Assumption B.1 can obtain

$$\sum_{k=1}^{K} \left[ p_t(k) \Big( \mathbb{E}_{\boldsymbol{z} \in C_t(k)} \big[ f_l(A_\gamma(\boldsymbol{z})) - f_l(A_\beta(\boldsymbol{z})) \big]^2 - \mathbb{E}_{\boldsymbol{x} \in C_s(k)} \big[ f_l(A_\gamma(\boldsymbol{x})) - f_l(A_\beta(\boldsymbol{x})) \big]^2 \Big) \right]$$

$$\leq 8B_2 M\mathcal{K}\rho \sum_{k=1}^{K} p_t(k)$$

$$= 8B_2 M\mathcal{K}\rho,$$

For the second term in the last inequality, just need to notice that $f_l(A_\gamma(\boldsymbol{x})) - f_l(A_\beta(\boldsymbol{x})) \leq 2B_2$, and then apply Assumption B.1 to yield

$$\sum_{k=1}^{K} \left[ (p_t(k) - p_s(k)) \mathbb{E}_{\boldsymbol{x} \in C_s(k)} \big[ f_l(A_\gamma(\boldsymbol{x})) - f_l(A_\beta(\boldsymbol{x})) \big]^2 \right] \leq 4B_2^2 K\eta.$$

Hence we have

$$\mathbb{E}_{\boldsymbol{z}} \mathbb{E}_{\boldsymbol{z}_1, \boldsymbol{z}_2 \in \mathcal{A}(\boldsymbol{z})} \|f(\boldsymbol{z}_1) - f(\boldsymbol{z}_2)\|_2^2 \leq \mathbb{E}_{\boldsymbol{x}} \mathbb{E}_{\boldsymbol{x}_1, \boldsymbol{x}_2 \in \mathcal{A}(\boldsymbol{x})} \|f(\boldsymbol{x}_1) - f(\boldsymbol{x}_2)\|_2^2 + 8B_2 d^* M\mathcal{K}\rho + 4B_2^2 d^* K\eta. \tag{25}$$

Combining (24) and (25) turn out the second inequality of Lemma B.4.

$$R_t^2(\varepsilon, f) \leq \frac{m^4}{\varepsilon^2} \mathcal{L}_{\mathrm{align}}(f) + \frac{8m^4}{\varepsilon^2} B_2 d^* M\mathcal{K}\rho + \frac{4m^4}{\varepsilon^2} B_2^2 d^* K\eta.$$

To prove the third part of this Lemma, first recall Lemma B.2 that $\forall k \in [K]$,

$$\|\mu_s(k) - \mu_t(k)\|_2 \leq \sqrt{d^*} M\mathcal{K}\rho.$$

Hence, $\forall i \neq j$, we have

$$\begin{aligned} |\mu_t(i)^\top \mu_t(j) - \mu_s(i)^\top \mu_s(j)| &= |\mu_t(i)^\top \mu_t(j) - \mu_t(i)^\top \mu_s(j) + \mu_t(i)^\top \mu_s(j) - \mu_s(i)^\top \mu_s(j)| \\ &\leq \|\mu_t(i)\|_2 \|\mu_t(j) - \mu_s(j)\|_2 + \|\mu_s(j)\|_2 \|\mu_t(i) - \mu_s(i)\|_2 \\ &\leq 2\sqrt{d^*} B_2 M\mathcal{K}\rho, \end{aligned}$$

so that we can further yield the relationship of class center divergence between the source domain and the target domain:

$$\max_{i \neq j} |\mu_t(i)^\top \mu_t(j)| \leq \max_{i \neq j} |\mu_s(i)^\top \mu_s(j)| + 2\sqrt{d^*} B_2 M\mathcal{K}\rho. \tag{26}$$

Next, we will attempt to derive an upper bound for $\max_{i \neq j} |\mu_s(i)^\top \mu_s(j)|$. To do this, let $U = \left( \sqrt{p_s(1)}\mu_s(1), \ldots, \sqrt{p_s(K)}\mu_s(K) \right) \in \mathbb{R}^{d^* \times K}$, then

$$\begin{aligned} \left\| \sum_{k=1}^{K} p_s(k)\mu_s(k)\mu_s(k)^\top - I_{d^*} \right\|_F^2 &= \|UU^\top - I_{d^*}\|_F^2 \\ &= \mathrm{Tr}(UU^\top UU^\top - 2UU^\top + I_{d^*}) \quad (\|A\|_F^2 = \mathrm{Tr}(A^\top A)) \\ &= \mathrm{Tr}(U^\top UU^\top U - 2U^\top U) + \mathrm{Tr}(I_K) + d^* - K \\ &\qquad\qquad\qquad\qquad\qquad\qquad (\mathrm{Tr}(AB) = \mathrm{Tr}(BA)) \\ &\geq \|U^\top U - I_K\|_F^2 \qquad\qquad\qquad (d^* > K) \\ &= \sum_{k=1}^{K} \sum_{l=1}^{K} (\sqrt{p_s(k)p_s(l)}\mu_s(k)^\top \mu_s(l) - \delta_{kl})^2 \\ &\geq p_s(i)p_s(j)(\mu_s(i)^\top \mu_s(j))^2. \end{aligned}$$

Therefore,

$$(\mu_s(i)^\top \mu_s(j))^2 \leq \frac{\left\| \sum_{k=1}^{K} p_s(k)\mu_s(k)\mu_s(k)^\top - I_{d^*} \right\|_F^2}{p_s(i)p_s(j)}$$

$$
= \frac{\left\| \mathbb{E}_{\boldsymbol{x}} \mathbb{E}_{\boldsymbol{x}_1,\boldsymbol{x}_2 \in \mathcal{A}(\boldsymbol{x})}[f(\boldsymbol{x}_1)f(\boldsymbol{x}_2)^\top] - I_{d^*} + \sum_{k=1}^{K} p_s(k)\mu_s(k)\mu_s(k)^\top - \mathbb{E}_{\boldsymbol{x}} \mathbb{E}_{\boldsymbol{x}_1,\boldsymbol{x}_2 \in \mathcal{A}(\boldsymbol{x})}[f(\boldsymbol{x}_1)f(\boldsymbol{x}_2)^\top] \right\|_F^2}{p_s(i)p_s(j)}
$$

$$
\leq \frac{2\left\| \mathbb{E}_{\boldsymbol{x}} \mathbb{E}_{\boldsymbol{x}_1,\boldsymbol{x}_2 \in \mathcal{A}(\boldsymbol{x})}[f(\boldsymbol{x}_1)f(\boldsymbol{x}_2)^\top] - I_{d^*} \right\|_F^2 + 2\left\| \sum_{k=1}^{K} p_s(k)\mu_s(k)\mu_s(k)^\top - \mathbb{E}_{\boldsymbol{x}} \mathbb{E}_{\boldsymbol{x}_1,\boldsymbol{x}_2 \in \mathcal{A}(\boldsymbol{x})}[f(\boldsymbol{x}_1)f(\boldsymbol{x}_2)^\top] \right\|_F^2}{p_s(i)p_s(j)}
$$

$$(27)$$

For the term $\left\| \sum_{k=1}^{K} p_s(k)\mu_s(k)\mu_s(k)^\top - \mathbb{E}_{\boldsymbol{x}} \mathbb{E}_{\boldsymbol{x}_1,\boldsymbol{x}_2 \in \mathcal{A}(\boldsymbol{x})}[f(\boldsymbol{x}_1)f(\boldsymbol{x}_2)^\top] \right\|_F^2$, note that

$$
\mathbb{E}_{\boldsymbol{x}} \mathbb{E}_{\boldsymbol{x_1},\boldsymbol{x_2} \in \mathcal{A}(\boldsymbol{x})}[f(\boldsymbol{x}_1)f(\boldsymbol{x}_2)^\top] - \sum_{k=1}^{K} p_s(k)\mu_s(k)\mu_s(k)^\top
$$

$$
= \sum_{k=1}^{K} p_s(k)\mathbb{E}_{\boldsymbol{x} \in C_s(k)} \mathbb{E}_{\boldsymbol{x}_1,\boldsymbol{x}_2 \in \mathcal{A}(\boldsymbol{x})}[f(\boldsymbol{x}_1)f(\boldsymbol{x}_2)^\top] - \sum_{k=1}^{K} p_s(k)\mu_s(k)\mu_s(k)^\top
$$

$$
= \sum_{k=1}^{K} p_s(k)\mathbb{E}_{\boldsymbol{x} \in C_s(k)} \mathbb{E}_{\boldsymbol{x_1} \in \mathcal{A}(\boldsymbol{x})}[f(\boldsymbol{x}_1)f(\boldsymbol{x}_1)^\top] - \sum_{k=1}^{K} p_s(k)\mu_s(k)\mu_s(k)^\top
$$

$$
+ \sum_{k=1}^{K} p_s(k)\mathbb{E}_{\boldsymbol{x} \in C_s(k)} \mathbb{E}_{\boldsymbol{x}_1,\boldsymbol{x}_2 \in \mathcal{A}(\boldsymbol{x})}[f(\boldsymbol{x}_1)(f(\boldsymbol{x}_2) - f(\boldsymbol{x}_1))^\top]
$$

$$
= \sum_{k=1}^{K} p_s(k)\mathbb{E}_{\boldsymbol{x} \in C_s(k)} \mathbb{E}_{\boldsymbol{x_1} \in \mathcal{A}(\boldsymbol{x})}[(f(\boldsymbol{x}_1) - \mu_s(k))(f(\boldsymbol{x}_1) - \mu_s(k))^\top]
$$

$$
+ \mathbb{E}_{\boldsymbol{x}} \mathbb{E}_{\boldsymbol{x}_1,\boldsymbol{x}_2 \in \mathcal{A}(\boldsymbol{x})}[f(\boldsymbol{x}_1)(f(\boldsymbol{x}_2) - f(\boldsymbol{x}_1))^\top], \tag{28}
$$

where the last equation is derived from

$$
\mathbb{E}_{\boldsymbol{x} \in C_s(k)} \mathbb{E}_{\boldsymbol{x_1} \in \mathcal{A}(\boldsymbol{x})}[f(\boldsymbol{x}_1)f(\boldsymbol{x}_1)^\top] - \mu_s(k)\mu_s(k)^\top
$$

$$
= \mathbb{E}_{\boldsymbol{x} \in C_s(k)} \mathbb{E}_{\boldsymbol{x_1} \in \mathcal{A}(\boldsymbol{x})}[f(\boldsymbol{x}_1)f(\boldsymbol{x}_1)^\top] + \mu_s(k)\mu_s(k)^\top - \left(\mathbb{E}_{\boldsymbol{x} \in C_s(k)} \mathbb{E}_{\boldsymbol{x_1} \in \mathcal{A}(\boldsymbol{x})}[f(\boldsymbol{x}_1)]\right)\mu_s(k)^\top
$$

$$
- \mu_s(k)\left(\mathbb{E}_{\boldsymbol{x} \in C_s(k)} \mathbb{E}_{\boldsymbol{x_1} \in \mathcal{A}(\boldsymbol{x})}[f(\boldsymbol{x}_1)]\right)^\top
$$

$$
= \mathbb{E}_{\boldsymbol{x} \in C_s(k)} \mathbb{E}_{\boldsymbol{x_1} \in \mathcal{A}(\boldsymbol{x})}[(f(\boldsymbol{x}_1) - \mu_s(k))(f(\boldsymbol{x}_1) - \mu_s(k))^\top].
$$

So its norm is

$$
\left\| \sum_{k=1}^{K} p_s(k)\mu_s(k)\mu_s(k)^\top - \mathbb{E}_{\boldsymbol{x}} \mathbb{E}_{\boldsymbol{x_1},\boldsymbol{x_2} \in \mathcal{A}(\boldsymbol{x})}[f(\boldsymbol{x}_1)f(\boldsymbol{x}_2)^\top] \right\|_F
$$

$$
\leq \sum_{k=1}^{K} p_s(k)\mathbb{E}_{\boldsymbol{x} \in C_s(k)} \mathbb{E}_{\boldsymbol{x_1} \in \mathcal{A}(\boldsymbol{x})}[\|(f(\boldsymbol{x}_1) - \mu_s(k))(f(\boldsymbol{x}_1) - \mu_s(k))^\top\|_F]
$$

$$
+ \mathbb{E}_{\boldsymbol{x}} \mathbb{E}_{\boldsymbol{x}_1,\boldsymbol{x}_2 \in \mathcal{A}(\boldsymbol{x})}[\|f(\boldsymbol{x}_1)(f(\boldsymbol{x}_2) - f(\boldsymbol{x}_1))^\top\|_F]
$$

$$
\leq \sum_{k=1}^{K} p_s(k) \mathbb{E}_{\boldsymbol{x} \in C_s(k)} \mathbb{E}_{\boldsymbol{x_1} \in \mathcal{A}(\boldsymbol{x})}\left[\|f(\boldsymbol{x}_1) - \mu_s(k)\|_2^2\right] + \mathbb{E}_{\boldsymbol{x}} \mathbb{E}_{\boldsymbol{x}_1,\boldsymbol{x}_2 \in \mathcal{A}(\boldsymbol{x})}\left[\|f(\boldsymbol{x}_1)\|_2 \|f(\boldsymbol{x}_2) - f(\boldsymbol{x}_1)\|_2\right]
$$

$$
\leq \sum_{k=1}^{K} p_s(k)\mathbb{E}_{\boldsymbol{x} \in C_s(k)} \mathbb{E}_{\boldsymbol{x_1} \in \mathcal{A}(\boldsymbol{x})}\left[\|f(\boldsymbol{x}_1) - \mu_s(k)\|_2^2\right]
$$

$$
+ \left[\mathbb{E}_{\boldsymbol{x}} \mathbb{E}_{\boldsymbol{x_1} \in \mathcal{A}(\boldsymbol{x})}\|f(\boldsymbol{x}_1)\|_2^2\right]^{\frac{1}{2}} \left[\mathbb{E}_{\boldsymbol{x}} \mathbb{E}_{\boldsymbol{x}_1,\boldsymbol{x}_2 \in \mathcal{A}(\boldsymbol{x})}\|f(\boldsymbol{x}_2) - f(\boldsymbol{x}_1)\|_2^2\right]^{\frac{1}{2}}
$$

$$
\leq \sum_{k=1}^{K} p_s(k)\mathbb{E}_{\boldsymbol{x} \in C_s(k)} \mathbb{E}_{\boldsymbol{x_1} \in \mathcal{A}(\boldsymbol{x})}\left[\|f(\boldsymbol{x}_1) - \mu_s(k)\|_2^2\right]
$$

$$
+ B_2\left[\varepsilon^2 + \mathbb{E}_{\boldsymbol{x}} \mathbb{E}_{\boldsymbol{x}_1,\boldsymbol{x}_2 \in \mathcal{A}(\boldsymbol{x})}\left[\|f(\boldsymbol{x}_2) - f(\boldsymbol{x}_1)\|_2^2 \mathbb{1}\{\boldsymbol{x} \notin S_s(\varepsilon, f)\}\right]\right]^{\frac{1}{2}}
$$

$$\left(\text{Review that } S_s(\varepsilon, f) := \{\boldsymbol{x} \in \cup_{k=1}^K C_s(k) : \sup_{\boldsymbol{x}_1, \boldsymbol{x}_2 \in \mathcal{A}(\boldsymbol{x})} \|f(\boldsymbol{x}_1) - f(\boldsymbol{x}_2)\|_2 \leq \varepsilon\}\right)$$

$$\leq \sum_{k=1}^K p_s(k) \mathbb{E}_{\boldsymbol{x} \in C_s(k)} \mathbb{E}_{\boldsymbol{x}_1 \in \mathcal{A}(\boldsymbol{x})} \left[\|f(\boldsymbol{x}_1) - \mu_s(k)\|_2^2\right] + B_2 \left[\varepsilon^2 + 4B_2^2 \mathbb{E}_{\boldsymbol{x}} \left[\mathbb{1}\{\boldsymbol{x} \notin S_s(\varepsilon, f)\}\right]\right]^{\frac{1}{2}}$$

$$= \sum_{k=1}^K p_s(k) \mathbb{E}_{\boldsymbol{x} \in C_s(k)} \mathbb{E}_{\boldsymbol{x}_1 \in \mathcal{A}(\boldsymbol{x})} \left[\|f(\boldsymbol{x}_1) - \mu_s(k)\|_2^2\right] + B_2 (\varepsilon^2 + 4B_2^2 R_s(\varepsilon, f))^{\frac{1}{2}}$$

$$\leq 4B_2^2 \sum_{k=1}^K p_s(k) \left[\left(1 - \sigma_s + \frac{\mathcal{K}\delta_s}{2B_2} + \frac{\varepsilon}{B_2} + \frac{R_s(\varepsilon, f)}{p_s(k)}\right)^2 + \left(1 - \sigma_s + \frac{R_s(\varepsilon, f)}{p_s(k)}\right)\right]$$

$$\quad + B_2 (\varepsilon^2 + 4B_2^2 R_s(\varepsilon, f))^{\frac{1}{2}} \qquad\qquad\qquad\qquad\qquad\qquad\qquad \text{(Lemma B.3)}$$

$$= 4B_2^2 \left[\left(1 - \sigma_s + \frac{\mathcal{K}\delta_s + 2\varepsilon}{2B_2}\right)^2 + (1 - \sigma_s) + K R_s(\varepsilon, f)\left(3 - 2\sigma_s + \frac{\mathcal{K}\delta_s + 2\varepsilon}{B_2}\right)\right.$$

$$\quad \left. + R_s^2(\varepsilon, f)\left(\sum_{k=1}^K \frac{1}{p_s(k)}\right)\right] + B_2 (\varepsilon^2 + 4B_2^2 R_s(\varepsilon, f))^{\frac{1}{2}}$$

If we define $\varphi(\sigma_s, \delta_s, \varepsilon, f) := 4B_2^2 \left[\left(1 - \sigma_s + \frac{\mathcal{K}\delta_s + 2\varepsilon}{2B_2}\right)^2 + (1 - \sigma_s) + K R_s(\varepsilon, f)\left(3 - 2\sigma_s + \frac{\mathcal{K}\delta_s + 2\varepsilon}{B_2}\right) + R_s^2(\varepsilon, f)\left(\sum_{k=1}^K \frac{1}{p_s(k)}\right)\right] + B_2 (\varepsilon^2 + 4B_2^2 R_s(\varepsilon, f))^{\frac{1}{2}}$, above derivation implies

$$\left\|\sum_{k=1}^K p_s(k)\mu_s(k)\mu_s(k)^\top - \mathbb{E}_{\boldsymbol{x}}\mathbb{E}_{\boldsymbol{x}_1, \boldsymbol{x}_2 \in \mathcal{A}(\boldsymbol{x})}[f(\boldsymbol{x}_1)f(\boldsymbol{x}_2)^\top]\right\|_F \leq \varphi(\sigma_s, \delta_s, \varepsilon, f). \qquad (29)$$

Besides that, Note that

$$\mathcal{L}_{\mathrm{div}}(f) = \sup_{G \in \mathcal{G}(f)} \langle \mathbb{E}_{\boldsymbol{x}}\mathbb{E}_{\boldsymbol{x}_1, \boldsymbol{x}_2 \in \mathcal{A}(\boldsymbol{x})}[f(\boldsymbol{x}_1)f(\boldsymbol{x}_2)^\top] - I_{d^*}, G \rangle_F$$

$$= \left\|\mathbb{E}_{\boldsymbol{x}}\mathbb{E}_{\boldsymbol{x}_1, \boldsymbol{x}_2 \in \mathcal{A}(\boldsymbol{x})}[f(\boldsymbol{x}_1)f(\boldsymbol{x}_2)^\top] - I_{d^*}\right\|_F^2, \qquad (30)$$

which is from the facts that $\mathbb{E}_{\boldsymbol{x}}\mathbb{E}_{\boldsymbol{x}_1, \boldsymbol{x}_2 \in \mathcal{A}(\boldsymbol{x})}[f(\boldsymbol{x}_1)f(\boldsymbol{x}_2)^\top] - I_{d^*} \in \mathcal{G}(f)$ and

$$\langle \mathbb{E}_{\boldsymbol{x}}\mathbb{E}_{\boldsymbol{x}_1, \boldsymbol{x}_2 \in \mathcal{A}(\boldsymbol{x})}[f(\boldsymbol{x}_1)f(\boldsymbol{x}_2)^\top] - I_{d^*}, G \rangle_F \leq \left\|\mathbb{E}_{\boldsymbol{x}}\mathbb{E}_{\boldsymbol{x}_1, \boldsymbol{x}_2 \in \mathcal{A}(\boldsymbol{x})}[f(\boldsymbol{x}_1)f(\boldsymbol{x}_2)^\top] - I_{d^*}\right\|_F \cdot \|G\|_F$$

Combining (27), (28), (29), (30) yields for any $i \neq j$

$$(\mu_s(i)^\top \mu_s(j))^2 \leq \frac{2}{p_s(i)p_s(j)}\left(\mathcal{L}_{\mathrm{div}}(f) + \varphi(\sigma_s, \delta_s, \varepsilon, f)\right),$$

which implies that

$$\max_{i \neq j} |\mu_s(i)^\top \mu_s(j)| \leq \sqrt{\frac{2}{\min_{i \neq j} p_s(i)p_s(j)}\left(\mathcal{L}_{\mathrm{div}}(f) + \varphi(\sigma_s, \delta_s, \varepsilon, f)\right)}.$$

So we can get what we desired according to (26)

$$\max_{i \neq j} |\mu_t(i)^\top \mu_t(j)| \leq \sqrt{\frac{2}{\min_{i \neq j} p_s(i)p_s(j)}\left(\mathcal{L}_{\mathrm{div}}(f) + \varphi(\sigma_s, \delta_s, \varepsilon, f)\right)} + 2\sqrt{d^*}B_2 M\mathcal{K}\rho.$$

$$\square$$

### B.2.4 CONNECTION BETWEEN PRETRAINING AND DOWNSTREAM TASK

Following theorem reveals that minimaxing our loss may achieve a small misclassification rate in downstream task.

**Theorem B.1.** *Given a $(\sigma_s, \sigma_t, \delta_s, \delta_t)$-augmentation, for any $\varepsilon > 0$, if $\Theta(\sigma_t, \delta_t, \varepsilon, \hat{f}_{n_s}) > 0$, then*

*with probability at least* $1 - \dfrac{\sqrt{\frac{2}{\min_{i \neq j} p_s(i) p_s(j)}\left(\frac{1}{\lambda}\mathbb{E}_{D_s}[\sup_{G \in \mathcal{G}(\hat{f}_{n_s})} \mathcal{L}(\hat{f}_{n_s}, G)] + \psi(\sigma_s, \delta_s, \varepsilon, \hat{f}_{n_s})\right) + 2\sqrt{d^*} B_2 M \mathcal{K} \rho}}{B_2^2 \Theta(\sigma_t, \delta_t, \varepsilon, \hat{f}_{n_s})}$,

*we have*

$$\mathbb{E}_{D_s}[\mathrm{Err}(Q_{\hat{f}_{n_s}})] \leq (1 - \sigma_t) + \frac{m^2}{\varepsilon}\sqrt{\mathbb{E}_{D_s}[\sup_{G \in \mathcal{G}(\hat{f}_{n_s})} \mathcal{L}(\hat{f}_{n_s}, G)] + 8 B_2 d^* M \mathcal{K} \rho + 4 B_2^2 d^* K \eta},$$

*where*

$$\psi(\sigma_s, \delta_s, \varepsilon, \hat{f}_{n_s}) := B_2 \Big( \varepsilon^2 + 4 B_2^2 \frac{m^2}{\varepsilon}\sqrt{\mathbb{E}_{D_s}[\sup_{G \in \mathcal{G}(\hat{f}_{n_s})} \mathcal{L}(\hat{f}_{n_s}, G)]} \Big)^{\frac{1}{2}} + 4 B_2^2 \Big[ \Big( 1 - \sigma_s + \frac{\mathcal{K}\delta_s + 2\varepsilon}{2 B_2} \Big)^2$$

$$+ (1 - \sigma_s) + \frac{K m^2}{\varepsilon}\sqrt{\mathbb{E}_{D_s}[\sup_{G \in \mathcal{G}(\hat{f}_{n_s})} \mathcal{L}(\hat{f}_{n_s}, G)]}\Big( 3 - 2\sigma_s + \frac{\mathcal{K}\delta_s + 2\varepsilon}{B_2} \Big)$$

$$+ \frac{m^4}{\varepsilon^2}\mathbb{E}_{D_s}[\sup_{G \in \mathcal{G}(\hat{f}_{n_s})} \mathcal{L}(\hat{f}_{n_s}, G)]\Big( \sum_{k=1}^{K} \frac{1}{p_s(k)} \Big) \Big],$$

$\Gamma_{\min}(\sigma_t, \delta_t, \varepsilon, f) = \Big( \sigma_t - \frac{R_t(\varepsilon, f)}{\min_i p_t(i)} \Big)\Big( 1 + \Big( \frac{B_1}{B_2} \Big)^2 - \frac{\mathcal{K}\delta_t}{B_2} - \frac{2\varepsilon}{B_2} \Big) - 1$, $\Delta_{\hat{\mu}_t} = 1 - \frac{\min_{k \in [K]} \|\hat{\mu}_t(k)\|^2}{B_2^2}$,
$R_t(\varepsilon, f) = P_t\big( z \in \cup_{k=1}^{K} \widetilde{C}_t(k) : \sup_{z_1, z_2 \in \mathcal{A}(z)} \|f(z_1) - f(z_2)\| > \varepsilon \big)$ *and* $\Theta(\sigma_t, \delta_t, \varepsilon, \hat{f}_{n_s}) =$
$\Gamma_{\min}(\sigma_t, \delta_t, \varepsilon, \hat{f}_{n_s}) - \sqrt{2 - 2\Gamma_{\min}(\sigma_t, \delta_t, \varepsilon, \hat{f}_{n_s})} - \frac{\Delta_{\hat{\mu}_t}}{2} - \frac{2\max_{k \in [K]} \|\hat{\mu}_t(k) - \mu_t(k)\|_2}{B_2}$.

*In addition, the following inequalities always hold*

$$\mathbb{E}_{D_s}[R_t^2(\varepsilon, \hat{f}_{n_s})] \leq \frac{m^4}{\varepsilon^2}\Big( \mathbb{E}_{D_s}[\sup_{G \in \mathcal{G}(\hat{f}_{n_s})} \mathcal{L}(\hat{f}_{n_s}, G)] + 8 B_2 d^* M \mathcal{K} \rho + 4 B_2^2 d^* K \eta \Big).$$

*Proof.* Note the facts that $\sup_{G \in \mathcal{G}(f)} \mathcal{L}(f, G) \geq \max\{\mathcal{L}_{\mathrm{align}}(f), \lambda \mathcal{L}_{\mathrm{div}}(f)\}$, $B_1 \leq \|\hat{f}_{n_s}\|_2 \leq B_2$
and $\mathcal{K}$-Lipschitz continuity of $\hat{f}_{n_s}$, apply Lemma B.4 to $\hat{f}_{n_s}$ to obtain

$$R_s^2(\varepsilon, \hat{f}_{n_s}) \leq \frac{m^4}{\varepsilon^2}\sup_{G \in \mathcal{G}(\hat{f}_{n_s})} \mathcal{L}(\hat{f}_{n_s}) \tag{31}$$

$$R_t^2(\varepsilon, \hat{f}_{n_s}) \leq \frac{m^4}{\varepsilon^2}\sup_{G \in \mathcal{G}(\hat{f}_{n_s})} \mathcal{L}(\hat{f}_{n_s}, G) + \frac{8 m^4}{\varepsilon^2} B_2 d^* M \mathcal{K} \rho + \frac{4 m^4}{\varepsilon^2} B_2^2 d^* K \eta \tag{32}$$

and

$$\max_{i \neq j} |\mu_t(i)^\top \mu_t(j)| \leq \sqrt{\frac{2}{\min_{i \neq j} p_s(i) p_s(j)}\Big( \frac{1}{\lambda}\sup_{G \in \mathcal{G}(\hat{f}_{n_s})} \mathcal{L}(\hat{f}_{n_s}, G) + \varphi(\sigma_s, \delta_s, \varepsilon, \hat{f}_{n_s}) \Big)}$$
$$+ 2\sqrt{d^*} B_2 M \mathcal{K} \rho \tag{33}$$

Take expectation w.r.t $D_s$ in the both side of (31), (32), (33) and apply Jensen inequality to yield

$$\mathbb{E}_{D_s}[R_s^2(\varepsilon, \hat{f}_{n_s})] \leq \frac{m^4}{\varepsilon^2}\mathbb{E}_{D_s}[\sup_{G \in \mathcal{G}(\hat{f}_{n_s})} \mathcal{L}(\hat{f}_{n_s}, G)]$$

$$\mathbb{E}_{D_s}[R_t^2(\varepsilon, \hat{f}_{n_s})] \leq \frac{m^4}{\varepsilon^2}\mathbb{E}_{D_s}[\sup_{G \in \mathcal{G}(\hat{f}_{n_s})} \mathcal{L}(\hat{f}_{n_s}, G)] + \frac{8 m^4}{\varepsilon^2} B_2 d^* M \mathcal{K} \rho + \frac{4 m^4}{\varepsilon^2} B_2^2 d^* K \eta$$

$$\mathbb{E}_{D_s}[\max_{i \neq j} |\mu_t(i)^\top \mu_t(j)|] \leq \sqrt{\frac{2}{\min_{i \neq j} p_s(i) p_s(j)}\Big( \frac{1}{\lambda}\mathbb{E}_{D_s}[\sup_{G \in \mathcal{G}(\hat{f}_{n_s})} \mathcal{L}(\hat{f}_{n_s}, G)] + \mathbb{E}_{D_s}[\varphi(\sigma_s, \delta_s, \varepsilon, \hat{f}_{n_s})] \Big)}$$
$$+ 2\sqrt{d^*} B_2 M \mathcal{K} \rho$$

where $\mathbb{E}_{D_s}[\varphi(\sigma_s, \delta_s, \varepsilon, \hat{f}_{n_s})] = 4B_2^2\Big[\Big(1 - \sigma_s + \frac{\mathcal{K}\delta_s + 2\varepsilon}{2B_2}\Big)^2 + (1 - \sigma_s) + K\mathbb{E}_{D_s}[R_s(\varepsilon, \hat{f}_{n_s})]\Big(3 - 2\sigma_s + \frac{\mathcal{K}\delta_s + 2\varepsilon}{B_2}\Big) + \mathbb{E}_{D_s}[R_s^2(\varepsilon, \hat{f}_{n_s})]\Big(\sum_{k=1}^{K} \frac{1}{p_s(k)}\Big)\Big] + B_2\mathbb{E}_{D_s}[(\varepsilon^2 + 4B_2^2 R_s(\varepsilon, \hat{f}_{n_s}))^{\frac{1}{2}}].$

Therefore, by Jensen inequality, we have

$$\mathbb{E}_{D_s}[\varphi(\sigma_s, \delta_s, \varepsilon, R_s(\varepsilon, \hat{f}_{n_s}))]$$

$$\leq 4B_2^2\Big[\Big(1 - \sigma_s + \frac{\mathcal{K}\delta_s + 2\varepsilon}{2B_2}\Big)^2 + (1 - \sigma_s) + K\mathbb{E}_{D_s}[R_s(\varepsilon, \hat{f}_{n_s})]\Big(3 - 2\sigma_s + \frac{\mathcal{K}\delta_s + 2\varepsilon}{B_2}\Big)$$

$$+ \mathbb{E}_{D_s}[R_s^2(\varepsilon, \hat{f}_{n_s})]\Big(\sum_{k=1}^{K} \frac{1}{p_s(k)}\Big)\Big] + B_2(\varepsilon^2 + 4B_2^2\mathbb{E}_{D_s}[R_s(\varepsilon, \hat{f}_{n_s})])^{\frac{1}{2}}$$

$$\leq 4B_2^2\Big[\Big(1 - \sigma_s + \frac{\mathcal{K}\delta_s + 2\varepsilon}{2B_2}\Big)^2 + (1 - \sigma_s) + \frac{Km^2}{\varepsilon}\sqrt{\mathbb{E}_{D_s}[\sup_{G \in \mathcal{G}(\hat{f}_{n_s})} \mathcal{L}(\hat{f}_{n_s}, G)]}\Big(3 - 2\sigma_s +$$

$$\frac{\mathcal{K}\delta_s + 2\varepsilon}{B_2}\Big) + \frac{m^4}{\varepsilon^2}\mathbb{E}_{D_s}[\sup_{G \in \mathcal{G}(\hat{f}_{n_s})} \mathcal{L}(\hat{f}_{n_s}, G)]\Big(\sum_{k=1}^{K} \frac{1}{p_s(k)}\Big)\Big]$$

$$+ B_2\Big(\varepsilon^2 + \frac{4B_2^2 m^2}{\varepsilon}\sqrt{\mathbb{E}_{D_s}[\sup_{G \in \mathcal{G}(\hat{f}_{n_s})} \mathcal{L}(\hat{f}_{n_s}, G)]}\Big)^{\frac{1}{2}}$$

$$:= \psi(\sigma_s, \delta_s, \varepsilon, \hat{f}_{n_s}).$$

Recall Lemma B.1 reveals that we can obtain

$$\text{Err}(Q_{\hat{f}_{n_s}}) \leq (1 - \sigma_t) + R_t(\varepsilon, \hat{f}_{n_s})$$

if $\max_{i \neq j} |(\mu_t(i))^\top \mu_t(j)| < B_2^2\Theta(\sigma_t, \delta_t, \varepsilon, \hat{f}_{n_s}).$

So that if $\Theta(\sigma_t, \delta_t, \varepsilon, \hat{f}_{n_s}) > 0$, apply Markov inequality to know with probability at least

$$1 - \frac{\sqrt{\frac{2}{\min_{i \neq j} p_s(i) p_s(j)}\Big(\frac{1}{\lambda}\mathbb{E}_{D_s}[\sup_{G \in \mathcal{G}(\hat{f}_{n_s})} \mathcal{L}(\hat{f}_{n_s}, G)] + \psi(\sigma_s, \delta_s, \varepsilon, \hat{f}_{n_s})\Big)} + 2\sqrt{d^*}B_2 M\mathcal{K}\rho}{B_2^2\Theta(\sigma_t, \delta_t, \varepsilon, \hat{f}_{n_s})},$$

we have

$$\max_{i \neq j} |\mu_t(i)^\top \mu_t(j)| < B_2^2\Theta(\sigma_t, \delta_t, \varepsilon, \hat{f}_{n_s}),$$

so that we can get what we desired.

$$\mathbb{E}_{D_s}[\text{Err}(Q_{\hat{f}_{n_s}})] \leq (1 - \sigma_t) + R_t(\varepsilon, \hat{f}_{n_s})$$

$$\leq (1 - \sigma_t) + \frac{m^2}{\varepsilon}\sqrt{\mathbb{E}_{D_s}[\sup_{G \in \mathcal{G}(\hat{f}_{n_s})} \mathcal{L}(\hat{f}_{n_s}, G)] + 8B_2 d^* M\mathcal{K}\rho + 4B_2^2 d^* K\eta},$$

where the last inequality is due to (32). $\qquad \square$

### B.2.5 PRELIMINARIES FOR ERROR ANALYSIS

To prove Theorem 4.2 based on Theorem B.1, we need to first introduce some related definitions and conclusions, which are going to be used in subsequent contents.

Recall that for any $\boldsymbol{x} \in \mathcal{X}_s, \boldsymbol{x}_1, \boldsymbol{x}_2 \overset{\text{i.i.d.}}{\sim} A(\boldsymbol{x}), \tilde{\boldsymbol{x}} = (\boldsymbol{x}_1, \boldsymbol{x}_2) \in \mathbb{R}^{2d^*}$. If we define $\ell(\tilde{\boldsymbol{x}}, G) := \|f(\boldsymbol{x}_1) - f(\boldsymbol{x}_2)\|_2^2 + \lambda\langle f(\boldsymbol{x}_1)f(\boldsymbol{x}_2)^\top - I_{d^*}, G\rangle_F$, then our loss function at sample level can be rewritten as

$$\widehat{\mathcal{L}}(f, G) := \frac{1}{n_s}\sum_{i=1}^{n_s}\Big[\|f(\boldsymbol{x}_1^{(i)}) - f(\boldsymbol{x}_2^{(i)})\|_2^2 + \lambda\langle f(\boldsymbol{x}_1^{(i)})f(\boldsymbol{x}_2^{(i)})^\top - I_{d^*}, G\rangle_F\Big] = \frac{1}{n_s}\sum_{i=1}^{n_s}\ell(\tilde{\boldsymbol{x}}^{(i)}, G),$$

furthermore, denote $\mathcal{G}_1 := \{G \in \mathbb{R}^{d^* \times d^*} : \|G\|_F \leq B_2^2 + \sqrt{d^*}\}$. It is obvious that both $\mathcal{G}(f)$ for any $f : \|f\|_2 \leq B_2$ and $\widehat{\mathcal{G}}(f)$ for any $f \in \mathcal{NN}_{d,d^*}(W, L, \mathcal{K}, B_1, B_2)$ are contained in $\mathcal{G}_1$. Apart from that, following Proposition B.1 reveals that $\ell(\boldsymbol{u}, G)$ is a Lipschitz function on the domain $\{\boldsymbol{u} \in \mathbb{R}^{2d^*} : \|\boldsymbol{u}\|_2 \leq \sqrt{2}B_2\} \times \mathcal{G}_1 \subseteq \mathbb{R}^{2d^* + (d^*)^2}$.

**Proposition B.1.** $\ell$ is a Lipschitz function on the domain $\{\boldsymbol{u} \in \mathbb{R}^{2d^*} : \|\boldsymbol{u}\|_2 \leq \sqrt{2}B_2\} \times \mathcal{G}_1$.

*Proof.* At first step, we will prove $\|\ell(\cdot, G)\|_{\text{Lip}} < \infty$ for any fixed $G \in \mathcal{G}_1$. To this end, denote $\boldsymbol{u} = (\boldsymbol{u}_1, \boldsymbol{u}_2)$, where $\boldsymbol{u}_1, \boldsymbol{u}_2 \in \mathbb{R}^{d^*}$, we firstly show $J(\boldsymbol{u}) = \|\boldsymbol{u}_1 - \boldsymbol{u}_2\|_2^2$ is Lipschtiz function. let $g(\boldsymbol{u}) := \boldsymbol{u}_1 - \boldsymbol{u}_2$, then

$$
\begin{aligned}
\|g(\boldsymbol{u}_1, \boldsymbol{u}_2) - g(\boldsymbol{v}_1, \boldsymbol{v}_2)\|_2^2 &= \|\boldsymbol{u}_1 - \boldsymbol{u}_2 - \boldsymbol{v}_1 + \boldsymbol{v}_2\|_2^2 \\
&\leq \left(\|\boldsymbol{u}_1 - \boldsymbol{v}_1\|_2 + \|\boldsymbol{u}_2 - \boldsymbol{v}_2\|_2\right)^2 \\
&= \|\boldsymbol{u}_1 - \boldsymbol{v}_1\|_2^2 + \|\boldsymbol{u}_2 - \boldsymbol{v}_2\|_2^2 + 2\|\boldsymbol{u}_1 - \boldsymbol{v}_1\|_2\|\boldsymbol{u}_2 - \boldsymbol{v}_2\|_2 \\
&\leq 2(\|\boldsymbol{u}_1 - \boldsymbol{v}_1\|_2^2 + \|\boldsymbol{u}_2 - \boldsymbol{v}_2\|_2^2) \\
&= 2\|(\boldsymbol{u}_1, \boldsymbol{u}_2) - (\boldsymbol{v}_1, \boldsymbol{v}_2)\|_2^2,
\end{aligned}
$$

which implies that $g(\boldsymbol{u}) \in \text{Lip}(\sqrt{2})$. Apart from that, $g$ also possess the property that $\|g(\boldsymbol{u})\|_2 = \|\boldsymbol{u}_1 - \boldsymbol{u}_2\|_2 \leq \|\boldsymbol{u}_1\|_2 + \|\boldsymbol{u}_2\|_2 \leq 2\|\boldsymbol{u}\|_2 \leq 2\sqrt{2}B_2$. Moreover, let $h(\boldsymbol{v}) := \|\boldsymbol{v}\|_2^2$, we know that

$$
\left\|\frac{\partial h}{\partial \boldsymbol{v}}(g(\boldsymbol{u}))\right\|_2 = 2\|g(\boldsymbol{u})\|_2 \leq 4\sqrt{2}B_2.
$$

Therefore, $J(\boldsymbol{u}) = h(g(\boldsymbol{u})) = \|\boldsymbol{u}_1 - \boldsymbol{u}_2\|_2^2 \in \text{Lip}(8B_2)$

To show $Q(\boldsymbol{u}) = \langle \boldsymbol{u}_1 \boldsymbol{u}_2^\top - I_{d^*}, G \rangle_F$ is also a Lipschtiz function. Define $\tilde{g}(\boldsymbol{u}) := \boldsymbol{u}_1 \boldsymbol{u}_2^\top$, we know that

$$
\begin{aligned}
\|\tilde{g}(\boldsymbol{u}) - \tilde{g}(\boldsymbol{v})\|_F &= \|\boldsymbol{u}_1 \boldsymbol{u}_2^\top - \boldsymbol{v}_1 \boldsymbol{v}_2^\top\|_F \\
&= \|\boldsymbol{u}_1 \boldsymbol{u}_2^\top - \boldsymbol{u}_1 \boldsymbol{v}_2^\top + \boldsymbol{u}_1 \boldsymbol{v}_2^\top - \boldsymbol{v}_1 \boldsymbol{v}_2^\top\|_F \\
&= \|\boldsymbol{u}_1 (\boldsymbol{u}_2 - \boldsymbol{v}_2)^\top + (\boldsymbol{u}_1 - \boldsymbol{v}_1)\boldsymbol{v}_2^\top\|_F \\
&\leq \|\boldsymbol{u}_1\|_F \|\boldsymbol{u}_2 - \boldsymbol{v}_2\|_F + \|\boldsymbol{u}_1 - \boldsymbol{v}_1\|_F \|\boldsymbol{v}_2\|_F \\
&\leq (\|\boldsymbol{u}_1\|_2 + \|\boldsymbol{v}_2\|_2)\|\boldsymbol{u} - \boldsymbol{v}\|_2 \\
&\leq 2\sqrt{2}B_2 \|\boldsymbol{u} - \boldsymbol{v}\|_2.
\end{aligned}
$$

Furthermore, denote $\tilde{h}(A) := \langle A - I_{d^*}, G \rangle_F$, then $\|\nabla \tilde{h}(A)\|_F = \|G\|_F \leq B_2^2 + \sqrt{d^*}$. So that $Q(\boldsymbol{u}) = \tilde{h}(\tilde{g}(\boldsymbol{u})) \in \text{Lip}(2\sqrt{2}B_2(B_2^2 + \sqrt{d^*}))$.

Combining above conclusions knows that for any $G \in \mathcal{G}_1$, we have $\|\ell(\cdot, G)\|_{\text{Lip}} < \infty$ on the domain $\{\boldsymbol{u} : \|\boldsymbol{u}\|_2 \leq \sqrt{2}B_2\}$.

Next, fixed $\boldsymbol{u} \in \mathbb{R}^{2d^*}$ such that $\|\boldsymbol{u}\|_2 \leq \sqrt{2}B_2$, we have

$$
|\ell(\boldsymbol{u}, G_1) - \ell(\boldsymbol{u}, G_2)| = |\langle \boldsymbol{u}, G_1 - G_2 \rangle_F| \leq \|\boldsymbol{u}\|_2 \|G_1 - G_2\|_F = \sqrt{2}B_2 \|G_1 - G_2\|_F,
$$

which implies that $\ell(\boldsymbol{u}, \cdot) \in \text{Lip}(\sqrt{2}B_2)$.

Finally, note that

$$
\begin{aligned}
|\ell(\boldsymbol{u}_1, G_1) - \ell(\boldsymbol{u}_2, G_2)|^2 &\leq (|\ell(\boldsymbol{u}_1, G_1) - \ell(\boldsymbol{u}_2, G_1)| + |\ell(\boldsymbol{u}_2, G_1) - \ell(\boldsymbol{u}_2, G_2)|)^2 \\
&\leq \left((\sqrt{2} + 2\sqrt{2}B_2(B_2^2 + \sqrt{d^*}))\|\boldsymbol{u}_1 - \boldsymbol{u}_2\|_2 + \sqrt{2}B_2\|G_1 - G_2\|_F\right)^2 \\
&\leq 2(\sqrt{2} + 2\sqrt{2}B_2(B_2^2 + \sqrt{d^*}))^2 \|\boldsymbol{u}_1 - \boldsymbol{u}_2\|_2^2 + 4B_2^2 \|G_1 - G_2\|_F^2 \\
&\leq C\|\text{vec}(\boldsymbol{u}_1, G_1) - \text{vec}(\boldsymbol{u}_2, G_2)\|_2^2
\end{aligned}
$$

where $C$ is a constant s.t $C \geq \max\{2(\sqrt{2} + 2\sqrt{2}B_2(B_2^2 + \sqrt{d^*}))^2, 4B_2^2\}$, which yields what we desired. $\square$

Table 2: Lipschitz constant of $\ell$ with respect to each component

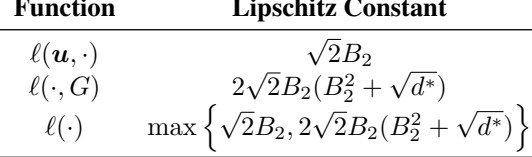

| Function | Lipschitz Constant |
|---|---|
| $\ell(\boldsymbol{u}, \cdot)$ | $\sqrt{2}B_2$ |
| $\ell(\cdot, G)$ | $2\sqrt{2}B_2(B_2^2 + \sqrt{d^*})$ |
| $\ell(\cdot)$ | $\max\left\{\sqrt{2}B_2, 2\sqrt{2}B_2(B_2^2 + \sqrt{d^*})\right\}$ |

We summary the Lipschitz constants of $\ell(\boldsymbol{u}, G)$ with respect to both $\boldsymbol{u} \in \{\boldsymbol{u} \in \mathbb{R}^{2d^*} : \|\boldsymbol{u}\|_2 \leq \sqrt{2}B_2\}$ and $G \in \mathcal{G}_1$ in Table 2.

**Definition B.1** (Rademacher complexity). Given a set $S \subseteq \mathbb{R}^n$, the Rademacher complexity of $S$ is denoted by

$$\mathcal{R}_n(S) := \mathbb{E}_\xi \Big[ \sup_{(s_1, \ldots, s_n) \in S} \frac{1}{n} \sum_{i=1}^n \xi_i s_i \Big],$$

where $\{\xi_i\}_{i \in [n]}$ is a sequence of i.i.d Radmacher random variables which take the values $1$ and $-1$ with equal probability $1/2$.

Following vector-contraction principle of Rademacher complexity will be used in later contents.

**Lemma B.5** (Vector-contraction principle). *Let $\mathcal{X}$ be any set, $(x_1, \ldots, x_n) \in \mathcal{X}^n$, let $F$ be a class of functions $f : \mathcal{X} \to \ell_2$ and let $h_i : \ell_2 \to \mathbb{R}$ have Lipschitz norm $L$. Then*

$$\mathbb{E} \sup_{f \in F} \Big| \sum_i \epsilon_i h_i(f(x_i)) \Big| \leq 2\sqrt{2}L\mathbb{E} \sup_{f \in F} \Big| \sum_{i,k} \varepsilon_{ik} f_k(x_i) \Big|,$$

*where $\epsilon_{ik}$ is an independent doubly indexed Rademacher sequence and $f_k(x_i)$ is the $k$-th component of $f(x_i)$.*

*Proof.* Combining Maurer (2016) and Theorem 3.2.1 of Giné & Nickl (2016) obtains the desired result. $\square$

Recall $\mathcal{NN}_{d_1,d_2}(W, L, \mathcal{K}) := \{\phi_\theta(\boldsymbol{x}) = \boldsymbol{A}_L\sigma(\boldsymbol{A}_{L-1}\sigma(\cdots\sigma(\boldsymbol{A}_0\boldsymbol{x})) : \kappa(\theta) \leq \mathcal{K}\}$, which is defined in (14). The second lemma we will employed is related to the upper bound for Rademacher complexity of hypothesis space consisting of norm-constrained neural networks, which was provided by Golowich et al. (2018).

**Lemma B.6** (Theorem 3.2 of Golowich et al. (2018)). *$\forall n \in \mathbb{N}^+, \forall \boldsymbol{x}_1, \ldots, \boldsymbol{x}_n \in [-B, B]^d$ with $B \geq 1, S := \{(\phi(\boldsymbol{x}_1), \ldots, \phi(\boldsymbol{x}_n)) : \phi \in \mathcal{NN}_{d,1}(W, L, \mathcal{K})\} \subseteq \mathbb{R}^n$, then*

$$\mathcal{R}_n(S) \leq \frac{1}{n}\mathcal{K}\sqrt{2(L + 2 + \log(d+1))} \max_{1 \leq j \leq d+1} \sqrt{\sum_{i=1}^n x_{i,j}^2} \leq \frac{B\mathcal{K}\sqrt{2(L + 2 + \log(d+1))}}{\sqrt{n}},$$

*where $x_{i,j}$ is the $j$-th coordinate of the vector $(\boldsymbol{x_i}^\top, 1)^\top \in \mathbb{R}^{d+1}$.*

**Definition B.2** (Covering number). $\forall n \in \mathbb{N}^+$, Fix $\mathcal{S} \subseteq \mathbb{R}^n$ and $\varrho > 0$, the set $\mathcal{N}$ is called an $\varrho$-net of $\mathcal{S}$ with respect to a norm $\|\cdot\|$ on $\mathbb{R}^n$, if $\mathcal{N} \subseteq \mathcal{S}$ and for any $\boldsymbol{u} \in \mathcal{S}$, there exists $\boldsymbol{v} \in \mathcal{N}$ such that $\|\boldsymbol{u} - \boldsymbol{v}\| \leq \varrho$. The covering number of $\mathcal{S}$ is defined as

$$\mathcal{N}(\mathcal{S}, \|\cdot\|, \varrho) := \min\{|\mathcal{Q}| : \mathcal{Q} \text{ is an } \varrho\text{-cover of } \mathcal{S}\}$$

where $|\mathcal{Q}|$ is the cardinality of the set $\mathcal{Q}$.

According to the Corollary 4.2.13 of Vershynin (2018), $|\mathcal{N}(\mathcal{B}_2, \|\cdot\|_2, \varrho)|$, which is the the covering number of 2-norm unit ball in $\mathbb{R}^{(d^*)^2}$, can be bounded by $(\frac{3}{\varrho})^{(d^*)^2}$, so that if we denote $\mathcal{N}_{\mathcal{G}_1}(\varrho)$ is a cover of $\mathcal{G}_1$ with radius $\varrho$ whose cardinality $|\mathcal{N}_{\mathcal{G}_1}(\varrho)|$ is equal to the covering number of $\mathcal{G}_1$, then $|\mathcal{N}_{\mathcal{G}_1}(\varrho)| \leq (\frac{3}{(B_2^2 + \sqrt{d^*})\varrho})^{(d^*)^2}$.

Apart from that, we need to employ following finite maximum inequality, which is stated in Lemma 2.3.4 of Giné & Nickl (2016), in later deduction.

**Lemma B.7** (Finite maximum inequality). *For any $N \geq 1$, if $X_i, i \leq N$, are sub-Gaussian random variables admitting constants $\sigma_i$, then*

$$\mathbb{E} \max_{i \leq N} |X_i| \leq \sqrt{2 \log 2N} \max_{i \leq N} \sigma_i$$

**Definition B.3** (Excess risk). The difference between $\sup\limits_{G \in \mathcal{G}(\hat{f}_{n_s})} \mathcal{L}(\hat{f}_{n_s}, G)$ and $\sup\limits_{G \in \mathcal{G}(f^*)} \mathcal{L}(f^*, G)$ is called excess risk, i.e.,

$$\mathcal{E}(\hat{f}_{n_s}) = \sup_{G \in \mathcal{G}(\hat{f}_{n_s})} \mathcal{L}(\hat{f}_{n_s}, G) - \sup_{G \in \mathcal{G}(f^*)} \mathcal{L}(f^*, G).$$

### B.2.6 DEAL WITH $\sup_{G \in \mathcal{G}(f^*)} \mathcal{L}(f^*)$

We aim to claim $\sup_{G \in \mathcal{G}(f^*)} \mathcal{L}(f^*) = 0$ in two step. At first, we assert that if there exists a measurable map $f$ satisfying $\Sigma = \mathbb{E}_{\boldsymbol{x} \sim P_s}[f(\boldsymbol{x})f(\boldsymbol{x})^\top]$ be positive definite, then we can conduct some minor rectification on it to get $\tilde{f}$ such that $\sup_{G \in \mathcal{G}(\tilde{f})} \mathcal{L}(\tilde{f}) = 0$. At the second step, we are going to show the required $f$ does exist under Assumption 4.5 and the rectification $\tilde{f}$ also fulfill the requirement that $B_1 \leq \|\tilde{f}\|_2 \leq B_2$, which implies that $\sup_{G \in \mathcal{G}(f^*)} \mathcal{L}(f^*) = 0$ as the definition of $f^*$ implies $\sup_{G \in \mathcal{G}(f^*)} \mathcal{L}(f^*) \leq \sup_{G \in \mathcal{G}(\tilde{f})} \mathcal{L}(\tilde{f})$.

Our final target is to result in a measurable map $f$, s.t $B_1 \leq \|f\|_2 \leq B_2$ and $\sup_{f \in \mathcal{G}(f)} \mathcal{L}(f) = 0$, it suffices to find a $f : B_1 \leq \|f\|_2 \leq B_2$ satisfying both $\mathcal{L}_{\text{align}}(f) = 0$ and $\left\| \mathbb{E}_{\boldsymbol{x}} \mathbb{E}_{\boldsymbol{x_1}, \boldsymbol{x_2} \in \mathcal{A}(\boldsymbol{x})}[f(\boldsymbol{x}_1)f(\boldsymbol{x}_2)^\top] - I_{d^*} \right\|_F = 0$. Note that

$$\left\| \mathbb{E}_{\boldsymbol{x}} \mathbb{E}_{\boldsymbol{x_1}, \boldsymbol{x_2} \in \mathcal{A}(\boldsymbol{x})}[f(\boldsymbol{x}_1)f(\boldsymbol{x}_2)^\top] - I_{d^*} \right\|_F$$

$$= \left\| \mathbb{E}_{\boldsymbol{x}} \mathbb{E}_{\boldsymbol{x_1}, \boldsymbol{x_2} \in \mathcal{A}(\boldsymbol{x})}[f(\boldsymbol{x}_1)f(\boldsymbol{x}_1)^\top] + \mathbb{E}_{\boldsymbol{x}} \mathbb{E}_{\boldsymbol{x_1}, \boldsymbol{x_2} \in \mathcal{A}(\boldsymbol{x})}[f(\boldsymbol{x}_1)\big(f(\boldsymbol{x}_2) - f(\boldsymbol{x}_1)\big)^\top] - I_{d^*} \right\|_F$$

$$\leq \left\| \mathbb{E}_{\boldsymbol{x}} \mathbb{E}_{\boldsymbol{x_1} \in \mathcal{A}(\boldsymbol{x})}[f(\boldsymbol{x}_1)f(\boldsymbol{x}_1)^\top] - I_{d^*} \right\|_F + \mathbb{E}_{\boldsymbol{x}} \mathbb{E}_{\boldsymbol{x_1}, \boldsymbol{x_2}}[\|f(\boldsymbol{x}_1)\|_2 \|f(\boldsymbol{x}_1) - f(\boldsymbol{x}_2)\|_2]$$

$$\leq \left\| \mathbb{E}_{\boldsymbol{x}} \mathbb{E}_{\boldsymbol{x'} \in \mathcal{A}(\boldsymbol{x})}[f(\boldsymbol{x}')f(\boldsymbol{x}')^\top] - I_{d^*} \right\|_F + B_2 \mathbb{E}_{\boldsymbol{x}} \mathbb{E}_{\boldsymbol{x_1}, \boldsymbol{x_2}} \|f(\boldsymbol{x}_1) - f(\boldsymbol{x}_2)\|_2. \qquad (\|f\|_2 \leq B_2)$$

Above deduction tells us that finding a measurable map $f : B_1 \leq \|f\|_2 \leq B_2$ making both $\mathcal{L}_{\text{align}}(f)$ and $\left\| \mathbb{E}_{\boldsymbol{x}} \mathbb{E}_{\boldsymbol{x'} \in \mathcal{A}(\boldsymbol{x})}[f(\boldsymbol{x}')f(\boldsymbol{x}')^\top] - I_{d^*} \right\|_F$ vanished is just enough to achieve our goal.

**Lemma B.8.** *If there exists a measurable map $f$ making $\Sigma = \mathbb{E}_{\boldsymbol{x} \sim P_s}[f(\boldsymbol{x})f(\boldsymbol{x})^\top]$ positive definite, then there exists a measurable map $\tilde{f}$ making both*

$$\mathcal{L}_{\text{align}}(\tilde{f}) = 0 \text{ and } \|\mathbb{E}_{\boldsymbol{x}} \mathbb{E}_{\boldsymbol{x'} \in \mathcal{A}(\boldsymbol{x})}[\tilde{f}(\boldsymbol{x}')\tilde{f}(\boldsymbol{x}')^\top] - I_{d^*}\|_F = 0.$$

*Proof.* We conduct following revision for given $f$ to obtain $\tilde{f}$.

For any $\boldsymbol{x} \in \mathcal{X}$, define

$$\tilde{f}_{\boldsymbol{x}}(\boldsymbol{x}') = \begin{cases} V^{-1} f(\boldsymbol{x}) & \text{if } \boldsymbol{x}' \in \mathcal{A}(\boldsymbol{x}) \\ f(\boldsymbol{x}) & \text{if } \boldsymbol{x}' \notin \mathcal{A}(\boldsymbol{x}) \end{cases}$$

where $\Sigma = VV^\top$, which is the Cholesky decomposition of $\Sigma$. It is well-defined as $\Sigma$ is positive definite. Iteratively repeat this argument for all $\boldsymbol{x} \in \mathcal{X}$ to yield $\tilde{f}$, then we have

$$\mathbb{E}_{\boldsymbol{x}} \mathbb{E}_{\boldsymbol{x'} \in \mathcal{A}(\boldsymbol{x})}[\tilde{f}(\boldsymbol{x}')\tilde{f}(\boldsymbol{x}')^\top] = V^{-1} \mathbb{E}_{\boldsymbol{x}}[f(\boldsymbol{x})f(\boldsymbol{x})^\top] V^{-T} = I_{d^*}$$

and

$$\forall \boldsymbol{x} \in \mathcal{X}, \boldsymbol{x}_1, \boldsymbol{x}_2 \in \mathcal{A}(\boldsymbol{x}), \|\tilde{f}(\boldsymbol{x}_1) - \tilde{f}(\boldsymbol{x}_2)\|_2 = \|\tilde{f}(\boldsymbol{x}) - \tilde{f}(\boldsymbol{x})\|_2 = 0.$$

That is what we desired. □

*Remark* B.2. If we have a measurable partition $\mathcal{X} = \cup_{i=1}^{d^*} \mathcal{P}_i$ stated in Assumption 4.5 such that $\mathcal{P}_i \cap \mathcal{P}_j = \emptyset$ and $\forall i \in [d^*], \frac{1}{B_2^2} \leq P_s(\mathcal{P}_i) \leq \frac{1}{B_1^2}$, just set the $f(\boldsymbol{x}) = \boldsymbol{e}_i$ if $\boldsymbol{x} \in \mathcal{P}_i$, where $\boldsymbol{e}_i$ is the standard basis of $\mathbb{R}^{d^*}$, then $\Sigma = \mathrm{diag}\{P_s(\mathcal{P}_1), \ldots, P_s(\mathcal{P}_i), \ldots, P_s(\mathcal{P}_{d^*})\}, V^{-1} = \mathrm{diag}\{\sqrt{\frac{1}{P_s(\mathcal{P}_1)}}, \ldots, \sqrt{\frac{1}{P_s(\mathcal{P}_i)}}, \ldots, \sqrt{\frac{1}{P_s(\mathcal{P}_{d^*})}}\}, \tilde{f}(\boldsymbol{x}) = \sqrt{\frac{1}{P_s(\mathcal{P}_i)}} \boldsymbol{e}_i$ if $\boldsymbol{x} \in \mathcal{P}_i$, it is obviously that $B_1 \leq \|\tilde{f}\|_2 \leq B_2$.

### B.2.7 RISK DECOMPOSITION

If denote $\widehat{G}(f) = \frac{1}{n_s} \sum_{i=1}^{n_s} f(\boldsymbol{x}_1^{(i)}) f(\boldsymbol{x}_2^{(i)})^\top - I_{d^*}$ and $G^*(f) = \mathbb{E}_{\boldsymbol{x}} \mathbb{E}_{\boldsymbol{x}_1, \boldsymbol{x}_2 \in \mathcal{A}(\boldsymbol{x})}[f(\boldsymbol{x}_1) f(\boldsymbol{x}_2)^\top] - I_{d^*}$,

we can decompose $\mathcal{E}(\hat{f}_{n_s})$ into three terms shown as follow and then deal each term successively. To achieve conciseness in subsequent conclusions, we employ $X \lesssim Y$ or $Y \gtrsim X$ to indicate the statement that $X \leq CY$ form some $C > 0$ if $X$ and $Y$ are two quantities.

**Lemma B.9.** *The excess risk $\mathcal{E}(\hat{f}_{n_s})$ satisfies*

$$\mathcal{E}(\hat{f}_{n_s}) \leq 2 \underbrace{\sup_{f \in \mathcal{F}, G \in \widehat{\mathcal{G}}(f)} |\mathcal{L}(f,G) - \widehat{\mathcal{L}}(f,G)|}_{\text{statistical error}\,:\,\mathcal{E}_{\text{sta}}} + \underbrace{\inf_{f \in \mathcal{F}} \{\sup_{G \in \mathcal{G}(f)} \mathcal{L}(f,G) - \sup_{G \in \mathcal{G}(f^*)} \mathcal{L}(f^*,G)\}}_{\text{approximation error of } \mathcal{F}\,:\,\mathcal{E}_{\mathcal{F}}}$$

$$+ \underbrace{\sup_{f \in \mathcal{F}} \{\sup_{G \in \mathcal{G}(f)} \mathcal{L}(f,G) - \mathcal{L}(f,\widehat{G}(f))\} + 2(B_2^2 + \sqrt{d^*}) \sup_{f \in \mathcal{F}} \{\mathbb{E}_{D_s}[\|\widehat{G}(f)\|_F] - \|G^*(f)\|_F\}}_{\text{approximation error of } \widehat{\mathcal{G}}\,:\,\mathcal{E}_{\widehat{\mathcal{G}}}},$$

*That is,*

$$\mathcal{E}(\hat{f}_{n_s}) \leq 2\mathcal{E}_{\text{sta}} + \mathcal{E}_{\mathcal{F}} + \mathcal{E}_{\widehat{\mathcal{G}}}.$$

*Proof.* Recall $\mathcal{F} = \mathcal{NN}_{d,d^*}(W, L, \mathcal{K}, B_1, B_2)$, for any $f \in \mathcal{F}$,

$$\sup_{G \in \mathcal{G}(\hat{f}_{n_s})} \mathcal{L}(\hat{f}_{n_s}, G) - \sup_{G \in \mathcal{G}(f^*)} \mathcal{L}(f^*, G)$$

$$= \left[ \sup_{G \in \mathcal{G}(\hat{f}_{n_s})} \mathcal{L}(\hat{f}_{n_s}, G) - \sup_{G \in \widehat{\mathcal{G}}(\hat{f}_{n_s})} \mathcal{L}(\hat{f}_{n_s}, G) \right] + \left[ \sup_{G \in \widehat{\mathcal{G}}(\hat{f}_{n_s})} \mathcal{L}(\hat{f}_{n_s}, G) - \sup_{G \in \widehat{\mathcal{G}}(\hat{f}_{n_s})} \widehat{\mathcal{L}}(\hat{f}_{n_s}, G) \right]$$

$$+ \left[ \sup_{G \in \widehat{\mathcal{G}}(\hat{f}_{n_s})} \widehat{\mathcal{L}}(\hat{f}_{n_s}, G) - \sup_{G \in \widehat{\mathcal{G}}(f)} \widehat{\mathcal{L}}(f, G) \right] + \left[ \sup_{G \in \widehat{\mathcal{G}}(f)} \widehat{\mathcal{L}}(f, G) - \sup_{G \in \widehat{\mathcal{G}}(f)} \mathcal{L}(f, G) \right]$$

$$+ \left[ \sup_{G \in \widehat{\mathcal{G}}(f)} \mathcal{L}(f, G) - \sup_{G \in \mathcal{G}(f)} \mathcal{L}(f, G) \right] + \left[ \sup_{G \in \mathcal{G}(f)} \mathcal{L}(f, G) - \sup_{G \in \mathcal{G}(f^*)} \mathcal{L}(f^*, G) \right],$$

where the second and fourth terms can be bounded by $\mathcal{E}_{\text{sta}}$. In fact, regarding to the fourth term, we have

$$\sup_{G \in \widehat{\mathcal{G}}(f)} \widehat{\mathcal{L}}(f, G) - \sup_{G \in \widehat{\mathcal{G}}(f)} \mathcal{L}(f, G) \leq \sup_{G \in \widehat{\mathcal{G}}(f)} \{\widehat{\mathcal{L}}(f, G) - \mathcal{L}(f, G)\}$$

$$\leq \sup_{G \in \widehat{\mathcal{G}}(f)} |\widehat{\mathcal{L}}(f, G) - \mathcal{L}(f, G)|$$

$$\leq \sup_{f \in \mathcal{F}, G \in \widehat{\mathcal{G}}(f)} |\widehat{\mathcal{L}}(f, G) - \mathcal{L}(f, G)|,$$

and the same conclusion holds for the second term.

The addition of first term and fifth term can be bounded by $\mathcal{E}_{\widehat{\mathcal{G}}}$. Actually, for the first term

$$\sup_{G \in \mathcal{G}(\hat{f}_{n_s})} \mathcal{L}(\hat{f}_{n_s}, G) - \sup_{G \in \widehat{\mathcal{G}}(\hat{f}_{n_s})} \mathcal{L}(\hat{f}_{n_s}, G) \leq \sup_{f \in \mathcal{F}} \{\sup_{G \in \mathcal{G}(f)} \mathcal{L}(f, G) - \sup_{G \in \widehat{\mathcal{G}}(f)} \mathcal{L}(f, G)\}$$

$$\leq \sup_{f \in \mathcal{F}} \{\sup_{G \in \mathcal{G}(f)} \mathcal{L}(f, G) - \mathcal{L}(f, \widehat{G}(f))\},$$

$$(\text{As } \widehat{G}(f) \in \widehat{\mathcal{G}}(f))$$

and for the fifth term, we have

$$\sup_{G \in \widehat{\mathcal{G}}(f)} \mathcal{L}(f, G) - \sup_{G \in \mathcal{G}(f)} \mathcal{L}(f, G)$$

$$= \sup_{G \in \widehat{\mathcal{G}}(f)} \mathbb{E}_{D_s}\big[\langle \widehat{G}(f), G \rangle_F\big] - \sup_{G \in \mathcal{G}(f)} \langle G^*(f), G \rangle_F \qquad (\langle G^*(f), G \rangle_F = \mathbb{E}_{D_s}\big[\langle \widehat{G}(f), G \rangle_F\big])$$

$$\leq \mathbb{E}_{D_s}\Big[ \sup_{G \in \widehat{\mathcal{G}}(f)} \langle \widehat{G}(f), G \rangle_F\Big] - \sup_{G \in \mathcal{G}(f)} \langle G^*(f), G \rangle_F$$

$$= \mathbb{E}_{D_s}\big[\|\widehat{G}(f)\|_F^2\big] - \|G^*(f)\|_F^2$$

$$\leq 2(B_2^2 + \sqrt{d^*})\big(\mathbb{E}_{D_s}\big[\|\widehat{G}(f)\|_F\big] - \|G^*(f)\|_F\big)$$
$$\text{(Both } \|\widehat{G}(f)\|_F \leq B_2^2 + \sqrt{d^*} \text{ and } \|G^*(f)\|_F \leq B_2^2 + \sqrt{d^*} \text{ hold)}$$

$$\leq 2(B_2^2 + \sqrt{d^*})\big( \sup_{f \in \mathcal{F}}\{\mathbb{E}_{D_s}\big[\|\widehat{G}(f)\|_F\big] - \|G^*(f)\|_F\}\big)$$

which yields what we desired.

Apart from that, the third term $\sup_{G \in \widehat{\mathcal{G}}(\hat{f}_{n_s})} \widehat{\mathcal{L}}(\hat{f}_{n_s}, G) - \sup_{G \in \widehat{\mathcal{G}}(f)} \widehat{\mathcal{L}}(f, G) \leq 0$ because of the definition of $\hat{f}_{n_s}$. Taking infimum over all $f \in \mathcal{NN}_{d, d^*}(W, L, \mathcal{K}, B_1, B_2)$ yields

$$\mathcal{E}(\hat{f}_{n_s}) \leq 2\mathcal{E}_{\text{sta}} + \mathcal{E}_{\mathcal{F}} + \mathcal{E}_{\widehat{\mathcal{G}}},$$

which completes the proof. $\qquad \square$

### B.2.8 BOUND $\mathcal{E}_{\text{sta}}$

**Lemma B.10.** *Regarding to $\mathcal{E}_{\text{sta}}$, we have*

$$\mathbb{E}_{D_s}[\mathcal{E}_{\text{sta}}] \lesssim \frac{\mathcal{K}\sqrt{L}}{\sqrt{n_s}}.$$

*Proof.* We are going to be introducing the relevant notations at first.

For any $f : \mathbb{R}^d \to \mathbb{R}^{d^*}$, let $\tilde{f} : \mathbb{R}^{2d} \to \mathbb{R}^{2d^*}$ such that $\tilde{f}(\tilde{\boldsymbol{x}}) = (f(\boldsymbol{x}_1), f(\boldsymbol{x}_2))$, where $\tilde{\boldsymbol{x}} = (\boldsymbol{x}_1, \boldsymbol{x}_2) \in \mathbb{R}^{2d}$. Furthermore, define $\widetilde{\mathcal{F}} := \{\tilde{f} : f \in \mathcal{NN}_{d, d^*}(W, L, \mathcal{K})\}$ and denote $D'_s = \{\tilde{\boldsymbol{x}}'^{(i)}\}_{i=1}^{n_s}$ as an independent identically distributed samples to $D_s$, which is called as ghost samples of $D_s$.

Next, we are attempt to establish the relationship between $\mathbb{E}_{D_s}[\mathcal{E}_{\text{sta}}]$ and the Rademacher complexity of $\mathcal{NN}_{d, d^*}(W, L, \mathcal{K})$. By the definition of $\mathcal{E}_{\text{sta}}$, we have

$$\mathbb{E}_{D_s}[\mathcal{E}_{\text{sta}}] = \mathbb{E}_{D_s}\Big[ \sup_{f \in \mathcal{NN}_{d, d^*}(W, L, \mathcal{K}, B_1, B_2), G \in \widehat{\mathcal{G}}(f)} |\mathcal{L}(f, G) - \widehat{\mathcal{L}}(f, G)|\Big]$$

$$\leq \mathbb{E}_{D_s}\Big[ \sup_{(f, G) \in \mathcal{NN}_{d, d^*}(W, L, \mathcal{K}, B_1, B_2) \times \mathcal{G}_1} |\mathcal{L}(f, G) - \widehat{\mathcal{L}}(f, G)|\Big]$$
$$\text{(As } \widehat{\mathcal{G}}(f) \subseteq \mathcal{G}_1 \text{ for any } f \in \mathcal{NN}_{d, d^*}(W, L, \mathcal{K}, B_1, B_2))$$

$$\leq \mathbb{E}_{D_s}\Big[ \sup_{(f, G) \in \mathcal{NN}_{d, d^*}(W, L, \mathcal{K}) \times \mathcal{G}_1} |\mathcal{L}(f, G) - \widehat{\mathcal{L}}(f, G)|\Big]$$
$$\text{(As } \mathcal{NN}_{d, d^*}(W, L, \mathcal{K}, B_1, B_2) \subseteq \mathcal{NN}_{d, d^*}(W, L, \mathcal{K}))$$

$$= \mathbb{E}_{D_s}\Big[ \sup_{(\tilde{f}, G) \in \widetilde{\mathcal{F}} \times \mathcal{G}_1} \Big| \frac{1}{n_s} \sum_{i=1}^{n_s} \mathbb{E}_{D'_s}[\ell(\tilde{f}(\tilde{\boldsymbol{x}}'^{(i)}), G)] - \frac{1}{n_s} \sum_{i=1}^{n_s} \ell(\tilde{f}(\tilde{\boldsymbol{x}}^{(i)}), G)\Big|\Big]$$

$$\leq \mathbb{E}_{D_s, D'_s}\Big[ \sup_{(\tilde{f}, G) \in \widetilde{\mathcal{F}} \times \mathcal{G}_1} \Big| \frac{1}{n_s} \sum_{i=1}^{n_s} \ell(\tilde{f}(\tilde{\boldsymbol{x}}'^{(i)}), G) - \frac{1}{n_s} \sum_{i=1}^{n_s} \ell(\tilde{f}(\tilde{\boldsymbol{x}}^{(i)}), G)\Big|\Big]$$

$$= \mathbb{E}_{D_s, D'_s, \xi}\Big[ \sup_{(\tilde{f}, G) \in \widetilde{\mathcal{F}} \times \mathcal{G}_1} \Big| \frac{1}{n_s} \sum_{i=1}^{n_s} \xi_i \big(\ell(\tilde{f}(\tilde{\boldsymbol{x}}'^{(i)}), G) - \ell(\tilde{f}(\tilde{\boldsymbol{x}}^{(i)}), G)\big)\Big|\Big] \qquad (34)$$

$$\leq 2\mathbb{E}_{D_s,\xi}\Big[\sup_{(\widetilde{f},G)\in\widetilde{\mathcal{F}}\times\mathcal{G}_1}\Big|\frac{1}{n_s}\sum_{i=1}^{n_s}\xi_i\ell(\widetilde{f}(\tilde{\boldsymbol{x}}^{(i)}),G)\Big|\Big]$$

$$\leq 4\sqrt{2}\|\ell\|_{\mathrm{Lip}}\Big(\mathbb{E}_{D_s,\xi}\Big[\sup_{f\in\mathcal{NN}_{d,d^*}(W,L,\mathcal{K})}\Big|\frac{1}{n_s}\sum_{i=1}^{n_s}\sum_{j=1}^{d^*}\xi_{i,j,1}f_j(\boldsymbol{x}_1^{(i)})+\xi_{i,j,2}f_j(\boldsymbol{x}_2^{(i)})\Big|\Big]$$

$$+\mathbb{E}_{\xi}\Big[\sup_{G\in\mathcal{G}_1}\Big|\frac{1}{n_s}\sum_{i=1}^{n_s}\sum_{j=1}^{d^*}\sum_{k=1}^{d^*}\xi_{i,j,k}G_{jk}\Big|\Big]\Big) \tag{35}$$

$$\leq 8\sqrt{2}\|\ell\|_{\mathrm{Lip}}\mathbb{E}_{D_s,\xi}\Big[\sup_{f\in\mathcal{NN}_{d,d^*}(W,L,\mathcal{K})}\Big|\frac{1}{n_s}\sum_{i=1}^{n_s}\sum_{j=1}^{d^*}\xi_{i,j,1}f_j(\boldsymbol{x}_1^{(i)})\Big|\Big]+4\sqrt{2}d^*\|\ell\|_{\mathrm{Lip}}\varrho$$

$$+4\sqrt{2}\|\ell\|_{\mathrm{Lip}}\mathbb{E}_{\xi}\Big[\max_{G\in\mathcal{N}_{\mathcal{G}_1}(\varrho)}\Big|\frac{1}{n_s}\sum_{i=1}^{n_s}\sum_{j=1}^{d^*}\sum_{k=1}^{d^*}\xi_{i,j,k}G_{jk}\Big|\Big] \tag{36}$$

$$\leq 8\sqrt{2}\|\ell\|_{\mathrm{Lip}}\mathbb{E}_{D_s,\xi}\Big[\sup_{f\in\mathcal{NN}_{d,d^*}(W,L,\mathcal{K})}\Big|\frac{1}{n_s}\sum_{i=1}^{n_s}\sum_{j=1}^{d^*}\xi_{i,j}f_j(\boldsymbol{x}_1^{(i)})\Big|\Big]+4\sqrt{2}d^*\|\ell\|_{\mathrm{Lip}}\varrho$$

$$+4\sqrt{2}(B_2^2+\sqrt{d^*})\|\ell\|_{\mathrm{Lip}}\sqrt{\frac{2\log\big(2|\mathcal{N}_{\mathcal{G}_1}(\varrho)|\big)}{n_s}} \tag{37}$$

$$\leq 8\sqrt{2}d^*\|\ell\|_{\mathrm{Lip}}\mathbb{E}_{D_s,\xi}\Big[\sup_{f\in\mathcal{NN}_{d,1}(W,L,\mathcal{K})}\Big|\frac{1}{n_s}\sum_{i=1}^{n_s}\xi_if(\boldsymbol{x}_1^{(i)})\Big|\Big]+4\sqrt{2}d^*\|\ell\|_{\mathrm{Lip}}\varrho$$

$$+4\sqrt{2}(B_2^2+\sqrt{d^*})\|\ell\|_{\mathrm{Lip}}\sqrt{\frac{2\log\big(2(\frac{3}{(B_2^2+\sqrt{d^*})\varrho})^{(d^*)^2}\big)}{n_2}}$$

$$\hspace{3cm}(|\mathcal{N}_{\mathcal{G}_1}(\varrho)|\leq(\tfrac{3}{(B_2^2+\sqrt{d^*})\varrho})^{(d^*)^2})$$

$$\lesssim\frac{\mathcal{K}\sqrt{L}}{\sqrt{n_s}}+\sqrt{\frac{\log n_s}{n_s}} \hspace{2cm}(\text{Lemma B.6 and set }\varrho=\mathcal{O}(1/\sqrt{n_s}))$$

$$\lesssim\frac{\mathcal{K}\sqrt{L}}{\sqrt{n_s}} \hspace{3cm}(\text{If }\mathcal{K}\gtrsim\sqrt{\log n_s})$$

Where (34) stems from the fact that $\xi_i\big(\ell(\widetilde{f}(\tilde{\boldsymbol{x}}'^{(i)}),G)-\ell(\widetilde{f}(\tilde{\boldsymbol{x}}^{(i)}),G)\big)$ has identical distribution with $\ell(\widetilde{f}(\tilde{\boldsymbol{x}}'^{(i)}),G)-\ell(\widetilde{f}(\tilde{\boldsymbol{x}}^{(i)}),G)$. As we have shown that $\|\ell\|_{\mathrm{Lip}}<\infty$, just apply Lemma B.5 to obtain (35). Regarding (36), as $\mathcal{N}_{\mathcal{G}_1}(\rho)$ is a $\rho$-covering, for any fixed $G\in\mathcal{G}_1$, we can find a $H_G\in\mathcal{N}_{\mathcal{G}_1}(\rho)$ satisfying $\|G-H_G\|_F\leq\rho$, therefore we have

$$\mathbb{E}_{\xi}\Big[\max_{G\in\mathcal{G}_1}\Big|\frac{1}{n_s}\sum_{i=1}^{n_s}\sum_{j=1}^{d^*}\sum_{k=1}^{d^*}\xi_{i,j,k}\big((H_G)_{jk}+G_{jk}-(H_G)_{jk}\big)\Big|\Big]$$

$$\leq\mathbb{E}_{\xi}\Big[\max_{G\in\mathcal{G}_1}\Big|\frac{1}{n_s}\sum_{i=1}^{n_s}\sum_{j=1}^{d^*}\sum_{k=1}^{d^*}\xi_{i,j,k}(H_G)_{jk}\Big|\Big]+\mathbb{E}_{\xi}\Big[\max_{G\in\mathcal{G}_1}\Big|\frac{1}{n_s}\sum_{i=1}^{n_s}\sum_{j=1}^{d^*}\sum_{k=1}^{d^*}\xi_{i,j,k}\big(G_{jk}-(H_G)_{jk}\big)\Big|\Big]$$

$$\leq\mathbb{E}_{\xi}\Big[\max_{G\in\mathcal{N}_{\mathcal{G}_1}(\rho)}\Big|\frac{1}{n_s}\sum_{i=1}^{n_s}\sum_{j=1}^{d^*}\sum_{k=1}^{d^*}\xi_{i,j,k}G_{jk}\Big|\Big]+\frac{1}{n_s}\sqrt{(d^*)^2n_s}\sqrt{n_s\sum_{j=1}^{d^*}\sum_{k=1}^{d^*}\big(G_{jk}-(H_G)_{jk}\big)^2}$$

$$\hspace{6cm}(\text{Cauchy-Schwarz inequality})$$

$$\leq\mathbb{E}_{\xi}\Big[\max_{G\in\mathcal{N}_{\mathcal{G}_1}(\rho)}\Big|\frac{1}{n_s}\sum_{i=1}^{n_s}\sum_{j=1}^{d^*}\sum_{k=1}^{d^*}\xi_{i,j,k}G_{jk}\Big|\Big]+d^*\rho.$$

To turn out the last term of (37), notice that $\|G\|_F \leq B_2^2 + \sqrt{d^*}$ implies that $\sum_{j=1}^{d^*} \sum_{k=1}^{d^*} \xi_{i,j,k} G_{jk} \sim$

$\mathrm{subG}(B_2^2 + \sqrt{d^*})$, therefore $\frac{1}{n_s} \sum_{i=1}^{n_s} \sum_{j=1}^{d^*} \sum_{k=1}^{d^*} \xi_{i,j,k} G_{jk} \sim \mathrm{subG}(B_2^2 + \sqrt{d^*})$, just apply Lemma B.7 to finish the proof. $\qquad\square$

### B.2.9  BOUND $\mathcal{E}_{\mathcal{F}}$

If we denote

$$\mathcal{E}(\mathcal{H}^\alpha, \mathcal{NN}_{d,1}(W, L, \mathcal{K})) := \sup_{g \in \mathcal{H}^\alpha} \inf_{f \in \mathcal{NN}_{d,1}(W,L,\mathcal{K})} \|f - g\|_{C([0,1]^d)},$$

where $C([0,1]^d)$ is the space of continuous functions on $[0,1]^d$ equipped with the sup-norm. Theorem 3.2 of Jiao et al. (2023) has already proven $\mathcal{E}(\mathcal{H}^\alpha, \mathcal{NN}_{d,1}(W, L, \mathcal{K}))$ can be bound by a quantity related to $\mathcal{K}$ when setting appropriate architecture of network, that is

**Lemma B.11** (Theorem 3.2 of Jiao et al. (2023)). *Let $d \in \mathbb{N}$ and $\alpha = r + \beta > 0$, where $r \in \mathbb{N}_0$ and $\beta \in (0,1]$. There exists $c > 0$ such that for any $\mathcal{K} \geq 1$, any $W \geq c\mathcal{K}^{(2d+\alpha)/(2d+2)}$ and $L \geq 2\lceil \log_2(d+r) \rceil + 2$,*

$$\mathcal{E}(\mathcal{H}^\alpha, \mathcal{NN}_{d,1}(W, L, \mathcal{K})) \lesssim \mathcal{K}^{-\alpha/(d+1)}.$$

For utilizing this conclusion, first notice that

$$\inf_{f \in \mathcal{NN}_{d,d^*}(W,L,\mathcal{K})} \|f(\boldsymbol{u}) - f^*(\boldsymbol{u})\|_2$$

$$= \inf_{f \in \mathcal{NN}_{d,d^*}(W,L,\mathcal{K})} \sqrt{\sum_{i=1}^{d^*} (f_i(\boldsymbol{u}) - f_i^*(\boldsymbol{u}))^2}$$

$$\leq \inf_{f \in \mathcal{NN}_{d,d^*}(W,L,\mathcal{K})} \sqrt{\sum_{i=1}^{d^*} \|f_i - f_i^*\|_{C([0,1]^d)}^2}$$

$$\leq \sup_{g \in \mathcal{H}^\alpha} \inf_{f \in \mathcal{NN}_{d,d^*}(W,L,\mathcal{K})} \sqrt{\sum_{i=1}^{d^*} \|f_i - g\|_{C([0,1]^d)}^2}$$

$$\leq \sup_{g \in \mathcal{H}^\alpha} \sqrt{\sum_{i=1}^{d^*} \inf_{f \in \mathcal{NN}_{d,1}(\lfloor W/d^* \rfloor, L, \mathcal{K})} \|f - g\|_{C([0,1]^d)}^2}$$

$$\leq \sqrt{d^*} \mathcal{E}(\mathcal{H}^\alpha, \mathcal{NN}_{d,1}(\lfloor W/d^* \rfloor, L, \mathcal{K}))$$

$$\lesssim \mathcal{K}^{-\alpha/(d+1)},$$

where the third to last line inequality is from following reason: if $f_i \in \mathcal{NN}_{d,1}(\lfloor W/d^* \rfloor, L, \mathcal{K})$, where $i \in [d^*]$, whose parameter are independent with each other, then their concatenation $f = (f_1, f_2, \cdots, f_{d^*})^\top$ can be regarded as an elements of $\mathcal{NN}_{d,d^*}(W, D, \mathcal{K})$ with specific parameters, by following Proposition B.2, we have $f \in \mathcal{NN}_{d,d^*}(W, L, \mathcal{K})$.

**Proposition B.2** ((iii) of Proposition 2.5 in Jiao et al. (2023)). *Let $\phi_1 \in \mathcal{NN}_{d,d_1^*}(w_1, L_1, \mathcal{K}_1)$ and $\phi_2 \in \mathcal{NN}_{d,d_2^*}(W_2, L_2, \mathcal{K}_2)$, define $\phi(\boldsymbol{x}) := (\phi_1(\boldsymbol{x}), \phi_2(\boldsymbol{x}))$, then $\phi \in \mathcal{NN}_{d,d_1^*+d_2^*}(W_1 + W_2, \max\{L_1, L_2\}, \max\{\mathcal{K}_1, \mathcal{K}_2\})$.*

Above conclusion implies optimal approximation element of $f^*$ in $\mathcal{NN}_{d,d^*}(W, L, \mathcal{K})$ can be arbitrarily close to $f^*$ under the setting that $\mathcal{K}$ is large enough. Hence we can conclude optimal approximation element of $f^*$ is also contained in $\mathcal{F} = \mathcal{NN}_{d,d^*}(W, L, \mathcal{K}, B_1, B_2)$ as the setting that $B_1 \leq \|f^*\|_2 \leq B_2$.

Therefore, if we denote

$$\mathcal{T}(f) := \mathbb{E}_{\boldsymbol{x}} \mathbb{E}_{\boldsymbol{x}_1, \boldsymbol{x}_2 \in \mathcal{A}(\boldsymbol{x})} [\|f(\boldsymbol{x}_1) - f(\boldsymbol{x}_2)\|_2^2] + \lambda \|\mathbb{E}_{\boldsymbol{x}} \mathbb{E}_{\boldsymbol{x}_1, \boldsymbol{x}_2 \in \mathcal{A}(\boldsymbol{x})} [f(\boldsymbol{x}_1) f(\boldsymbol{x}_2)^\top] - I_{d^*}\|_F^2,$$

we can yield the upper bound of $\mathcal{E}_{\mathcal{F}}$ by following deduction

$$
\begin{aligned}
\mathcal{E}_{\mathcal{F}} &= \inf_{f \in \mathcal{F}} \{ \sup_{G \in \mathcal{G}(f)} \mathcal{L}(f, G) - \sup_{G \in \mathcal{G}(f^*)} \mathcal{L}(f^*, G) \} \\
&= \inf_{f \in \mathcal{F}} \{ \mathcal{T}(f) - \mathcal{T}(f^*) \} \\
&= \inf_{f \in \mathcal{NN}_{d,d^*}(W,L,\mathcal{K})} \{ \mathcal{T}(f) - \mathcal{T}(f^*) \} \\
&\leq \|\ell\|_{\mathrm{Lip}} \inf_{f \in \mathcal{NN}_{d,d^*}(W,L,\mathcal{K})} \mathbb{E}_{\boldsymbol{x}} \mathbb{E}_{\tilde{\boldsymbol{x}}} \| \widetilde{f}(\tilde{\boldsymbol{x}}) - \tilde{f}^*(\tilde{\boldsymbol{x}}) \|_2 \qquad \text{(Proposition B.1)} \\
&\leq \|\ell\|_{\mathrm{Lip}} \inf_{f \in \mathcal{NN}_{d,d^*}(W,L,\mathcal{K})} \mathbb{E}_{\boldsymbol{x}} \mathbb{E}_{\boldsymbol{x}' \in \mathcal{A}(\boldsymbol{x})} \sqrt{2 \sum_{i=1}^{d^*} (f_i(\boldsymbol{x}') - f_i^*(\boldsymbol{x}'))^2} \\
&\leq \sqrt{2 d^*} \|\ell\|_{\mathrm{Lip}} \sup_{g \in \mathcal{H}^\alpha} \inf_{f \in \mathcal{NN}_{d,1}(\lfloor W/d^* \rfloor, L, \mathcal{K}/\sqrt{d^*})} \| f - g \|_{C([0,1]^d)} \\
&\leq \sqrt{2 d^*} \|\ell\|_{\mathrm{Lip}} \mathcal{E}(\mathcal{H}^\alpha, \mathcal{NN}_{d,1}(\lfloor W/d^* \rfloor, L, \mathcal{K}/\sqrt{d^*})) \\
&\lesssim \mathcal{K}^{-\alpha/(d+1)}.
\end{aligned}
$$

### B.2.10 BOUND $\mathcal{E}_{\widehat{\mathcal{G}}}$

Recall

$$
\mathcal{E}_{\widehat{\mathcal{G}}} = \sup_{f \in \mathcal{F}} \{ \sup_{G \in \mathcal{G}(f)} \mathcal{L}(f, G) - \mathcal{L}(f, \widehat{G}(f)) \} + 2(B_2^2 + \sqrt{d^*}) \sup_{f \in \mathcal{F}} \{ \mathbb{E}_{D_s} [\|\widehat{G}(f)\|_F] - \|G^*(f)\|_F \},
$$

then for the first item of $\mathcal{E}_{\widehat{\mathcal{G}}}$, we have

$$
\begin{aligned}
&\sup_{f \in \mathcal{F}} \{ \sup_{G \in \mathcal{G}(f)} \mathcal{L}(f, G) - \mathcal{L}(f, \widehat{G}(f)) \} \\
&= \sup_{f \in \mathcal{F}} \{ \mathcal{L}(f, G^*(f)) - \mathcal{L}(f, \widehat{G}(f)) \} \\
&\leq \sqrt{2} B_2 \sup_{f \in \mathcal{F}} \| G^*(f) - \widehat{G}(f) \|_F \qquad \text{(Look up Table 2 to yield } \ell(\boldsymbol{u}, \cdot) \in \mathrm{Lip}(\sqrt{2} B_2)) \\
&\leq \sqrt{2} B_2 \sup_{f \in \mathcal{F}} \left\| \mathbb{E}_{\boldsymbol{x}} \mathbb{E}_{\boldsymbol{x}_1, \boldsymbol{x}_2 \in \mathcal{A}(\boldsymbol{x})} [f(\boldsymbol{x}_1) f(\boldsymbol{x}_2)^\top] - \frac{1}{n_s} \sum_{i=1}^{n_s} f(\boldsymbol{x}_1^{(i)}) f(\boldsymbol{x}_2^{(i)})^\top \right\|_F.
\end{aligned}
$$

And regrading to the second term, we can yield

$$
\begin{aligned}
&\sup_{f \in \mathcal{F}} \{ \mathbb{E}_{D_s} [\|\widehat{G}(f)\|_F] - \|G^*(f)\|_F \} \\
&= \sup_{f \in \mathcal{F}} \left\{ \mathbb{E}_{D_s} \left[ \left\| \frac{1}{n_s} \sum_{i=1}^{n_s} f(\boldsymbol{x}_1^{(i)}) f(\boldsymbol{x}_2^{(i)})^\top - I_{d^*} \right\|_F - \left\| \mathbb{E}_{\boldsymbol{x}} \mathbb{E}_{\boldsymbol{x}_1, \boldsymbol{x}_2 \in \mathcal{A}(\boldsymbol{x})} [f(\boldsymbol{x}_1) f(\boldsymbol{x}_2)^\top] - I_{d^*} \right\|_F \right] \right\} \\
&\leq \sup_{f \in \mathcal{F}} \left\{ \mathbb{E}_{D_s} \left[ \left\| \frac{1}{n_s} \sum_{i=1}^{n_s} f(\boldsymbol{x}_1^{(i)}) f(\boldsymbol{x}_2^{(i)})^\top - \mathbb{E}_{\boldsymbol{x}} \mathbb{E}_{\boldsymbol{x}_1, \boldsymbol{x}_2 \in \mathcal{A}(\boldsymbol{x})} [f(\boldsymbol{x}_1) f(\boldsymbol{x}_2)^\top] \right\|_F \right] \right\} \\
&\leq \mathbb{E}_{D_s} \left[ \sup_{f \in \mathcal{F}} \left\{ \left\| \frac{1}{n_s} \sum_{i=1}^{n_s} f(\boldsymbol{x}_1^{(i)}) f(\boldsymbol{x}_2^{(i)})^\top - \mathbb{E}_{\boldsymbol{x}} \mathbb{E}_{\boldsymbol{x}_1, \boldsymbol{x}_2 \in \mathcal{A}(\boldsymbol{x})} [f(\boldsymbol{x}_1) f(\boldsymbol{x}_2)^\top] \right\|_F \right\} \right]
\end{aligned}
$$

Combine above two inequalities to turn out

$$
\begin{aligned}
\mathbb{E}_{D_s}[\mathcal{E}_{\widehat{\mathcal{G}}}] &\lesssim \mathbb{E}_{D_s} \left[ \sup_{f \in \mathcal{F}} \left\| \mathbb{E}_{\boldsymbol{x}} \mathbb{E}_{\boldsymbol{x}_1, \boldsymbol{x}_2 \in \mathcal{A}(\boldsymbol{x})} \left[ \frac{1}{n_s} \sum_{i=1}^{n_s} [\mathcal{M}(\widetilde{f}(\tilde{\boldsymbol{x}})) - \mathcal{M}(\widetilde{f}(\tilde{\boldsymbol{x}}^{(i)}))] \right] \right\|_F \right] \\
&\leq \|\mathcal{M}\|_{\mathrm{Lip}} \mathbb{E}_{D_s} \left[ \left\| \mathbb{E}_{\boldsymbol{x}} \mathbb{E}_{\boldsymbol{x}_1, \boldsymbol{x}_2 \in \mathcal{A}(\boldsymbol{x})} [\widetilde{f}(\tilde{\boldsymbol{x}})] - \frac{1}{n_s} \sum_{i=1}^{n_s} \widetilde{f}(\tilde{\boldsymbol{x}}^{(i)}) \right\|_2 \right]
\end{aligned}
$$

where $\mathcal{M}(\boldsymbol{u}) = \boldsymbol{u}_1 \boldsymbol{u}_2^\top$, where $\boldsymbol{u}_1, \boldsymbol{u}_2 \in \mathbb{R}^{d^*}$, we have shown it is a Lipchitz map on $\{\boldsymbol{u} \in \mathbb{R}^{2d^*} : \boldsymbol{u} \le \sqrt{2} B_2\}$ in Proposition B.1. By Multidimensional Chebyshev's inequality, we know that $P_s \big( \big\| \frac{1}{n_s} \sum_{i=1}^{n_s} \widetilde{f}(\tilde{\boldsymbol{x}}^{(i)}) - \mathbb{E}_{\boldsymbol{x}} \mathbb{E}_{\boldsymbol{x}_1, \boldsymbol{x}_2 \in \mathcal{A}(\boldsymbol{x})}[\widetilde{f}(\tilde{\boldsymbol{x}})] \big\|_2 \ge \frac{1}{n_s^{1/4}} \big) \le \frac{\mathbb{E}\|\widetilde{f}(\tilde{\boldsymbol{x}}) - \mathbb{E}[\widetilde{f}(\tilde{\boldsymbol{x}})]\|_2^2}{\sqrt{n_s}} \le \frac{8B_2^2}{\sqrt{n_s}}$ as $\|\widetilde{f}(\tilde{\boldsymbol{x}})\|_2 \le \sqrt{2} B_2$. Thus we have

$$\mathbb{E}_{D_s}[\mathcal{E}_{\widehat{\mathcal{G}}}] \lesssim \frac{1}{n_s^{1/4}} \cdot P_s \big( \big\| \frac{1}{n_s} \sum_{i=1}^{n_s} \widetilde{f}(\tilde{\boldsymbol{x}}^{(i)}) - \mathbb{E}_{\boldsymbol{x}} \mathbb{E}_{\boldsymbol{x}_1, \boldsymbol{x}_2 \in \mathcal{A}(\boldsymbol{x})}[\widetilde{f}(\tilde{\boldsymbol{x}})] \big\|_2 \ge \frac{1}{n_s^{1/4}} \big) + 2\sqrt{2} B_2 \cdot \frac{8B_2^2}{\sqrt{n_s}}$$

$$\text{(As } \|\widetilde{f}(\tilde{\boldsymbol{x}})\|_2 \le \sqrt{2} B_2)$$

$$\le \frac{1}{n_s^{1/4}} + 16\sqrt{2} B_2^3 \frac{1}{\sqrt{n_s}}$$

$$\lesssim \frac{1}{n_s^{1/4}}.$$

### B.2.11 TRADE OFF BETWEEN STATISTICAL ERROR AND APPROXIMATION ERROR

Let $W \ge c\mathcal{K}^{(2d+\alpha)/(2d+2)}$ and $L \ge 2\lceil \log_2(d+r) \rceil + 2$, combine the bound results of statistical error and approximation error to yield

$$\mathbb{E}_{D_s}[\mathcal{E}(\hat{f}_{n_s})] \le 2\mathbb{E}_{D_s}[\mathcal{E}_{\text{sta}}] + \mathcal{E}_{\mathcal{F}} + 2\mathbb{E}_{D_s}[\mathcal{E}_{\widehat{\mathcal{G}}}] \lesssim \frac{\mathcal{K}}{\sqrt{n_s}} + \mathcal{K}^{-\alpha/(d+1)}.$$

Taking $\mathcal{K} = n_s^{\frac{d+1}{2(\alpha+d+1)}}$ to yield

$$\mathbb{E}_{D_s}[\mathcal{E}(\hat{f}_{n_s})] \lesssim n_s^{-\frac{\alpha}{2(\alpha+d+1)}}.$$

As we have shown that $\sup_{G \in \mathcal{G}(f^*)} \mathcal{L}(f^*, G) = 0$, above inequality implies

$$\mathbb{E}_{D_s}\big[ \sup_{G \in \mathcal{G}(\hat{f}_{n_s})} \mathcal{L}(\hat{f}_{n_s}, G) \big] \lesssim n_s^{-\frac{\alpha}{2(\alpha+d+1)}}.$$

To ensure above deduction holds, We need to set $W \ge c n_s^{\frac{2d+\alpha}{4(\alpha+d+1)}}$ and $L \ge 2\lceil \log_2(d+r) \rceil + 2$.

### B.2.12 THE PROOF OF MAIN THEOREM

Next, we are going to prove our main theorem 4.2. We will state its formal version at first and then conclude Theorem 4.2 as a corollary.

To notation conciseness, let $p = \dfrac{\sqrt{\frac{2}{\min_{i \ne j} p_s(i) p_s(j)} \left( \frac{C}{\lambda} n_s^{-\frac{\alpha}{2(\alpha+d+1)}} + \psi(n_s) \right)} + 2\sqrt{d^*} B_2 M n_s^{-\frac{\nu}{2(\alpha+d+1)}}}{B_2^2 \Theta(\sigma_s^{(n_s)}, \delta_s^{(n_s)}, \varepsilon_{n_s}, \hat{f}_{n_s})}$, where

$C$ is a constant, $0 \le \psi(n_s) \lesssim (1 - \sigma_s^{(n_s)} + n_s^{-\frac{\min\{\alpha, \nu, \varsigma, \tau\}}{4(\alpha+d+1)}})^2 + (1 - \sigma_s^{(n_s)}) + n_s^{-\frac{\min\{\alpha, \nu, \varsigma, \tau\}}{8(\alpha+d+1)}}$, then the formal version of our main theoretical result can be stated as follow.

**Lemma B.12.** *When Assumption 4.1-4.5 all hold, set $\varepsilon_{n_s} = m^2 n_s^{-\frac{\min\{\alpha, \nu, \varsigma, \tau\}}{8(\alpha+d+1)}}, W \ge c n_s^{\frac{2d+\alpha}{4(\alpha+d+1)}}$, $L \ge 2\lceil \log_2(d+r) \rceil + 2, \mathcal{K} = n_s^{\frac{d+1}{2(\alpha+d+1)}}$ and $\mathcal{A} = \mathcal{A}_{n_s}$ in Assumption 4.3, then we have*

$$\mathbb{E}_{D_s}[R_t^2(\varepsilon_{n_s}, \hat{f}_{n_s})] \lesssim n_s^{-\frac{\min\{\alpha, \nu, \varsigma\}}{4(\alpha+d+1)}} \tag{38}$$

*and*

$$\mathbb{E}_{D_s}[\max_{i \ne j} |\mu_t(i)^\top \mu_t(j)|] \lesssim 1 - \sigma_s^{(n_s)} + n_s^{-\frac{\min\{\alpha, 2\tau\}}{4(\alpha+d+1)}}. \tag{39}$$

*Furthermore, If $\Theta(\sigma_s^{(n_s)}, \delta_s^{(n_s)}, \varepsilon_{n_s}, \hat{f}_{n_s}) > 0$, then with probability at least $1 - p$, we have*

$$\mathbb{E}_{D_s}[\text{Err}(Q_{\hat{f}_{n_s}})] \le (1 - \sigma_t^{(n_s)}) + \mathcal{O}(n_s^{-\frac{\min\{\alpha, \nu, \varsigma\}}{8(\alpha+d+1)}}).$$

*Proof.* First recall the conclusion we've got in Theorem B.1

$$\mathbb{E}_{D_s}[R_t^2(\varepsilon, \hat{f}_{n_s})] \leq \frac{m^4}{\varepsilon^2}\Big(\mathbb{E}_{D_s}[\sup_{G \in \mathcal{G}(\hat{f}_{n_s})} \mathcal{L}(\hat{f}_{n_s}, G)] + 8B_2 d^* M\mathcal{K}\rho + 4B_2^2 d^* K\eta\Big),$$

$$\mathbb{E}_{D_s}[\max_{i \neq j} |\mu_t(i)^\top \mu_t(j)|] \leq \sqrt{\frac{2}{\min_{i \neq j} p_s(i)p_s(j)}\Big(\frac{1}{\lambda}\mathbb{E}_{D_s}[\sup_{G \in \mathcal{G}(\hat{f}_{n_s})} \mathcal{L}(\hat{f}_{n_s}, G)] + \mathbb{E}_{D_s}[\psi(\sigma_s, \delta_s, \varepsilon, \hat{f}_{n_s})]\Big)}$$
$$+ 2\sqrt{d^*}B_2 M\mathcal{K}\rho,$$

and with probability at least

$$1 - \frac{\sqrt{\frac{2}{\min_{i \neq j} p_s(i)p_s(j)}\Big(\frac{1}{\lambda}\mathbb{E}_{D_s}[\sup_{G \in \mathcal{G}(\hat{f}_{n_s})} \mathcal{L}(\hat{f}_{n_s}, G)] + \psi(\sigma_s, \delta_s, \varepsilon, \hat{f}_{n_s})\Big)} + 2\sqrt{d^*}B_2 M\mathcal{K}\rho}{B_2^2 \Theta(\sigma_t, \delta_t, \varepsilon, \hat{f}_{n_s})},$$

we have

$$\mathbb{E}_{D_s}[\text{Err}(Q_{\hat{f}_{n_s}})] \leq (1 - \sigma_t) + \frac{m^2}{\varepsilon}\sqrt{\mathbb{E}_{D_s}[\sup_{G \in \mathcal{G}(\hat{f}_{n_s})} \mathcal{L}(\hat{f}_{n_s}, G)] + 8B_2 d^* M\mathcal{K}\rho + 4B_2^2 d^* K\eta},$$

where

$$\psi(\sigma_s, \delta_s, \varepsilon, \hat{f}_{n_s}) = 4B_2^2 \Big[\Big(1 - \sigma_s + \frac{\mathcal{K}\delta_s + 2\varepsilon}{2B_2}\Big)^2 + (1 - \sigma_s) + \frac{Km^2}{\varepsilon}\sqrt{\mathbb{E}_{D_s}[\sup_{G \in \mathcal{G}(\hat{f}_{n_s})} \mathcal{L}(\hat{f}_{n_s}, G)]}\Big(3 -$$

$$2\sigma_s + \frac{\mathcal{K}\delta_s + 2\varepsilon}{B_2}\Big) + \frac{m^4}{\varepsilon^2}\mathbb{E}_{D_s}[\sup_{G \in \mathcal{G}(\hat{f}_{n_s})} \mathcal{L}(\hat{f}_{n_s}, G)]\Big(\sum_{k=1}^K \frac{1}{p_s(k)}\Big)\Big] + B_2\Big(\varepsilon^2 +$$

$$\frac{4B_2^2 m^2}{\varepsilon}\sqrt{\mathbb{E}_{D_s}[\sup_{G \in \mathcal{G}(\hat{f}_{n_s})} \mathcal{L}(\hat{f}_{n_s}, G)]}\Big)^{\frac{1}{2}}.$$

To obtain the conclusion shown in this theorem from above formulations, first notice $\rho = n_s^{-\frac{\nu+d+1}{2(\alpha+d+1)}}$ and $\eta = n_s^{-\frac{\varsigma}{2(\alpha+d+1)}}$ by comparing Assumption 4.4 and Assumption B.1, apart from that, we have shown $\mathbb{E}_{D_s}[\sup_{G \in \mathcal{G}(\hat{f}_{n_s})} \mathcal{L}(\hat{f}_{n_s}, G)] \lesssim n_s^{-\frac{\alpha}{2(\alpha+d+1)}}$ in B.2.11 and known $\delta_s^{(n_s)} \leq n_s^{-\frac{\tau+d+1}{2(\alpha+d+1)}}$, combining with the setting $\varepsilon_{n_s} = m^2 n_s^{-\frac{\min\{\alpha,\nu,\varsigma,\tau\}}{8(\alpha+d+1)}}, \mathcal{K} = n_s^{\frac{d+1}{2(\alpha+d+1)}}$ implies that $\mathcal{K}\rho/\varepsilon_{n_s}^2 \leq n_s^{-\frac{\tau}{2(\alpha+d+1)}}, \eta/\varepsilon_{n_s}^2 \leq n_s^{-\frac{\tau}{2(\alpha+d+1)}}, \mathcal{K}\delta_s^{(n_s)} \leq n_s^{-\frac{\tau}{2(\alpha+d+1)}}$ and $\mathbb{E}_{D_s}[\sup_{G \in \mathcal{G}(\hat{f}_{n_s})} \mathcal{L}(\hat{f}_{n_s}, G)]/\varepsilon_{n_s}^2 \leq n_s^{-\frac{\alpha}{4(\alpha+d+1)}}$.

Plugin these facts into the corresponding term of above formulations to get what we desired. □

Let us first state the formal version of Theorem 4.2 and then prove it.

**Theorem B.3** (Formal version of Theorem 4.2). *If Assumptions 4.1-4.5 all hold, set $W \geq cn_s^{\frac{2d+\alpha}{4(\alpha+d+1)}}, L \geq 2\lceil\log_2(d+r)\rceil + 2, \mathcal{K} = n_s^{\frac{d+1}{2(\alpha+d+1)}}$ and $\mathcal{A} = \mathcal{A}_{n_s}$ in Assumption 4.3, then, provided that $n_s$ is sufficiently large, with probability at least $\sigma_s^{(n_s)} - \mathcal{O}\big(n_s^{-\frac{\min\{\alpha,\nu,\varsigma,\tau\}}{16(\alpha+d+1)}}\big) - \mathcal{O}\big(\frac{1}{\sqrt{\min_k n_t(k)}}\big)$, we have*

$$\mathbb{E}_{D_s}[\text{Err}(Q_{\hat{f}_{n_s}})] \leq (1 - \sigma_t^{(n_s)}) + \mathcal{O}(n_s^{-\frac{\min\{\alpha,\nu,\varsigma\}}{8(\alpha+d+1)}}).$$

*Proof of Theorem 4.2.* Note that the main difference between Theorem B.12 and Theorem 4.2 is the condition $\Theta(\sigma_s^{(n_s)}, \delta_s^{(n_s)}, \varepsilon_{n_s}, \hat{f}_{n_s}) > 0$, so we are going to focus on whether this condition holds under the condition of Theorem 4.2.

To show this, first recall

$$\Theta(\sigma_t^{(n_s)}, \delta_t^{(n_s)}, \varepsilon_{n_s}, \hat{f}_{n_s}) = \Gamma_{\min}(\sigma_t^{(n_s)}, \delta_t^{(n_s)}, \varepsilon_{n_s}, \hat{f}_{n_s}) - \sqrt{2 - 2\Gamma_{\min}(\sigma_t^{(n_s)}, \delta_t^{(n_s)}, \varepsilon_{n_s}, \hat{f}_{n_s})} - \frac{\Delta_{\hat{\mu}_t}}{2}$$

$$-\frac{2\max_{k\in[K]}\|\hat{\mu}_t(k)-\mu_t(k)\|_2}{B_2}.$$

Note (32) and dominated convergence theorem imply $R_t(\varepsilon_{n_s},\hat{f}_{n_s})\to 0$ a.s., thus

$$\Gamma_{\min}(\sigma_t^{(n_s)},\delta_t^{(n_s)},\varepsilon_{n_s},\hat{f}_{n_s}) = \Big(\sigma_t^{(n_s)}-\frac{R_t(\varepsilon_{n_s},\hat{f}_{n_s})}{\min_i p_t(i)}\Big)\Big(1+\big(\frac{B_1}{B_2}\big)^2-\frac{\mathcal{K}\delta_t^{(n_s)}}{B_2}-\frac{2\varepsilon_{n_s}}{B_2}\Big)-1$$

$$\to \Big(\frac{B_1}{B_2}\Big)^2$$

Combining with the fact that $\frac{\Delta_{\hat{\mu}_t}}{2}=\frac{1-\min_{k\in[K]}\|\hat{\mu}_t(k)\|^2/B_2^2}{2}<\frac{1}{2}$ can yield

$$\Gamma_{\min}(\sigma_t^{(n_s)},\delta_t^{(n_s)},\varepsilon_{n_s},\hat{f}_{n_s})-\sqrt{2-2\Gamma_{\min}(\sigma_t^{(n_s)},\delta_t^{(n_s)},\varepsilon_{n_s},\hat{f}_{n_s})}-\frac{\Delta_{\hat{\mu}_t}}{2}>1/2$$

if we select proper $B_1$ and $B_2$.

Besides that, by Multidimensional Chebyshev's inequality, we know that

$$P_t\big(\|\hat{\mu}_t(k)-\mu_t(k)\|_2\ge\frac{B_2}{8}\big)\le\frac{64\sqrt{\mathbb{E}_{\boldsymbol{z}\in\widetilde{C}_t(k)}\mathbb{E}_{\boldsymbol{z}'\in\mathcal{A}(\boldsymbol{z})}\|f(\boldsymbol{z}')-\mu_t(k)\|_2^2}}{B_2^2\sqrt{2n_t(k)}}\le\frac{128}{B_2\sqrt{n_t(k)}},$$

so that $\Theta(\sigma_t^{(n_s)},\delta_t^{(n_s)},\varepsilon_{n_s},\hat{f}_{n_s})\ge\frac{1}{4}$ with probability at least $1-\frac{128K}{B_2\sqrt{\min_k n_t(k)}}$ if $n_s$ is large enough, of course the condition $\Theta(\sigma_t^{(n_s)},\delta_t^{(n_s)},\varepsilon_{n_s},\hat{f}_{n_s})>0$ in Theorem B.12 can be satisfied.

Therefore, with probability at least

$$1-p-\frac{128K}{B_2\sqrt{\min_k n_t(k)}}\gtrsim 1-(1-\sigma_s^{(n_s)})-\mathcal{O}\big(n_s^{-\frac{\min\{\alpha,\nu,\varsigma,\tau\}}{16(\alpha+d+1)}}\big)-\mathcal{O}\big(\frac{1}{\sqrt{\min_k n_t(k)}}\big)$$

$$=\sigma_s^{(n_s)}-\mathcal{O}\big(n_s^{-\frac{\min\{\alpha,\nu,\varsigma,\tau\}}{16(\alpha+d+1)}}\big)-\mathcal{O}\big(\frac{1}{\sqrt{\min_k n_t(k)}}\big).$$

we have the conclusions shown in Theorem 4.2, which completes the proof. □

## C EXPERIMENTAL DETAILS

**Implementation details.** Except for tuning $\lambda$ for different dataset, all other hyper parameters used in our experiments are align with Ermolov et al. (2021). To be specific, we train $1,000$ epochs with learning rate $3\times10^{-3}$ for CIFAR-10, CIFAR-100 and $2\times10^{-3}$ for Tiny ImageNet. The learning rate warm-up is used for the first $500$ iterations of the optimizer, in addition to a $0.2$ learning rate drop $50$ and $25$ epochs before the end. We adopt a mini-batch size of $256$. Same as W-MSE 4 of Ermolov et al. (2021), we also set $4$ as the number of positive samples per image. The dimension of the hidden layer of the projection head is set as $1024$. The weight decay is $10^{-6}$. We adopt an embedding size $(d^*)$ of $64$ for CIFAR10, CIFAR100 and $128$ for Tiny ImageNet and employ the trick mentioned in Ermolov et al. (2021) during the pretraining process. The embedding size of BarlowTwins (Zbontar et al., 2021) is different from above as BarlowTwins need much larger representation size (1024) to guarantee its performance. As we see, the performance of our model can sufficiently outperform BarlowTwins, revealing the alignment term is pretty crucial for downstream performance practically. The backbone network used in our implementation is ResNet-18.

**Image transformation details.** We randomly extract crops with sizes ranging from $0.08$ to $1.0$ of the original area and aspect ratios ranging from $3/4$ to $4/3$ of the original aspect ratio. Furthermore, we apply horizontal mirroring with a probability of $0.5$. Additionally, color jittering is applied with a configuration of $(0.4; 0.4; 0.4; 0.1)$ and a probability of $0.8$, while grayscaling is applied with a probability of $0.2$. For CIFAR-10 and CIFAR-100, random Gaussian blurring is adopted with a probability of $0.5$ and a kernel size of $0.1$. During testing, only one crop is used for evaluation.

**Evaluation protocol.** During evaluation, we freeze the network encoder and remove the projection head after pretraining, then train a supervised linear classifier on top of it, which is a fully-connected

layer followed by softmax. we train the linear classifier for 500 epochs using the Adam optimizer with corresponding labeled training set without data augmentation. The learning rate is exponentially decayed from $10^{-2}$ to $10^{-6}$. The weight decay is set as $10^{-6}$. we also include the accuracy of a k-nearest neighbors classifier with $k = 5$, which does not require fine tuning.

All experiments were conducted using a single Tesla V100 GPU unit.

