# OpenReview forum: "Self-supervised Transfer Learning via Adversarial Contrastive Training"
_ICLR.cc/2025/Conference — Submitted to ICLR 2025_

### Official Review · Reviewer_dVpK · 2024-11-01

**Soundness:** 2
**Presentation:** 1
**Contribution:** 2
**Rating:** 3
**Confidence:** 2

**Summary:**

This paper considers the problem of contrastive learning that requires negative samples. The authors propose a variation of contrastive loss that removes the terms contrasting negative pairs from the loss and adds a new regularization term that is unbiased. Experimental results on small image datasets support the claim. The proposed loss comes with theoretical analysis, where the error for downstream tasks is proven to be bounded.

**Strengths:**

The proposed method is paired with a theoretical upper bound of the error for downstream tasks.

**Weaknesses:**

- Overall writing/organization of the paper could be improved, especially about the math part. The paper is generally not easily understandable.

- For example, <A,B>_F should be defined when first introduced.

- While G appears to be crucial throughout the paper, G is never explicitly defined, such that the overall flow of the paper is not easily understandable.

- While authors discuss biased/unbiased populations, there is no clear explanation on how exactly they are biased/unbiased.

- Experimental results are limited to small image datasets and the performance gain is overall marginal.

- L400: isn't one of the asterisk a typo?

**Questions:**

Please address concerns above.

---

> ### Author Response · Authors · 2024-11-22
> **Reply to Reviewer dVpK (Part I)**
>
> We thank you for your thorough review of our manuscript and for your constructive suggestions. We have revised our manuscript by incorporating all of your suggestions. The revised version is much strengthened. Our point-by-point responses to your comments are given below.
> > **W1**. Overall writing/organization of the paper could be improved, especially about the math part. The paper is generally not easily understandable.
>
> - We appreciate your feedback and have made revisions to the manuscript to improve clarity and help you better understand our paper.
> - Briefly speaking, contrastive learning consists of three critical parts: data augmentation, representation alignment, denoted by $\mathcal{L}\_{\mathrm{align}}(\mathcal{f})$, and the design ensuring that different class clusters are separated from each other, measured by $\mu_s(i)^T \mu_s(j)$ in our framework. The last part is the most important term, though it cannot be directly optimized because there is no information about the annotation of the latent class in the upstream task. An alternative approach to achieve this goal is to find an appropriate loss function, which serves as an upper bound for $\mu_s(i)^T \mu_s(j)$ but does not rely on annotation information. Inspired by [10], the following population objective can satisfy our requirement. $$\mathcal{L}\_{\mathrm{div}}(f) = \Big\Vert \mathbb{E}\_{x}\mathbb{E}\_{x\_1, x\_2 \in \mathcal{A}(x)}[f(x\_1)f(x\_2)^T] - I\_{d^*}\Big\Vert\_F^2.$$ We built the initial form of our population risk as  $$f^* \in \arg\min\_{f}\mathcal{L}(f) = \mathcal{L}\_{\mathrm{align}}(f) + \lambda \cdot \mathcal{L}\_{\mathrm{div}}(f)$$
>     Nevertheless, while conducting theoretical analysis at the sample level with relatively mild conditions comparing [11], it is critical to determine the unbiased version of this loss, as current learning theory methods require that the sample-level loss corresponds to an unbiased estimator of the population-level loss.
>     $$
>     \mathbb{E}\_{D\_s}[\widehat{\mathcal{L}}\_{\mathrm{div}}(f)] = \mathcal{L}\_{\mathrm{div}}(f),
>     $$
>     where $D\_s$ is augmented dataset.
>
>     As you can see, directly discretizing the expectation inside the Frobenius norm leads to a biased estimator, i.e.,
>     $$
>     \mathbb{E}\_{D\_s}\Big\Vert \frac{1}{n\_s}\sum\_{i=1}^{n\_s}f(x^{(i)}\_1)f(x^{(i)}\_2)^T] - I\_{d^*}\Big\Vert\_F^2 \neq \Big\Vert \mathbb{E}\_{x}\mathbb{E}\_{x\_1, x\_2 \in \mathcal{A}(x)}[f(x\_1)f(x\_2)^T] - I\_{d^*}\Big\Vert\_F^2,
>     $$
>     which is our "unbiased" meaning.
>     This motivated us to alter $\mathcal{L}\_{\mathrm{div}}(f)$ into adversarial form, that is
>     $$
>     f^* \in \min\_{f}\sup\_{G \in \mathcal{G}(f)}\mathcal{L}(f, G) = \mathcal{L}\_{\mathrm{align}}(f) + \lambda \cdot \langle\mathbb{E}\_{x}\mathbb{E}\_{x\_1, x\_2 \in \mathcal{A}(x)}[f(x\_1)f(x\_2)^T] - I_{d^*}, G\rangle_F,
>     $$
>     under this transformation, the corresponding adversarial loss at sample level is
>     $$
>     \hat{f}\_{n\_s} \in \min_{f \in \mathcal{F}}\sup\_{G \in \widehat{\mathcal{G}}(f)}\widehat{\mathcal{L}}(f, G) = \widehat{\mathcal{L}}\_{\mathrm{align}}(f) + \lambda \cdot \langle\frac{1}{n\_s}\sum\_{i=1}^{n\_s}f(x^{(i)}_1)f(x^{(i)}_2)^T - I\_{d^*}, G\rangle\_F,
>     $$By the linearity of Forbenius inner product, it is easy to see $$\mathbb{E}\_{D\_s}[\widehat{\mathcal{L}}(f,G)] = \mathcal{L}(f, G)$$
>     for any fixed $G$, combining this property with the new decomposition method proposed by us, we build an end-to-end theoretical guarantee in the transfer learning setting to provide an explanation for few shot learning. And using an alternative optimization method to optimize this loss is natural.
>
> > **W2**. For example, <A,B>_F should be defined when first introduced.
> - We have clarified its definition in the revised manuscript when first introduced.
>
> > **W3**. While G appears to be crucial throughout the paper, G is never explicitly defined, such that the overall flow of the paper is not easily understandable.
> - $G$ is a matrix variable of dimension $d^* \times d^*$ whose value may vary from place to place. Please see line 231, equation (6), line 368, and so on, as indicated in the previous manuscript.
>
> > **W4**. While authors discuss biased/unbiased populations, there is no clear explanation on how exactly they are biased/unbiased.
> - Sorry for the confusion. The term 'unbiased' indicates that $\mathbb{E}\_{D\_s}[\widehat{\mathcal{L}}(f, G)] = \mathcal{L}(f, G)$ for any fixed $G$, where $D\_s$ is the upstream augmented dataset.  Otherwise, it is biased if $\mathbb{E}\_{D\_s}[\widehat{\mathcal{L}}(f, G)] \neq \mathcal{L}(f, G)$.

---

> ### Author Response · Authors · 2024-11-22
> **Reply to Reviewer dVpK (Part II)**
>
> > **W5**. Experimental results are limited to small image datasets and the performance gain is overall marginal.
>
> - We have updated our experimental comparison according to the other reviews' requirements.
>
>     | Method                                         | CIFAR-10 linear | CIFAR-10 5-NN | CIFAR-100 linear | CIFAR-100 5-NN | Tiny ImageNet linear | Tiny ImageNet 5-NN |
>     |------------------------------------------------|-----------------|---------------|------------------|----------------|----------------------|--------------------|
>     | SimCLR                       | 91.80           | 88.42         | 66.83            | 56.56          | 48.84                | 32.86              |
>     | BYOL                        | 91.73           | 89.45         | 66.60            | 56.82          | **51.00**            | 36.24              |
>     | W-MSE 2                      | 91.55           | 89.69         | 66.10            | 56.69          | 48.20                | 34.16              |
>     | W-MSE 4                      | 91.99           | 89.87         | 67.64            | 56.45          | 49.20                | 35.44              |
>     | BarlowTwins  | 87.76           | 86.66         | 61.64            | 55.94          | 41.80                | 33.60              |
>     | VICReg                            | 86.76           | 83.70         | 57.13            | 44.63          | 40.04                | 30.46              |
>     | Haochen22  | 86.53           | 84.20         | 59.68            | 49.26          | 35.80                | 20.36              |
>     | **ACT (our repro.)**                           | **92.11**       | **90.01**     | **68.24**        | **58.35**      | 49.72                | **36.40**          |
>
>     Our results indicate ACT significantly outperform Haochen22 [1], VICReg[2], BarlowTwins[3] and better than SimCLR[4], BYOL[5], W-MSE[6].
>
>     Based on our experience, for most theoretically-based contrastive learning methods, such as [7]-[9], using small to medium-sized datasets is sufficient to demonstrate the effectiveness of the method. Of course, complementing the experiments with ImageNet would be ideal, but it would require significantly more time. Regarding the comment on our performance gain being marginal, we displayed some experimental results from previous papers published in ICLR, ICML, and NeurIPS, as follows:
>
>     ### Whitening for Self-Supervised Representation Learning. ICML 2021.
>
>     | Method    | CIFAR-10 (Linear) | CIFAR-10 (5-NN) | CIFAR-100 (Linear) | CIFAR-100 (5-NN) | STL-10 (Linear) | STL-10 (5-NN) | Tiny ImageNet (Linear) | Tiny ImageNet (5-NN) |
>     |-----------|-------------------|-----------------|--------------------|------------------|-----------------|---------------|------------------------|----------------------|
>     | SimCLR    | 91.80             | 88.42           | 66.83              | 56.56            | 90.51           | 85.68         | 48.84                  | 32.86                |
>     | BYOL      | 91.73             | 89.45           | 66.60              | **56.82**        | **91.99**       | **88.64**     | **51.00**              | **36.24**            |
>     | W-MSE 2   | 91.55             | 89.69           | 66.10              | 56.69            | 90.36           | 87.10         | 48.20                  | 34.16                |
>     | W-MSE 4   | **91.99**         | **89.87**       | **67.64**          | 56.45            | 91.75           | 88.59         | 49.22                  | 35.44                |
>
>
>     ### Provable Guarantees for Self-Supervised Deep Learning with Spectral Contrastive Loss. NeurIPS 2021 oral.
>
>     | Datasets                | CIFAR-10 (200 epochs) | CIFAR-10 (400 epochs) | CIFAR-10 (800 epochs) | CIFAR-100 (200 epochs) | CIFAR-100 (400 epochs) | CIFAR-100 (800 epochs) | Tiny-ImageNet (200 epochs) | Tiny-ImageNet (400 epochs) | Tiny-ImageNet (800 epochs) |
>     |-------------------------|-----------------------|-----------------------|-----------------------|------------------------|------------------------|------------------------|----------------------------|----------------------------|----------------------------|
>     | **SimCLR (repro.)**      | 83.73                 | 87.72                 | 90.60                 | 54.74                  | 61.05                  | 63.88                  | **43.30**                  | **46.46**                  | 48.12                      |
>     | **SimSiam (repro.)**     | 87.54                 | **90.31**             | 91.40                 | 61.56                  | 64.96                  | 65.87                  | 34.82                      | 39.46                      | 46.76                      |
>     | **Ours**                 | **88.66**             | 90.17                 | **92.07**             | **62.45**              | **65.82**              | **66.18**              | 41.30                      | 45.36                      | **49.86**                  |

---

> ### Author Response · Authors · 2024-11-22
> **Reply to Reviewer dVpK (Part III)**
>
> - ### Non-negative Contrastive Learning. ICLR 2024.
>
>     | Method | CIFAR-100 (LP) | CIFAR-100 (FT) | CIFAR-10 (LP) | CIFAR-10 (FT) | ImageNet-100 (LP) | ImageNet-100 (FT) |
>     |--------|----------------|----------------|---------------|---------------|-------------------|-------------------|
>     | CL     | $58.6 \pm 0.2$  | $72.6 \pm 0.1$ | $87.6 \pm 0.2$| $92.3 \pm 0.1$| $68.7 \pm 0.3$    | $77.3 \pm 0.5$    |
>     | NCL    | $59.7 \pm 0.4$  | $73.0 \pm 0.2$ | $87.8 \pm 0.2$| $92.6 \pm 0.1$| $69.4 \pm 0.3$    | $79.2 \pm 0.4$    |
>
>
>     Based on the above examples, we are confident that our experimental improvements are valuable. At the very least, they should not be seen as a weakness, not to mention that we have a solid theoretical foundation.
>
> > **W6**. L400: isn't one of the asterisk a typo?
> - Thanks for pointing this typo. We have fixed it.
>
> > **Summary**.
> - We greatly appreciate the reviewer’s feedback. We hope that the revised manuscript will help the reviewer better understand our work. If the reviewer has more specific suggestions or questions, we would be happy to receive further feedback at any time.
>
> > **Reference**
>
> [1] Beyond Separability: Analyzing the Linear Transferability of Contrastive Representations to Related Subpopulations. NeurIPS 2022.
>
> [2] VICReg: Variance-invariance-covariance regularization for self-supervised learning. ICLR 2022.
>
> [3] Barlow twins: Self-supervised learning via redundancy reduction. ICML 2021.
>
> [4] A Simple Framework for Contrastive Learning of Visual Representations. ICML 2020.
>
> [5] Bootstrap your own latent: A new approach to self-supervised Learning. NeurIPS 2020.
>
> [6] Whitening for Self-Supervised Representation Learning. ICML 2021.
>
> [7] Non-negative Contrastive Learning. ICLR 2024.
>
> [8] Adversarial Examples Are Not Real Features. NeurIPS 2023.
>
> [9] Understanding the Role of Equivariance in Self-supervised Learning. NeurIPS 2024.
>
> [10] Towards the generalization of contrastive self-supervised learning. ICLR 2023.
>
> [11] A theoretical study of inductive biases in contrastive learning. ICLR, 2023.

---

> ### Comment · Reviewer_dVpK · 2024-11-25
> **Additional comments**
>
> Thank you for your response. However, the paper is still not so understandable when reading it in the casual order. Please read your manuscript carefully, without looking ahead, and check if it is understandable without iterating multiple times. Below I leave additional comments:
>
> 1. For example, I asked the definition of G, which is first introduced in L55 in the revision. However, your response is just "(it) is a matrix variable" and redirected me to L231. Even this is not the definition of G. Again, what is the definition of G, and how it is interpreted?
>
> 2. Around L56, I expected motivative statements to introduce R(f,G) and its interpretation, but the revised manuscript simply states the math expression only and saying it is a novel regularization term *"to prevents model collapse"*, which requires further elaboration.
>
> 3. Apart from this, I agree with Reviewer Gv21 that the advantage of the proposed approach over non-contrastive methods like SimSiam, Barlow Twins, and VICReg is not clear. Regarding your answer "BarlowTwins[5] and VICReg[3] only introduce their loss at the sample level, whereas its population loss is unknown, leading to the difficulty in theoretical understanding,"
> what makes the population loss better than the sample-level loss for theoretical understanding? There are several theoretical works on non-contrastive learning such as [Zhang et al.] and [Balestriero and LeCun], where some of them connect contrastive and non-contrastive learning methods.
>
> [Zhang et al.] How Does SimSiam Avoid Collapse Without Negative Samples? A Unified Understanding with Self-supervised Contrastive Learning. ICLR 2022.
>
> [Balestriero and LeCun] Contrastive and Non-Contrastive Self-Supervised Learning Recover Global and Local Spectral Embedding Methods. NeurIPS 2022.
>
> 4. Could you compare the efficiency of compared methods, in terms of the memory and training time? It seems the proposed method requires multiple rounds of updates, which results in longer training time than others. If the comparison was not fair, you may reduce the batch size / number of iterations for the proposed method to match the training time with compared methods.

---

> ### Author Response · Authors · 2024-11-26
> **Reply to the additional comments of  Reviewer dVpK (Part I)**
>
> > Thank you for your response. However, the paper is still not so understandable when reading it in the casual order. Please read your manuscript carefully, without looking ahead, and check if it is understandable without iterating multiple times.
> - Thank you for your feedback. We will do our best to revise the paper to make it clearer. The intuition behind the regularization term has been added to the introduction and Appendix A.
>
> > For example, I asked the definition of G, which is first introduced in L55 in the revision. However, your response is just "(it) is a matrix variable" and redirected me to L231. Even this is not the definition of G. Again, what is the definition of G, and how it is interpreted?
> - $G$ is a matrix variable of dimension $d^* \times d^*$. We have clarify it  in our paper.
>
> > Around L56, I expected motivative statements to introduce R(f,G) and its interpretation, but the revised manuscript simply states the math expression only and saying it is a novel regularization term "to prevents model collapse", which requires further elaboration.
> - Thanks for your suggestion. We have updated it in line 52-56 of our paper as follows.
> >The core idea of ACT is the introduction of a novel regularization term that encourages the separation of category centers within the latent space, thereby improving classification accuracy in downstream tasks. Moreover, this regularization term incorporates an unbiased sample version, enabling rigorous theoretical analysis.
>
> - Further interpretation for why our regularization term works is minimizing our regularization term can enlarge the angles between the centers of different categories in  the representation space. In the revised manuscript, we've provided more details about it in additional Appendix A.
> - In brief, contrastive learning utilizes data augmentation to construct the loss function (specifically, the first term in our loss) that aligns representations of the same class. However, to avoid trivial solutions, an additional regularization term is necessary to ensure that clusters representing different classes are well-separated. We measure this separation using the angles between the centroids of different classes. While these angles are ideal for quantifying separation, they cannot be directly optimized because the latent class annotations are unavailable in the upstream task. As an alternative, we propose finding an appropriate computable loss function that serves as an upper bound for these angles, effectively achieving the desired separation.
> Denote $$\mathcal{L}\_{\mathrm{div}}(f) = \Big\Vert \mathbb{E}\_{x}\mathbb{E}\_{x\_1, x\_2 \in \mathcal{A}(x)}[f(x\_1)f(x\_2)^\top] - I\_{d^*}\Big\Vert\_F^2.$$ It can severs as a regularization term since in Lemma B.4 in our paper, we can show$$
> \mu\_s(i)^{\top}\mu\_s(j)  \lesssim \Big\Vert \mathbb{E}\_{x}\mathbb{E}\_{x\_1,x\_2\in\mathcal{A}(x)}[f(x\_1)f(x\_2)^{\top}]-I\_{d^*}\Big\Vert\_F,$$ where $\mu\_s(i)=\mathbb{E}\_{x\in C\_s(i)}\mathbb{E}\_{x^\prime\in\mathcal{A}(x)}[f(x^\prime)]$ is the center of the latent class $i$. This implies that a lower value of the regularization term leads the separation between different categories' center, thereby benefits classification in downstream tasks.
>
>     At the sample level, one can use $\widehat{\mathcal{L}}\_{\mathrm{div}}(f) = \Big\Vert \frac{1}{n\_s}\sum\_{i=1}^{n\_s}f(x^{(i)}\_1)f(x^{(i)}\_2)^\top - I\_{d^*}\Big\Vert\_F^2$ to estimate  $\mathcal{L}\_{\mathrm{div}}(f)$. However, this lead to a bias loss, i.e., $$\mathbb{E}\_{D\_s}[\widehat{\mathcal{L}}\_{\mathrm{div}}(f)] \neq \mathcal{L}\_{\mathrm{div}}(f),$$where $D\_s$ is augmented dataset.
>
>    This bias is caused by the non-commutativity of the expectation and the Frobenius norm. To overcome this we can reformulate it as an equivalent form $$\mathcal{L}\_{\mathrm{div}}(f)=\sup\_{G \in \mathcal{G}(f)}\mathcal{R}(f,G):= \langle\mathbb{E}\_{x}\mathbb{E}\_{x\_1, x\_2 \in \mathcal{A}(x)}[f(x\_1)f(x\_2)^\top] - I\_{d^*}, G\rangle\_F.  $$The counterpart of $\mathcal{R}(f, G)$ at the sample level is $$\widehat{\mathcal{R}}(f, G) = \langle\frac{1}{n\_s}\sum\_{i=1}^{n\_s}f(x^{(i)}\_1)f(x^{(i)}\_2)^\top - I\_{d^*}, G\rangle\_F.$$ We can see that $\mathbb{E}\_{D\_s}[\widehat{\mathcal{R}}(f, G)] = \mathcal{R}(f, G)$ for any fixed $G$ due to the linearity of Frobenius inner product, combining this property with the new decomposition method proposed by us, we build an end-to-end theoretical guarantee in the transfer learning setting to provide an explanation for few shot learning. And using an alternative optimization method to optimize this loss is natural.

---

> ### Author Response · Authors · 2024-11-26
> **Reply to the additional comments of Reviewer dVpK (Part II)**
>
> > Apart from this, I agree with Reviewer Gv21 that the advantage of the proposed approach over non-contrastive methods like SimSiam, Barlow Twins, and VICReg is not clear. Regarding your answer "BarlowTwins[5] and VICReg[3] only introduce their loss at the sample level, whereas its population loss is unknown, leading to the difficulty in theoretical understanding," what makes the population loss better than the sample-level loss for theoretical understanding? There are several theoretical works on non-contrastive learning such as [Zhang et al.] and [Balestriero and LeCun], where some of them connect contrastive and non-contrastive learning methods.
> - Our theoretical analysis is to explore the statistical convergence rate, thus the population loss is necessary.
> - In this study, "theoretical understanding" refers to the statistical convergence rate of the method, while the theoretical findings in [Zhang et al.] and [Balestriero and LeCun] elucidate the relationship between contrastive and non-contrastive learning approaches.
> - The statistical convergence rate plays a pivotal role in learning theory, particularly within the realm of self-supervised learning. This is because the convergence rate indicates how the error diminishes towards zero as the number of pretraining unlabeled samples and the quantity of labeled samples in downstream tasks increase, respectively. Additionally, it can offer theoretical insights into determining the appropriate size of deep neural networks.
> - The analysis of statistical convergence rate necessitates consideration of the population loss. Specifically, the statistical convergence rate comprises the approximation error and the generalization error. The generalization error quantifies the disparity between the population loss and the loss at the sample level. Given that the population loss for models like Barlow Twins and VICReg is not clearly defined, a comprehensive analysis of their generalization capabilities cannot be established, thereby rendering the statistical convergence rate unavailable.
>
> >Could you compare the efficiency of compared methods, in terms of the memory and training time? It seems the proposed method requires multiple rounds of updates, which results in longer training time than others. If the comparison was not fair, you may reduce the batch size / number of iterations for the proposed method to match the training time with compared methods.
> - Our method does not require multiple rounds of updates, which would result in longer training times, as the inner maximization problem has an explicit solution. We have compared the efficiency of the methods discussed in the paper as follows.
> | Method       | Memory (MiB) | Training Time (seconds per epoch) |
> |--------------|--------------|-----------------------------------|
> | SimCLR       | 3275         | 31                                |
> | BYOL         | 3431         | 32                                |
> | WMSE         | 3125         | 26                                |
> | BarlowTwins  | 3297         | 41                                |
> | VICReg       | 3275         | 27                                |
> | Haochen22    | 3119         | 27                                |
> | ACT          | 3119         | 27                                |

---

> ### Author Response · Authors · 2024-11-30
> **Official Comment by Authors**
>
> With the discussion period extended until December 2nd, we again appreciate the contributions of all reviewers.
>
> If there are any questions so far, please feel free to contact us.
>
> Warm regards,
>
> The Authors of Submission3353

---

### Official Review · Reviewer_Gv21 · 2024-11-02

**Soundness:** 1
**Presentation:** 2
**Contribution:** 1
**Rating:** 3
**Confidence:** 3

**Summary:**

The paper proposes a contrastive learning loss that does not require negative sampling, with the objective framed as a min-max problem. The authors state that the empirical loss of the proposed method is an unbiased estimator of the population loss and provide a theoretical analysis guaranteeing its performance. Experiments are conducted, comparing the proposed method with some existing techniques and demonstrating improvements.

**Strengths:**

- The authors state that the proposed loss function does not suffer from bias between the empirical loss and the population loss.
- Theoretical analysis is conducted on the proposed loss.

**Weaknesses:**

- The authors seem to claim that one advantage of the proposed method is that it does not require negative sampling. However, there are already many existing techniques that do not require negative sampling, such as Barlow Twins [3], VICREG [4], and others referenced in [5]. The paper lacks a detailed discussion and comparison with these methods, and there is no clear advantage demonstrated over them, nor any comparison in terms of performance in the experiments section.

- Another point of emphasis in this paper appears to be that the proposed loss function is an unbiased estimator of the population loss. However, this is not explicitly and clearly presented. For example, (1) is there any theoretical analysis directly showing that this proposed loss is unbiased? (2) Is there a theoretical analysis showing that all existing losses, including those mentioned in the previous points, are biased, and why?

- There needs to be a more intuitive explanation of how and why the regularization term is designed in this specific way.

- More specifically, looking at the regularization term in equations (1) or (2), to solve the inner max problem, it seems that we could set $ G = \mathbb{E}[f(x_1)f(x_2)^T] - I_d $. Then the regularization reduces simply to the norm of $ \mathbb{E}[f(x_1)f(x_2)^T] - I_d $, so I don’t see why it is necessary to frame it as a min-max problem or as some form of adversarial training. Moreover, if the above is true, after converting the regularization term, it becomes the same as the one analyzed in [2] (in fact, the entire loss is also the same; and also the same as the one proposed by [1] as mentioned by [2]), which is also very similar to the well-known Barlow Twins [3]. In this sense, I don’t see any novelty in the proposed method.

- To motivate the need for proposing an unbiased loss, the authors mention that [2] requires too strong an assumption, although it has already performed sample-level analysis. Could this point be elaborated further? Also, given the similarity between the proposed loss and the loss analyzed in [2], how does this paper address the issues present in [2]?

- According to [5], there is some unification between contrastive and non-contrastive losses. It would be beneficial to discuss the connection and differences compared to prior work in the context of [5], so that the uniqueness of the proposed loss can be more clearly understood.

- Regarding the experiments, beyond the lack of many baseline comparisons mentioned earlier, the reported improvements also seem marginal.



[1] HaoChen, Jeff Z., et al. "Provable guarantees for self-supervised deep learning with spectral contrastive loss." Advances in Neural Information Processing Systems 34 (2021): 5000-5011.

[2] HaoChen, Jeff Z., and Tengyu Ma. "A theoretical study of inductive biases in contrastive learning." arXiv preprint arXiv:2211.14699 (2022).

[3] Zbontar, Jure, et al. "Barlow twins: Self-supervised learning via redundancy reduction." International conference on machine learning. PMLR, 2021.

[4] Bardes, Adrien, Jean Ponce, and Yann LeCun. "Vicreg: Variance-invariance-covariance regularization for self-supervised learning." arXiv preprint arXiv:2105.04906 (2021).

[5] Garrido, Quentin, et al. "On the duality between contrastive and non-contrastive self-supervised learning." arXiv preprint arXiv:2206.02574 (2022).

**Questions:**

See the questions in the Weaknesses section.

---

> ### Author Response · Authors · 2024-11-22
> **Reply to Reviewer Gv21 (Part I)**
>
> We thank you for your thorough review of our manuscript and for your constructive suggestions. We have revised our manuscript by incorporating all of your suggestions. The revised version is much strengthened. Our point-by-point responses to your comments are given below.
> >**W1**. The authors seem to claim that one advantage of the proposed method is that it does not require negative sampling. However, there are already many existing techniques that do not require negative sampling, such as Barlow Twins [3], VICREG [4], and others referenced in [5]. The paper lacks a detailed discussion and comparison with these methods, and there is no clear advantage demonstrated over them, nor any comparison in terms of performance in the experiments section.
> - Thank you for your feedback. Indeed, many techniques do not rely on negative sampling. However, for example, BarlowTwins[5] and VICReg[3] only introduce their loss at the sample level, whereas its population loss is unknown, leading to the difficulty in theoretical understanding. For Haochen22[2], we prove that its sample-level loss introduces bias between the population and sample loss, which would influence the numerical results shown below, while bringing some theoretical challenges discussed in **W5**.
> - We have also completed the comparison as requested, which is presented in the following table. All results related to BarlowTwins are taken directly from [3]. For Haochen22 [2] and VICReg [3], the corresponding results over these datasets were not available in the original papers. We made our best effort to implement both methods independently based on the available resources, including a GitHub repository. Additionally, the repository at https://github.com/AsafShul/VICReg_selfsupervised_representation_learning claims that their implementation of VICReg achieves an accuracy of only ~85.5% in linear evaluation.
>     | Method          | CIFAR-10 (linear) | CIFAR-10 (5-NN) | CIFAR-100 (linear) | CIFAR-100 (5-NN) | Tiny ImageNet (linear) | Tiny ImageNet (5-NN) |
>     |-----------------|-------------------|-----------------|---------------------|------------------|------------------------|----------------------|
>     | BarlowTwins[1] | 87.76  | 86.66| 61.64 | 55.94 | 41.80 | 33.60 |
>     | Haochen22 (our repro.) | 86.53 | 84.20| 59.68| 49.26| 35.80 |20.36|
>     | VICReg (our repro.) |86.76|83.70 |57.13 | 44.63 | 40.04  |         30.46        |
>     | ACT             | **92.11**         | **90.01**       | **68.24**           | **58.35**        | **49.72**              | **36.40**            |
>
> >**W2**. Another point of emphasis in this paper appears to be that the proposed loss function is an unbiased estimator of the population loss. However, this is not explicitly and clearly presented. For example, (1) is there any theoretical analysis directly showing that this proposed loss is unbiased? (2) Is there a theoretical analysis showing that all existing losses, including those mentioned in the previous points, are biased, and why?
> - We first show that our proposed loss is unbiased. The population loss of ACT is
>     $$
>     \mathcal{L}(f, G) = \mathbb{E}\_{x}\mathbb{E}\_{x\_1, x\_2 \in \mathcal{A}(x)}\Vert f(x\_1) - f(x\_2)\Vert\_2^2 + \lambda \cdot \langle\mathbb{E}\_{x}\mathbb{E}_{x\_1, x\_2 \in \mathcal{A}(x)}[f(x\_1)f(x\_2)^T] - I\_{d^*}, G\rangle_F,
>     $$
>     and the corresponding adversarial loss at sample level is
>     $$
>     \widehat{\mathcal{L}}(f, G) = \frac{1}{n\_s}\sum\_{i=1}^{n\_s}\Vert f(x\_1^{(i)}) - f(x\_2^{(i)})\Vert\_2^2 + \lambda \cdot \langle\frac{1}{n\_s}\sum\_{i=1}^{n\_s}f(x^{(i)}\_1)f(x^{(i)}\_2)^T - I\_{d^*}, G\rangle_F,
>     $$
>     By the linearity of Forbenius inner product, it is easy to see
>     $$
>         \mathbb{E}\_{D\_s}[\widehat{\mathcal{L}}(f,G)] = \mathcal{L}(f, G)
>     $$
>     for any fixed $G$, where $D_s$ is augmented dataset.
> - To the best of our knowledge,  techniques that do not rely on negative sampling often have biased sample-level loss  or lack a well-defined population-level loss. For example, the sample-level loss in existing work such as Haochen22[2] is biased. The population loss of [2] is defined as follows
>     $$
>     \mathcal{R}(f) = \mathbb{E}\_{x}\mathbb{E}\_{x\_1, x\_2 \in \mathcal{A}(x)}\Vert f(x\_1) - f(x\_2)\Vert\_2^2 + \lambda \cdot \Big\Vert \mathbb{E}\_{x}\mathbb{E}\_{x^\prime \in \mathcal{A}(x)}[f(x^\prime)f(x^\prime)^T] - I\_{d^*}\Big\Vert\_F^2.
>     $$
>     The sample loss used in [2] is
>     $$
>     \widehat{\mathcal{R}}(f) = \frac{1}{n\_s}\sum\_{i=1}^{n\_s}\Vert f(x\_1^{(i)}) - f(x\_2^{(i)})\Vert\_2^2 + \lambda \cdot \Big\Vert \frac{1}{n\_s}\sum\_{i=1}^{n\_s}f(x^\prime\_i)f(x^\prime\_i)^T] - I\_{d^*}\Big\Vert\_F^2.
>     $$
>     Obviously $\mathbb{E}\_{D\_s}[\widehat{\mathcal{R}}(f)] \neq \mathcal{R}(f)$ since the operations of expectation and Frobenius norm can not be exchanged.
> - Barlow Twins [5] and VICReg [3] only provide the sample-level loss, and their corresponding population loss functions are unknown.

---

> ### Author Response · Authors · 2024-11-22
> **Reply to Reviewer Gv21 (Part II)**
>
> > **W3**. There needs to be a more intuitive explanation of how and why the regularization term is designed in this specific way.
>
> - In previous literature, it is common to include a regularization term in the self-supervised loss, which helps separate the centers of different categories to prevent model collapse. Model collapse can be formulated as $\mu\_s(i)^T\mu\_s(j) = 1$ for $i \neq j$, where $\mu\_s(i) = \mathbb{E}\_{x \in C_s(i)}\mathbb{E}\_{x^\prime \in \mathcal{A}(x)}[f(x^\prime)]$. Therefore, it is necessary to find a method that ensures the directions of  $\mu\_s(i)$ and $\mu\_s(j)$ are not the same. In the Lemma A.4 of our paper, we have shown
>   $$
>      \mu_s(i)^T\mu_s(j) \lesssim \Big\Vert \sum\_{k=1}^{K}p\_s(k)\mu\_s(k)\mu\_s(k)^T - I\_{d^*}\Big\Vert\_F \lesssim \Big\Vert \mathbb{E}\_{x}\mathbb{E}\_{x\_1, x\_2 \in \mathcal{A}(x)}[f(x\_1)f(x\_2)^T] - I\_{d^*}\Big\Vert\_F, \qquad (1)
>   $$
> Hence $\Big\Vert \mathbb{E}\_{x}\mathbb{E}\_{x\_1, x\_2 \in \mathcal{A}(x)}[f(x\_1)f(x\_2)^T] - I\_{d^*}\Big\Vert\_F$ is used to be a regularization term. $(1)$  implies that a lower value of the regularization term leads the separation between different categories' center, thereby benefits classification in downstream tasks.
> - To help the readers understand $\mu_s(i)^T\mu_s(j) \lesssim \Big\Vert \sum\_{k=1}^{K}p\_s(k)\mu\_s(k)\mu\_s(k)^T - I\_{d^*}\Big\Vert\_F$ better, without loss of generality, let's consider a toy case with $K = 2, d^* = 2, p\_s(1) = p\_s(2) = \frac{1}{2}$. Since the rank of $\mu\_s(k)\mu\_s(k)^T$ is $1$ for any $k = 1, 2$, it is obvious that $\mu\_s(1) = [\sqrt{2}, 0], \mu\_s(2) = [0, \sqrt{2}]$, which are orthogonal to each other, is a minimizer of RHS. Thereby, we can achieve a representation space where the data distribution is well separated.
>
> **W4**. More specifically, looking at the regularization term in equations (1) or (2), to solve the inner max problem, it seems that we could set $G = \mathbb{E}[f(x\_1)f(x_2)^T] - I\_{d^*}$. Then the regularization reduces simply to the norm of $\mathbb{E}[f(x\_1)f(x\_2)^T] - I\_d$, so I don’t see why it is necessary to frame it as a min-max problem or as some form of adversarial training.
> Moreover, if the above is true, after converting the regularization term, it becomes the same as the one analyzed in [2] (in fact, the entire loss is also the same; and also the same as the one proposed by [1] as mentioned by [2]), which is also very similar to the well-known Barlow Twins [3]. In this sense, I don’t see any novelty in the proposed method.
>
> - As you said, the population loss of ACT is equivalent to $$ \mathbb{E}\_{x}\mathbb{E}\_{x\_1, x\_2 \in \mathcal{A}(x)}\Vert f(x\_1) - f(x\_2)\Vert\_2^2 + \Big\Vert \mathbb{E}\_{x}\mathbb{E}\_{x\_1, x\_2 \in \mathcal{A}(x)}[f(x\_1)f(x\_2)^T] - I\_{d^*}\Big\Vert\_F^2$$
>     Directly discretizing the expectation in Frobenius norm, as [2] and [3], obtains
>     $$
>     \widehat{\mathcal{L}}(f) =  \frac{1}{n\_s}\sum\_{i=1}^{n\_s}\Vert f(x\_1^{(i)}) - f(x\_2^{(i)})\Vert\_2^2 + \lambda \cdot \Big\Vert \frac{1}{n\_s}\sum\_{i=1}^{n\_s}f(x\_1^{(i)})f(x\_2^{(i)})^T] - I\_{d^*}\Big\Vert\_F^2,
>     $$
>     which will introduce a bias illustrated as **W2**. Due to this bias, the representation learned by [2] and [3] is not close to the population loss minimizer. Specifically, when trained on mini-batch data, the bias can be amplified by the limited sample size in each mini-batch.
> - However, the representation learned by ACT non-asymptotically converges to the minimizer of population loss, as proven by us. Specifically, reformulating it as a min-max problem results in an unbiased adversarial loss at the sample level (equation (5) in the manuscript) and an alternating optimization algorithm, benefiting both practical performance and theoretical analysis. ACT can outperform the methods you mentioned, as shown in the table in the response to **W1**. From a theoretical perspective, such an unbiased adversarial loss can help us perform an end-to-end analysis in the misspecification scenario, revealing the impact of both upstream and downstream sample sizes on the downstream testing error.
>
> - Barlow Twins[6] only provide its sample-level form, but its population loss is unknown, leading to difficulties in theoretical understanding.
> - The second term of the loss in Haochen22 [2] can be regarded as a special version of ACT at the population level, with the constraint $x\_1 = x\_2$. However, without an adversarial formulation, the biased sample loss forces them to make strong assumptions in order to conduct analysis at the sample level, even under a well-specified setting. See the response of **W5** for more details. Furthermore, the adversarial formulation leads to a entire different representation at the sample level, as discussed in Remark 2.1 of our paper.

---

> ### Author Response · Authors · 2024-11-22
> **Reply to Reviewer Gv21 (Part III)**
>
> >**W5**. To motivate the need for proposing an unbiased loss, the authors mention that [2] requires too strong an assumption, although it has already performed sample-level analysis. Could this point be elaborated further? Also, given the similarity between the proposed loss and the loss analyzed in [2], how does this paper address the issues present in [2]?
>
> - The mainly intractable assumption in [6] is its Assumption 4.2 and Assumption 4.4.
> - Assumption 4.2 of [6] assumes the existence of a neural network $f_{\mathrm{eig}}$ that almost vanishes the upstream risk $\mathcal{L}\_{\mathrm{align}}(f) + \lambda \cdot \Vert\mathbb{E}\_{x}\mathbb{E}\_{x^\prime \in \mathcal{A}(x)}[f\_{\mathrm{eig}}(x^\prime)f\_{\mathrm{eig}}(x^\prime)^T] - I\_{d^*}\Vert\_F^2$. To demonstrate its rationality, one needs to construct a network with specific parameters satisfying this requirement, which is generally intractable.
> - In contrast, based on unbiased property, we demonstrate the existence of a measurable function that can vanish our loss by accounting for additional approximation error, necessitating an extension of the well-specified setting to a misspecified setting.
> - Moreover, the most important problem in self-supervised transfer learning theory is explaining why the representation learned by the upstream task helps the learning of the downstream task. However, [6] directly avoid this problem by their Assumption 4.4. In contrast, our study surpasses the current body of literature by conducting a comprehensive investigation into the impact of approximation error and generalization error during the pre-training phase on downstream test error, even though the downstream sample size is limited.
>
> >**W6**. According to [5], there is some unification between contrastive and non-contrastive losses. It would be beneficial to discuss the connection and differences compared to prior work in the context of [5], so that the uniqueness of the proposed loss can be more clearly understood.
>
> - The referenced work provides a unified framework for sample-contrastive and dimension-contrastive methods, revealing the connections between them. Building on this theoretical foundation, many new self-supervised methods have been proposed. This is undoubtedly an outstanding piece of work, offering a novel perspective on existing self-supervised methods. Our research mind map is quite similar to this work; we begin by establishing theoretical foundations, and, facing the obstacles posed by bias along the way, we propose a new discretization method. While obtaining theoretical results, we also validate its effectiveness through experiments.
>
> >**W7**. Regarding the experiments, beyond the lack of many baseline comparisons mentioned earlier, the reported improvements also seem marginal.
>
> - We have updated our experimental comparison with [2], [3], [5], as table in the response to **W1**. Our results indicate ACT significantly outperform Haochen22[2], VICReg[3], BarlowTwins[5].
>
>
> >**Summary**
> We appreciate your valuable feedback and will emphasize these points in the revised manuscript.
>
>
> >**Reference**
>
> [1] Guarding Barlow Twins Against Overfitting with Mixed Samples. arXiv preprint arXiv:2312.02151 (2023).
>
> [2] Beyond Separability: Analyzing the Linear Transferability of Contrastive Representations to Related Subpopulations. NeurIPS 2022.
>
> [3] VICReg: Variance-invariance-covariance regularization for self-supervised learning. ICLR 2022.
>
> [4] Towards the generalization of contrastive self-supervised learning. ICLR 2023.
>
> [5] Barlow twins: Self-supervised learning via redundancy reduction. ICML 2021.
>
> [6] A theoretical study of inductive biases in contrastive learning. ICLR 2023.

---

> ### Comment · Reviewer_Gv21 · 2024-11-27
>
> Thank you for the response. However, upon closer examination, the argument regarding the proposed loss being unbiased still does not seem convincing to me. To obtain meaningful representations, G should not be treated as a constant. The actual loss in consideration is the min-max loss presented in Equation 2, where G depends on f. Treating G as a constant breaks this dependency and is not how the loss is intended to function.
>
> Given that the loss is minimized in a min-max fashion, as detailed in Algorithm 1, I don't think it is appropriate to analyze the loss while assuming G is fixed.
>
> Given the above, I still don’t see any significant difference between the proposed loss and that in [2], aside from its reformulation as a min-max expression.

---

> ### Author Response · Authors · 2024-11-27
> **Reply to Reviewer Gv21 (Part IIII)**
>
> - Thank you for your closer examination and question.
> - The unbiased property refers to the loss function itself for any inputs. More specifically, $\mathbb{E}\_{D\_s}[\widehat{\mathcal{L}}(f,G)] = \mathcal{L}(f, G)$ for each $f$ and $G$ which are independent of data, where $G$ can definitely depend on $f$.
> - At first, as stated in the introduction, our minimax loss
> $$
> \max\_{G \in \mathcal{G}(f)}\mathcal{L}(f, G)= \mathbb{E}\_x\mathbb{E}\_{x\_1, x\_2 \in \mathcal{A}(x)}\big[\Vert f(x\_1) - f(x\_2)\Vert\_2^2\big]+\lambda\mathcal{R}(f, G), \qquad (I)
> $$
> is an equivalent form of
> $$
> \mathcal{L}(f) := \mathbb{E}\_{x}\mathbb{E}\_{x\_1, x\_2 \in \mathcal{A}(x)}\big[\Vert f(x\_1) - f(x\_2)\Vert\_2^2\big]+\lambda \Big\Vert \mathbb{E}\_{x}\mathbb{E}\_{x\_1, x\_2 \in \mathcal{A}(x)}[f(x\_1)f(x\_2)^{\top}] - I\_{d^*}\Big\Vert\_F^2 \qquad (II)
> $$
> Thus we only need to focus on  $(I)$. In the analysis of $(I)$, we do not treat $G$ as a constant when we discussed the unbiased property. It definitely can depend on $f$ as your said.
> - However, $(I)$ is not computable in practice, it is necessary to figure out the sample level of $(I)$ as follow
> $$\widehat{\mathcal{L}}(f, G) := \frac{1}{n\_s}\sum\_{i=1}^{n\_s}\Big[\Vert f(x^{(i)}\_1) - f(x^{(i)}\_2)\Vert^2\_2+\lambda\big\langle f(x^{(i)}\_1)f(x^{(i)}\_2)^{\top} - I\_{d^*},G\big\rangle\_{F}\Big],$$
> which is the unbiased sample version of $(I)$. The unbiasedness is pretty crucial for the analysis of generalization error $\mathcal{E}_{\mathrm{sta}}$, more details can be found in Lemma B.10. of up to date manuscript. Whereas direct discretization of $(II)$ is biased, impeding us conduct analysis. This demonstrates the difference between our sample-level loss and loss in [2].
> - We now explain the difference between our method and [2] in details. For the theoretical analysis, the sample-level loss of [2] exhibits bias, presenting a significant challenge in terms of theoretical analysis. In the perspective of algorithm, the biased sample-level loss in [2] leads to different optimization directions compared to ACT. Specifically, when trained on mini-batch data, the limited sample size in each mini-batch can amplify the bias. As a consequence, [2] has a lower accuracy compared with ACT, as shown in the experimental result.
> - Then we briefly introduce our algorithm to show the updates of $G$ and $f$ during the training. After initializing $G$ and $f$, we fix $f$ and update $G$ by (8), and then update $f$ given fixed $G$ by minimizing the loss function (9). We update $f$ and $G$ alternately in this way until the algorithm converges. Thus $G$ is not a constant in our algorithm.

---

> ### Comment · Reviewer_Gv21 · 2024-11-27
>
> Thanks for your reply. However, when transitioning from the population loss to the sample-level loss, the expectation of the sample-level loss does not equal the population loss unless $G$ is treated as a constant—which it is not. $G$ is effectively $\mathbb{E}[f(x_1)f(x_2)^T] - I_d$, which is a function of $f(x)$, the representations.
>
> In other words, based on my understanding, your argument that $\mathbb{E}_{D_s}[\widehat{\mathcal{L}}(f,G)] = \mathcal{L}(f, G)$ holds only when $G$ is independent of $f$. In reality, however, $G$ is a function of $f$.

---

> ### Author Response · Authors · 2024-11-27
> **Reply to Official Comment by Reviewer Gv21**
>
> Thanks for your comment. We will add the detailed derivation in our revisited manuscript.
>
> For each function $f$ and matrix $G$ independent of $D\_{s}$, it is follows that
> $$
> \begin{aligned}
> \mathbb{E}\_{D\_{s}}[\widehat{\mathcal{L}}(f, G)]
> & := \mathbb{E}\_{D\_{s}}\Big[\frac{1}{n\_s}\sum\_{i=1}^{n\_s}\Vert f(x^{(i)}\_1) - f(x^{(i)}\_2)\Vert^2\_2+\lambda\big\langle f(x^{(i)}\_1)f(x^{(i)}\_2)^{\top} - I\_{d^*},G\big\rangle\_{F}\Big] \\\\
> & =  \mathbb{E}\_{D\_{s}}\Big[\frac{1}{n\_s}\sum\_{i=1}^{n\_s}\Vert f(x^{(i)}\_1) - f(x^{(i)}\_2)\Vert^2\_2\Big]+\lambda\mathbb{E}\_{D\_{s}}\Big[\frac{1}{n\_s}\sum\_{i=1}^{n\_s}\big\langle f(x^{(i)}\_1)f(x^{(i)}\_2)^{\top} - I\_{d^*},G\big\rangle\_{F}\Big] \\\\
> & =  \mathbb{E}\_{D\_{s}}\Big[\frac{1}{n\_s}\sum\_{i=1}^{n\_s}\Vert f(x^{(i)}\_1) - f(x^{(i)}\_2)\Vert^2\_2\Big]+\lambda\big\langle \mathbb{E}\_{D\_{s}}\Big[\frac{1}{n\_s}\sum\_{i=1}^{n\_s}f(x^{(i)}\_1)f(x^{(i)}\_2)^{\top} \Big]- I\_{d^*},G\big\rangle\_{F} \\\\
> &= \mathbb{E}\_{x}\mathbb{E}\_{x\_1, x\_2 \in \mathcal{A}(x)}\Vert f(x\_1) - f(x\_2)\Vert\_2^2 + \lambda \langle\mathbb{E}\_{x}\mathbb{E}_{x\_1, x\_2 \in \mathcal{A}(x)}[f(x\_1)f(x\_2)^T] - I\_{d^*}, G\rangle_F \\\\
> &=\mathcal{L}(f, G),
> \end{aligned}
> $$
> where the first equality holds from the definition of the sample-level loss, the second equality follows from the linearity of the expectation, the third equality invokes the linearity of the Frobenius inner product, and the fourth equality is due to the definition of the expectation.
>
> In summary, the equality $\mathbb{E}\_{D\_s}[\widehat{\mathcal{L}}(f,G)] = \mathcal{L}(f, G)$ holds for each $f$ and $G$ independent of data, even when $G$ depends on $f$.

---

> > ### Comment · Reviewer_Gv21 · 2024-11-27
> >
> > Thanks for the reply. However, as I have mentioned before, G effectively equals $\mathbb{E}[f(x_1)f(x_2)^T] - I_d$ which depends on f(x) and consequently on the data as well. Therefore the expectation of the sample-level loss does not equal the population loss.

---

> > > ### Author Response · Authors · 2024-11-27
> > > **Reply to Official Comment by Reviewer Gv21**
> > >
> > > For the sake of completeness of content, we detail here the derivation of the bound on generalization error. The detailed derivation can be found in Lemma B.10 in our manuscript.
> > > $$
> > > \begin{aligned}
> > > \mathbb{E}\_{D\_s}\big[\vert\mathcal{L}(\widehat{f},\widehat{G}) - \widehat{\mathcal{L}}(\widehat{f},\widehat{G})\vert\big]
> > > & \leq\mathbb{E}\_{D\_s}\big[\sup\_{f \in \mathcal{NN}\_{d, d^*}(W, L, \mathcal{K}, B\_1, B\_2), G \in \widehat{\mathcal{G}}(f)}\vert\mathcal{L}(f, G) - \widehat{\mathcal{L}}(f, G)\vert\big]  \\\\
> > > & \leq \mathbb{E}\_{D\_s}\big[\sup\_{(f, G) \in \mathcal{NN}\_{d, d^*}(W, L, \mathcal{K}, B\_1, B\_2) \times \mathcal{G}\_1}\vert\mathcal{L}(f, G) - \widehat{\mathcal{L}}(f, G)\vert\big]  \\\\
> > > & \leq \mathbb{E}\_{D\_s}\big[\sup\_{(f, G) \in \mathcal{NN}\_{d, d^*}(W, L, \mathcal{K}) \times \mathcal{G}\_1}\vert\mathcal{L}(f, G) - \widehat{\mathcal{L}}(f, G)\vert\big]   \\\\
> > > & = \mathbb{E}\_{D\_s}\big[\sup\_{(\widetilde{f}, G) \in \widetilde{\mathcal{F}} \times \mathcal{G}\_1} \Big\vert\frac{1}{n\_s}\sum\_{i=1}^{n\_s}\mathbb{E}\_{D^\prime\_s}[\ell(\widetilde{f}(\tilde{x}^{\prime(i)}), G)] - \frac{1}{n\_s}\sum\_{i=1}^{n\_s}\ell(\widetilde{f}(\tilde{x}^{(i)}), G)\Big\vert\big]  \\\\
> > > & \leq \mathbb{E}\_{D\_s, D^\prime\_s}\big[\sup\_{(\widetilde{f}, G) \in \widetilde{\mathcal{F}} \times \mathcal{G}\_1} \Big\vert\frac{1}{n\_s}\sum\_{i=1}^{n\_s}\ell(\widetilde{f}(\tilde{x}^{\prime(i)}), G) - \frac{1}{n\_s}\sum\_{i=1}^{n\_s}\ell(\widetilde{f}(\tilde{x}^{(i)}), G)\Big\vert\big]  \\\\
> > > & = \mathbb{E}\_{D\_s, D^\prime\_s, \xi}\big[\sup\_{(\widetilde{f}, G) \in \widetilde{\mathcal{F}} \times \mathcal{G}\_1} \Big\vert\frac{1}{n\_s}\sum\_{i=1}^{n\_s}\xi\_i\big(\ell(\widetilde{f}(\tilde{x}^{\prime(i)}), G) - \ell(\widetilde{f}(\tilde{x}^{(i)}), G)\big)\Big\vert\big]  \\\\
> > > & \leq 2\mathbb{E}\_{D\_s, \xi}\big[\sup\_{(\widetilde{f}, G) \in \widetilde{\mathcal{F}} \times \mathcal{G}\_1}\Big\vert\frac{1}{n\_s}\sum\_{i=1}^{n\_s}\xi\_i\ell(\widetilde{f}(\tilde{x}^{(i)}), G)\Big\vert\big]  \\\\
> > > & \leq 4\sqrt{2}\Vert\ell\Vert\_{\mathrm{Lip}}\Big(\mathbb{E}\_{D\_s, \xi}\big[\sup\_{f \in \mathcal{NN}\_{d,d^*}(W, L, \mathcal{K})}\Big\vert\frac{1}{n\_s}\sum\_{i=1}^{n\_s}\sum\_{j=1}^{d^*}\xi\_{i,j,1}f\_j(x\_1^{(i)}) + \xi\_{i,j,2}f\_j(x\_2^{(i)})\Big\vert\big]  + \mathbb{E}\_{\xi}\big[\sup\_{G \in \mathcal{G}\_1}\Big\vert\frac{1}{n\_s}\sum\_{i=1}^{n\_s}\sum\_{j=1}^{d^*}\sum\_{k=1}^{d^*}\xi\_{i,j,k}G\_{jk}\Big\vert\big]\Big)  \\\\
> > > & \leq 8\sqrt{2}\Vert\ell\Vert\_{\mathrm{Lip}}\mathbb{E}\_{D\_s, \xi}\big[\sup\_{f \in \mathcal{NN}\_{d,d^*}(W, L, \mathcal{K})}\Big\vert\frac{1}{n\_s}\sum\_{i=1}^{n\_s}\sum\_{j=1}^{d^*}\xi\_{i,j,1}f\_j(x\_1^{(i)})\Big\vert\big] + 4\sqrt{2}d^*\Vert\ell\Vert\_{\mathrm{Lip}}\varrho  + 4\sqrt{2}\Vert\ell\Vert\_{\mathrm{Lip}}\mathbb{E}\_{\xi}\big[\max\_{G \in \mathcal{N}\_{\mathcal{G}\_1}(\varrho)}\Big\vert\frac{1}{n\_s}\sum\_{i=1}^{n\_s}\sum\_{j=1}^{d^*}\sum\_{k=1}^{d^*}\xi\_{i,j,k}G\_{jk}\Big\vert\big] \\\\
> > > & \leq 8\sqrt{2}\Vert\ell\Vert\_{\mathrm{Lip}}\mathbb{E}\_{D\_s, \xi}\big[\sup\_{f \in \mathcal{NN}\_{d,d^*}(W, L, \mathcal{K})}\Big\vert\frac{1}{n\_s}\sum\_{i=1}^{n\_s}\sum\_{j=1}^{d^*}\xi\_{i,j}f\_j(x\_1^{(i)})\Big\vert\big] + 4\sqrt{2}d^*\Vert\ell\Vert\_{\mathrm{Lip}}\varrho  + 4\sqrt{2}(B\_2^2 + \sqrt{d^*})\Vert\ell\Vert\_{\mathrm{Lip}}\sqrt{\frac{2\log \big(2\vert\mathcal{N}\_{\mathcal{G}\_1}(\varrho)\vert\big)}{n\_s}}\\\\
> > > & \leq 8\sqrt{2}d^*\Vert\ell\Vert\_{\mathrm{Lip}}\mathbb{E}\_{D\_s, \xi}\big[\sup\_{f \in \mathcal{NN}\_{d,1}(W, L, \mathcal{K})}\Big\vert\frac{1}{n\_s}\sum\_{i=1}^{n\_s}\xi\_if(x\_1^{(i)})\Big\vert\big] + 4\sqrt{2}d^*\Vert\ell\Vert\_{\mathrm{Lip}}\varrho  + 4\sqrt{2}(B\_2^2 + \sqrt{d^*})\Vert\ell\Vert\_{\mathrm{Lip}}\sqrt{\frac{2\log \big(2(\frac{3}{(B\_2^2 + \sqrt{d^*})\varrho})^{(d^*)^2}\big)}{n\_2}} \\\\
> > > & \lesssim \frac{\mathcal{K}\sqrt{L}}{\sqrt{n\_s}} + \sqrt{\frac{\log n\_s}{n\_s}}.
> > > \end{aligned}
> > > $$
> > > Limited to lack of space, please refer to Lemma B.10 of the manuscript for more details。

---

> ### Author Response · Authors · 2024-11-27
> **Reply to Official Comment by Reviewer Gv21**
>
> Thanks for your comment.
>
> - In the context of statistics and machine learning, we say a sample-level loss $\widehat{\mathcal{L}}(f,G)$ is unbiased if its expectation is equal to the population loss $\mathcal{L}(f,G)$ for each fixed $f$ and $G$, which are independent of the data $D\_{s}$. In this sense, our sample loss is unbiased, while the sample loss in [2] is biased.
>
> - Your interest lies on $\mathbb{E}\_{D\_{s}}[\widehat{\mathcal{L}}(\widehat{f},\widehat{G})]$, where both $\widehat{f}$ and $\widehat{G}$ depend on the data $D\_{s}$. In fact, the expectation of $\mathbb{E}\_{D\_{s}}[\widehat{\mathcal{L}}(\widehat{f},\widehat{G})]$ is typically not equal to $\mathcal{L}(\widehat{f},\widehat{G})$ in machine learning tasks, which does not mean the method is biased. How to reveal the difference between them is a crucial question in the learning theory and empirical process.  By the same method of Lemma B.10 in our manuscript, we can give an answer. More specifically, we have shown that
> $$\mathbb{E}\_{D\_s}\big[\vert\mathcal{L}(\widehat{f},\widehat{G}) - \widehat{\mathcal{L}}(\widehat{f},\widehat{G})\vert\big] \lesssim\mathcal{O}\Big(\frac{1}{\sqrt{n\_s}}\Big),$$
> where $\widehat{f}$ and $\widehat{G}$ are estimators defined by ACT. It is evident that both $\widehat{f}$ and $\widehat{G}$ depend on the data. This inequality shows that the gap between expectation of $\widehat{\mathcal{L}}(\widehat{f},\widehat{G})$ and $\mathcal{L}(\widehat{f},\widehat{G})$ vanishes as the number of samples goes to infinity. The inequality is known as the generalization error bound.
> - The unbiasedness of the sample-level loss, as stated in the first point, is crucial for establishing the generalization error bound as the point two. One of our contributions in this work is constructing an unbiased sample-level loss, which enables us to provide a generalization analysis for our method.

---

> ### Author Response · Authors · 2024-11-30
> **Official Comment by Authors**
>
> With the discussion period extended until December 2nd, we again appreciate the contributions of all reviewers.
>
> If there are any questions so far, please feel free to contact us.
>
> Warm regards,
>
> The Authors of Submission3353

---

### Official Review · Reviewer_BFRW · 2024-11-04

**Soundness:** 4
**Presentation:** 3
**Contribution:** 3
**Rating:** 8
**Confidence:** 5

**Summary:**

This paper proposes a new contrastive self-supervised technique via ACT for transfer learning with both practical results and theoretical guarantees. The model seems to be motivated by the theoretical challenge posed by the non-commutativity between expectation and the Frobenius norm, inspiring the authors to vary the mathematical formulation into an adversarial form and adopt an alternative optimization algorithm to solve it. Moreover, they give an rigorous  analysis at  the sample level. The results reveal that the misclassification rate solely relies on the efficacy of data augmentation with high probability when the pretraining data is sufficiently large, despite the amount of data from the downstream task is quite limited as the order with respect to downstream sample size is $\frac{1}{\sqrt{\min_k n_t(k)}}$, which coincides with the definition of $K$-way, $\min_k n_t(k)$-shot learning.

**Strengths:**

- The way of rewriting the Frobenius norm in an adversarial form is very ingenious, which allows them to conduct error decomposition for excess risk without the assumption that $f^* \in \mathcal{F}$, referred to as the misspecified setting, which is fundamentally different from previous theoretical works. Meanwhile, in practice, this method introduces a novel discretization approach for problems where the expectation is taken within the Frobenius norm.
- This paper construct a representation $\tilde{f}$ making $\mathcal{L}(\tilde{f})$ vanished under a mild assumption 4.5, giving a great understanding to the population optimizer. The misspecified setting allows them to handle the task without having to consider the expressiveness of the network, thus eliminating the need for strong assumptions.
- The theoretical result of this paper looks solid and impressive. Under the case of domain adaptation, the final theoretical result clarified the role of sample size from both the source domain and the target domain. It reveals that, under a strong data augmentation, the downstream classification error will be small enough even when the sample size from the target domain is pretty limited, which can be regarded as a strong explanation for few-shot learning.
- The indication for the width $W$ and depth $L$ can explain the reason why over-parameterized setting is effective in practice.

**Weaknesses:**

- The title of Table 2 should be upon table.
- It is  better to consider dedicating some space to provide a brief explanation of misspecified and over-parameterized setting. Additionally, it would be beneficial to clarify how the final theoretical results achieve over-parameterization.
- The definition of latent classes lacks clarity. The semantics of latent classes $C_s(k)$ should be determined by the semantic meaning of $C_t(k)$ in downstream tasks when considering the domain adaptation scenario.

**Questions:**

- I suggest that the numerical comparison with BarlowTwins would be  added  since the proposed  model is a little bit similar to theirs. By the way, it would be  better to  add  additional experiments with  domain adaptation although  the existing experiments are acceptable for this paper.
- Why do you adopt the extra requirement $P_t(\cup_{k=1}^KC_t^*(k)) \geq \sigma_t$ for augmentation instead of **Correctly augmented parts** (See Definition 2 in Appendix F of [1]) to tackle the case where the $\mathcal{A}(C_k)$ intersects?
- Is it possible to obtain a better order with respect to $n_s$ for downstream classification error is still unclear. I mean, whether a tighter error bound exists remains an unresolved issue.

If the authors can appropriately address the weaknesses and questions I raised, I will consider updating my score.

[1] Towards the generalization of contrastive self-supervised learning. ICLR 2023.

---

> ### Author Response · Authors · 2024-11-22
> **Reply to Reviewer BFRW**
>
> We thank you for your thorough review of our manuscript and for your constructive suggestions. We have revised our manuscript by incorporating all of your suggestions. The revised version is much strengthened. Our point-by-point responses to your comments are given below.
>
> >**W1**. The title of Table 2 should be upon table.
> - Thank you for reminding, we've revised the title's position.
>
> > **W2**. It is better to consider dedicating some space to provide a brief explanation of miss-specified and over-parameterized setting. Additionally, it would be beneficial to clarify how the final theoretical results achieve over-parametrization.
>
> - The conditions of Theorem 4.2 only require $W \geq \mathcal{O}(n_s^\frac{2d + \alpha}{4(\alpha + d + 1)}), L \geq \mathcal{O}(1)$ and $\mathcal{K} = \mathcal{O}(n_s^{\frac{d + 1}{2(\alpha + d + 1)}})$. **Thus the number of network parameters could be arbitrarily large if we control the norm of weight properly.**
> - A well-specified setting assumes that the target function belongs to a parameterized hypothesis space, such as a class of linear models, RKHS, or neural networks. However, **this setting may fail to represent the actual situation when we have no prior knowledge in realistic scenarios**. A misspecified setting, on the other hand, can make our model more robust. It assumes only that the true underlying target function belongs to some general functional class, for instance, Hölder class or Sobolev space.
>
> > **W3**. The definition of latent classes lacks clarity. The semantics of latent classes $C_s(k)$ should be determined by the semantic meaning of $C_t(k)$ in downstream tasks when considering the domain adaptation scenario.
>
> - In the scenario where domain adaptation is slight, the latent classes represent unobserved labels of upstream data, which exist but cannot be accessed. For example, $C_s(k)$ and $C_t(k)$ represent "dog" and "sketch dog," respectively.
>
> > **Q1**. I suggest that the numerical comparison with BarlowTwins would be added since the proposed model is a little bit similar to theirs. By the way, it would be better to add additional experiments with domain adaptation although the existing experiments are acceptable for this paper.
>
> - Thanks for your suggestion. We have added the comparison with BarlowTwin.
>
> | Method          | CIFAR-10 Linear | CIFAR-10 5-NN | CIFAR-100 Linear | CIFAR-100 5-NN | Tiny ImageNet Linear | Tiny ImageNet 5-NN |
> |-----------------|-----------------|----------------|------------------|----------------|----------------------|--------------------|
> | BarlowTwins[3] | 87.76           | 86.66          | 61.64            | 55.94          | 41.80                | 33.60              |
> | ACT             | **92.11**       | **90.01**      | **68.24**        | **58.35**      | **49.72**            | **36.40**          |
>
> We can see that **our method outperform the BarlowTwins**, despite the BarlowTwins adopted a larger embedding size ($1024$) compared to the embedding sizes of our method $(64, 64, 128)$ for the three datasets, respectively.
>
> > **Q2**. Why do you adopt the extra requirement $P_t(\cup_{k=1}^K C_t^*(k)) \geq \sigma_t$ for augmentation instead of **Correctly augmented parts** (See Definition 2 in Appendix F of [1]) to tackle the case where the intersects?
>
> - The reason is that we generalized the setting of [2], altering the discriminative matrix $W$ at the population level in [2] to its estimator $\widehat{W}$ in order to explore the role of the downstream sample size $n_t$. If we follow the definition of correctly augmented parts directly, **$\widehat{W}$ becomes non-computable** because we lack information about which samples belong to $\tilde{C}_k$ according to the definition of Correctly augmented parts.
>
> > **Q3**. Is it possible to obtain a better order with respect to $n_s$ for downstream classification error is still unclear. I mean, whether a tighter error bound exists remains an unresolved issue.
>
> - Maybe there is some space for improvement regrading the order of $n_s$. However, to the best of our knowledge, the minimax optimal rate of the self-supervised transfer learning is still unknown.
>
> > **Summary**
> - In summary, thanks for your hard work, careful reading, valuable feedback as reviewer. If there are any further questions, please feel free to contact us.
>
> > **Reference**
>
> [1] A theoretical study of inductive biases in contrastive learning. ICLR, 2023.
>
> [2] Towards the generalization of contrastive self-supervised learning. ICLR 2023.
>
> [3] Guarding Barlow Twins Against Overfitting with Mixed Samples. arXiv preprint arXiv:2312.02151 (2023).

---

> > ### Comment · Reviewer_BFRW · 2024-11-23
> >
> > Thank you for your rebuttal and the effort you put into addressing the comments raised. I appreciate your careful revisions and the additional clarifications provided in the updated manuscript. The explanations, particularly regarding the misspecified and over-parameterized settings, and the inclusion of numerical comparisons with BarlowTwins, have strengthened the paper.
> >
> > I have also reviewed the other reviewers' feedback. After considering their insights alongside my own assessment, I believe this paper is deserving of acceptance. It makes both theoretical and empirical contributions to the field. Based on the improvements and its merits, I would like to increase my score to reflect my support for accepting this paper.

---

> > > ### Author Response · Authors · 2024-11-23
> > > **Reply to Reviewer BFRW**
> > >
> > > We greatly appreciate your valuable feedbacks, which are essential for improving the quality of our paper.

---

### Author Response · Authors · 2024-11-28
**Follow-up on Your Feedback for Submission 3353**

Dear Reviewers,

We would like to sincerely thank you for your thoughtful and detailed review, as well as the constructive feedback you have provided for our submission, Submission3353: Self-supervised Transfer Learning via Adversarial Contrastive Training.

As we move forward with the rebuttal phase, we are committed to addressing your concerns thoroughly and clarifying any potential misunderstandings. If there are any additional points you would like us to elaborate on, please do not hesitate to reach out.

We look forward to your feedback!

Warm regards,

The Authors of Submission3353

---

### Meta-Review · Area_Chair_1pEn · 2024-12-20

**Metareview:**

This paper focuses on self-supervised contrastive learning that eliminates the need for negative samples. The authors propose a contrastive learning loss through adversarial contrastive training. Additionally, the authors provide both theoretical analysis to guarantee its performance and empirical evaluation results. The reviewers acknowledge the theoretical contributions of this paper, including rewriting the Frobenius norm in an adversarial form (Reviewer BFRW) and the detailed theoretical analysis (Reviewers BFRW, Gv21, and dVpK). However, reviewers also raised significant concerns about both the theoretical and empirical aspects of the paper. These include the misspecified and over-parameterized setting (Reviewer BFRW), lack of clarity in the definition of latent classes (Reviewer BFRW), insufficient discussion of other methods that do not require negative samples (Reviewer Gv21), the need for more rigorous analysis of the bias in the proposed loss function (Reviewer Gv21), missing baselines (Reviewers Gv21 and dVpK), and unclear writing (Reviewer dVpK).

In the rebuttal, the authors provided additional analysis and experimental results. While Reviewer BFRW acknowledged that their concerns were well addressed, other reviewers continued to express reservations regarding the bias of the proposed loss function, its differences from related work, and the significance of its advantages over existing baselines. Considering the reviewers' opinions and the unresolved concerns mentioned above, this paper requires further improvement and cannot be accepted at this time.

**Additional Comments On Reviewer Discussion:**

The reviewers and authors have dedicated significant effort during the discussion period. Reviewer BFRW's main concerns included the misspecified and over-parameterized setting, lack of clarity in the definition of latent classes, and the placement of the title for Table 2. These concerns were adequately addressed by the authors, as acknowledged by Reviewer BFRW.

Reviewer Gv21's concerns, however, were not sufficiently resolved. The reviewer continues to express significant reservations regarding the bias in the proposed loss function, its differences from related work, and the significance of its advantages over other baselines. Reviewer dVpK raised concerns about writing clarity and performance. The authors responded by providing additional explanations and experiments. While Reviewer dVpK did not actively participate in the discussion, their opinion was properly considered but weighed less heavily compared to those of the other reviewers when making the final decision.

Taking all points into account, and given that some concerns remain unresolved, the Area Chair recommends further improvements before this paper can be accepted.

---

### Decision · Program_Chairs · 2025-01-22

Reject